# Detecting and quantifying clonal selection in somatic stem cells

Verena Körber [1,2] ✉, Niels Asger Jakobsen [2], Naser Ansari-Pour[2,3], Rachel Moore[2,4], Nina Claudino[1], Marlen Metzner[2], Eva Thielecke [1,2], Franziska Esau[1], Batchimeg Usukhbayar[2], Mirian Angulo Salazar[2], Simon Newman[3,5], Benjamin J. L. Kendrick[3,5], Adrian H. Taylor[3,5], Rasheed Afinowi-Luitz[5], Roger Gundle[3,5], Bridget Watkins[3], Kim Wheway[3], Debra Beazley[3], Stephanie G. Dakin[3], Antony Palmer[3,5], Andrew J. Carr[3], Paresh Vyas [2,6] ✉ & Thomas Höfer [1,6] ✉

As DNA variants accumulate in somatic stem cells, become selected or evolve neutrally, they may ultimately alter tissue function. When, and how, selection occurs in homeostatic tissues is incompletely understood. Here, we introduce SCIFER, a scalable method that identifies selection in an individual tissue, without requiring knowledge of the driver event. SCIFER also infers self-renewal and mutation dynamics of the tissue's stem cells, and the size and age of selected clones. Probing bulk whole-genome sequencing data of nonmalignant human bone marrow and brain, we detected pervasive selection in both tissues. Selected clones in hematopoiesis, with or without known drivers, were initiated uniformly across life. In the brain, we found pre-malignant clones with glioma-initiating mutations and clones without known drivers. In contrast to hematopoiesis, selected clones in the brain originated preferentially from childhood to young adulthood. SCIFER is broadly applicable to renewing somatic tissues to detect and quantify selection.

Normal stem cells accumulate somatic variation throughout development[1] and postnatal life[2–8]. Over time, stem cell clones vary in size owing to genetic drift[9,10], or selection[4–6,11–17]. Quantifying proliferation, differentiation, somatic mutation and selection dynamics in tissue stem cells is fundamental to our understanding of somatic mosaicism.

Clonal hematopoiesis (CH), the expansion of hematopoietic stem cell (HSC) clones, is a paradigm for somatic mosaicism. CH, detected with increased frequency as humans age, is associated with known driver events: either somatic variants in genes recurrently mutated in myeloid cancers (for example, *DNMT3A*, *TET2* and *ASXL1*), or chromosomal copy number alterations[18–26]. CH drivers not only provide a

selective advantage to HSCs[15,27], but also generate proinflammatory myeloid cells[28,29]. CH with known drivers is associated with a heightened risk of blood cancer[19,20,30], excess cardiovascular mortality[31], chronic pathology and infection[32–36]. Mathematical analysis of somatic variant distributions predicts that CH is far more prevalent than suggested by known drivers[14]; expanded clones without known driver events are possibly more frequent than those with known driver events[12,13].

These observations raise the following questions: (1) Do CH drivers emerge by chance, or does increased turnover and/or somatic mutation rate facilitate their acquisition? (2) Do stem cell clones with known and unknown drivers differ in their frequency and clinical phenotypes?

[1]Division of Theoretical Systems Biology, German Cancer Research Center (DKFZ), Heidelberg, Germany. [2]MRC Molecular Haematology Unit, Oxford Biomedical Research Center, Hematology Theme, Oxford Centre for Haematology, Weatherall Institute of Molecular Medicine, Radcliffe Department of Medicine, University of Oxford, Oxford, UK. [3]Nuffield Department of Orthopaedics, Rheumatology and Musculoskeletal Sciences, Botnar Research Centre, University of Oxford, Oxford, UK. [4]Cambridge Genomics Laboratory, Cambridge University Hospitals, Cambridge, UK. [5]Nuffield Orthopaedic Centre, Oxford University Hospitals NHS Foundation Trust, Oxford, UK. [6]These authors contributed equally: Paresh Vyas, Thomas Höfer. ✉e-mail: verena.korber@ndcls.ox.ac.uk; paresh.vyas@imm.ox.ac.uk; t.hoefer@dkfz.de

(3) When do selection events occur in life, and (4) how fast do the selected clones grow? These questions are best addressed with an affordable, scalable method to identify clonal selection and quantify stem cell dynamics.

Here, we present a method to detect clonal selection and quantify stem cell parameters without knowledge of the driver event, using a bulk primary sample from any tissue. The method applies population genetics theory for the accumulation and spread of somatic variants in development and subsequent homeostatic tissue renewal, and a Bayesian inference framework to discriminate clonal selection from neutral evolution and, moreover, quantify stem cell parameters. We validate our method with synthetic data and published experimental data and then apply it to a new cohort of human nonleukemic bone marrow (BM) samples and a published dataset of brain tissue samples.

## Results

### Quantifying clonal selection

We reasoned that the distribution of somatic variants in stem cells provides information on the dynamics of genetic drift and selection. To derive the expected variant allele frequency (VAF) distribution of a renewing tissue, we modeled expanding stem cell numbers during development, followed by an adult phase in which stem cell numbers are balanced (Fig. 1a and Methods). We focused on somatic single-nucleotide variants (SSNVs) acquired during both phases. In early life, stem cell numbers are small and few SSNVs are acquired, which will be at high VAF (Fig. 1b; variants A and B). Later, when stem cell numbers are larger, more variants are acquired in total but at lower VAF (Fig. 1b; variants C, D and E). Neutral SSNVs drift with time; some randomly increase their VAF (Fig. 1b; variants B, C, D and F) while others decrease or are extinguished (Fig. 1b; variant E). We model neutral clonal dynamics during development and homeostasis by linking expanding (supercritical) and homeostatic (critical) birth–death processes (Methods, 'Population genetics model'). We find an analytical solution for the VAF distribution as a function of time that lends itself to parameter inference from experimental data.

Our model suggests that during development genetic drift is negligible, resulting in an invariant linear relation between cumulative SSNV count and $1/VAF$ (Fig. 1c)[37,38]. During homeostasis, genetic drift alters the cumulative VAF histogram by increasing the number of SSNVs with low VAF (Fig. 1d). The extent of homeostatic drift depends on the number of stem cells and their self-renewal rate. With $5 \times 10^3$ stem cells dividing five times per year, drift increases the count of variants with high VAF over time (Fig. 1e; we assume that the transition from development to homeostasis occurs around birth). For $5 \times 10^4$ stem cells and the same division rate, the variant count will increase with age at lower VAFs and remain stable at high VAFs (Fig. 1f). Finally, with $5 \times 10^5$ stem cells, the distribution is nearly stable with age for all VAFs >1% (Fig. 1g and Extended Data Fig. 1a). Previous work inferred that $4 \times 10^4$ to $2 \times 10^5$ HSCs divide 0.6–6 times per year[2]. With these values, our model suggests that the VAF histogram is insensitive to genetic drift during life at high (≥10%) but not low VAF. Including extensively self-renewing progenitors in the model, as observed in mice[39], only marginally modifies the distribution of somatic variants with ≥1% VAF (Supplementary Note 1), supporting our focus on stem cell dynamics to interpret measured VAF histograms.

Next, we modeled how clonal selection alters the VAF distribution. We consider that the leading selected clone, born at time $t_s$ with selective advantage $s$, expands while displacing normal stem cells (Fig. 1h,i). With clonal selection, neutral variants may have originated: (1) in normal stem cells before the driver event, (2) in the selected clone, or (3) in nonselected stem cells after the driver event occurred. We find analytical expressions for the evolution of all three types of variants, which together yield the cumulative VAF distribution of SSNVs (Methods). Expansion of a selected clone increases the VAF of the founding driver mutation (mutation D in Fig. 1i) and of all other neutral SSNVs in the founding cell, which generates a shoulder in the cumulative VAF

distribution (Fig. 1j). Thus, the shape of the distribution discriminates between selection and neutral evolution, whereas total variant count may fail to do so (compare the VAF histograms in Fig. 1k, shaped by selection, with Fig. 1e,f, shaped by drift).

The shoulder height increases with the age at which the selected clone originated, $t_s$, (Fig. 1k), because the number of neutral variants in the clone's stem cell of origin is the higher the older the individual. As the selected clone expands, the shoulder reaches higher VAFs (Fig. 1l). Thus, the VAF reached between clone origin and sample acquisition yields the selective advantage $s$ (Fig. 1h, lower panel). Collectively, the shoulder in the cumulative VAF distribution provides a robust signature of clonal selection agnostic of driver identity. Moreover, its height and position allow inference of the clone's age and time-averaged selective advantage (Extended Data Fig. 1b,c).

### Selection detected in noisy whole-genome sequencing data

Sufficiently deep whole-genome sequencing (WGS) measures VAFs with approximately binomially distributed error[40]. To understand how this error influences detection of clonal selection in bulk WGS data, we simulated WGS data with our models of neutral evolution (Fig. 1a,b) and selection, focusing on a single selected clone (Fig. 1h,i). We performed repeated stochastic simulations, generating 10 datasets with neutral evolution and 70 datasets with selection (2.5% ≤ VAF ≤ 37.5%). To model measurement error, we drew variant reads from binomial distributions for sequencing depths of 30×, 90× or 270× (Methods and Supplementary Table 1). We then used approximate Bayesian computation (ABC) to fit both the model of neutral evolution and of selection to each dataset (Extended Data Fig. 1d), yielding posterior probabilities for both models to reproduce the noisy data.

To determine sensitivity and specificity in detecting selection, we used the posteriors to compute receiver operating curves (ROC) for identifying clones of different sizes. At 90× sequencing depth, we found that selected clones with VAF ≥ 5% were detected reliably (Fig. 2a; area under the curve (AUC) = 0.96 for VAF = 5% and AUC ~ 1 for VAF = 7.5%), whereas smaller clones were not (Fig. 2a; VAF = 2.5%; AUC = 0.4). Discrimination between selection and neutral evolution was optimal at a threshold of 15% for the posterior probability of the selection model (Fig. 2a, operating points correspond to the 15% threshold, achieving ~100% sensitivity and 90% specificity for clone size 5% VAF and ~100% sensitivity and specificity for clone size 7.5% VAF). Exemplary selection posteriors for simulated 90× WGS datasets show discrimination between neutral evolution and selection of clones ≥5% VAF (Fig. 2b). Further, our ROC analysis predicts that 30× WGS will allow detection of large, selected clones >20% VAF, while deeper WGS enables detection of smaller clones with VAF > 1% at 270× (Fig. 2c).

Taken together, our method for selected clone inference, termed SCIFER, is suitable for detecting clonal selection in bulk WGS data.

### SCIFER identifies sequential selected clones

Next we extended SCIFER beyond a single leading selected clone to study two competing clones, born at times $t_{s1}$ and $t_{s2}$, and conferring a selective advantage of, respectively, $s_1$ and $s_2$ (Fig. 2d). Considering branched (Extended Data Fig. 2a) and linear evolution (Extended Data Fig. 2b), we found that, in both cases, the VAF distribution has two subclonal shoulders, generated by the two competing clones (Extended Data Fig. 2c,d). As with a single clone, the shoulders have heights proportional to the clonal birth dates that increase in VAF according to the fitness of the selected clones (Fig. 2e). We found that the two-clone inference model is particularly appropriate for the resolution of deeply sequenced WGS data (270×).

### Benchmarking SCIFER with human hematopoiesis phylogenies

We benchmarked SCIFER with published WGS data from human hematopoietic stem and progenitor cell (HSPC) single-cell clones[2,12,13].

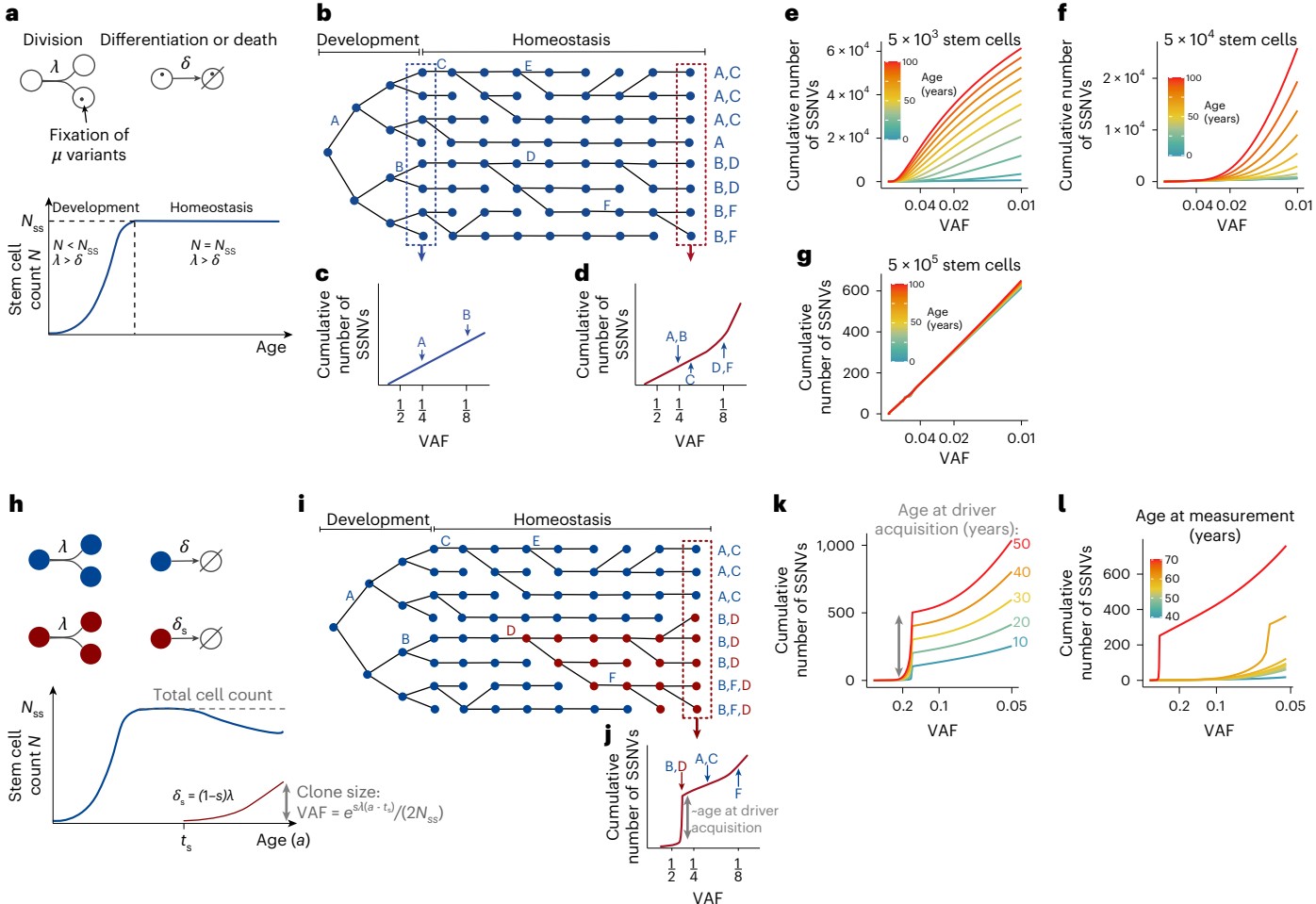

**Fig. 1 | Population genetics model of drift and selection in homeostatic tissues. a**, Modeled processes and associated parameters in the model of drift. Stem cells either divide symmetrically with rate $\lambda$, or exit the stem cell compartment by differentiating (or dying), with rate $\delta$. The stem cell count ($N$) increases in development ($\lambda > \delta$) until reaching steady-state numbers ($N_{ss}$) and remains constant during adulthood ($N = N_{ss}$ and $\lambda = \delta$). On average, cells acquire $\mu$ neutral variants during each cell division. **b**, Schematic illustrating variant accumulation during development and subsequent homeostasis. **c,d**, Cumulative number of SSNVs versus VAF in development (**c**) and in adult life (**d**) (the scaling of the $x$ axis is transformed to $\frac{1}{VAF}$ to spread out low-frequency variants). **e–g**, Simulated cumulative VAF distribution of SSNVs at selected ages between 0 and 100 years for $5 \times 10^3$ stem cells ($\lambda = 5$ per year, $\mu = 10$ per division) (**e**); for $5 \times 10^4$ stem cells ($\lambda = 5$ per year, $\mu = 10$ per division) (**f**); and for $5 \times 10^5$ stem cells ($\lambda = 5$ per year, $\mu = 10$ per division) (**g**). **h**, Model of clonal selection. A selective driver event

reduces the loss rate (differentiation or death) by a factor $s$, causing selective outgrowth of the mutant clone (red); the remaining parameters are defined in **a**. The VAF of the selected clone increases exponentially with the age at measurement, $a$. **i**, As **b**, but here an acquired driver mutation (D) causes selective outgrowth of the mutant clone (red). **j**, All variants in the selected clone's cell of origin are inherited by its progeny and hence reach a high VAF during clonal expansion, reflected in a shoulder in the cumulative VAF distribution. **k**, Simulated cumulative VAF distributions when a driver mutation is acquired at different ages, and the SSNVs are measured 45 years later, when the clone has reached a size of 32%. In the simulation, the selected clone grows by 22% per year ($s = 0.02$, $\lambda = 10$ per year, $\mu = 1$ per division, $N_{ss} = 25,000$). **l**, Simulated cumulative VAF distributions measured at varying ages after a driver mutation was acquired at 20 years of age. As **k**, the selected clone grows by 22% per year ($s = 0.02$, $\lambda = 10$ per year, $\mu = 1$ per division, $N_{ss} = 25,000$).

Previously determined HSC parameters of neutral hematopoiesis (HSC number, self-renewal rate and rate of somatic variants acquisition)[2] provided a quantitative reference for SCIFER. Because variant calling is a source of error in WGS, we used a stringent approach intersecting the results of two callers, Mutect2 and Strelka[41] (Extended Data Fig. 3a). To examine the robustness of SCIFER with respect to variant calling, we also performed our analyses on the original variant calls obtained with Caveman[2], which included substantially more SSNVs (Extended Data Fig. 3b,c). Reconstructing the phylogenetic tree with our calling approach recapitulated the original work (Fig. 3a). Applying SCIFER to pseudo-bulk data generated from single cells (Fig. 3b and Extended Data Fig. 3d) detected neutral evolution, irrespective of the variant calling algorithm, in agreement with the published findings (Fig. 3c). The inferred HSC number (Fig. 3d) and rate of self-renewing divisions (Fig. 3e) were similar for the two variant callers ($6 \times 10^4$ to $2 \times 10^5$ HSCs

dividing 3–8 times per year using Caveman; $2 \times 10^4$ to $5 \times 10^4$ HSCs dividing 0.4–6 times per year using Mutect2 and Strelka; ranges are 80% credible intervals). Both estimates overlapped with the previously published numbers ($4 \times 10^4$ to $2 \times 10^5$ HSCs dividing 0.6–6 times per year)[2]. The rate of SSNV acquisition inferred by SCIFER using Caveman caller (3–5 SSNVs per cell division; Fig. 3f) overlapped with the original estimate (3–28 SSNVs per division). As expected, this rate was smaller (0.6–1 SSNV per cell division) with our more conservative calling approach (intersecting Mutect2 and Strelka).

SCIFER infers HSC number, self-renewal rate and mutation rate separately (Extended Data Fig. 3e,f), which is based on: (1) using both expansion and homeostatic phases (Supplementary Note 2 and Supplementary Fig. 2), assuming a homogeneous mutation rate throughout; and (2) the use of absolute variant counts, because many rapidly dividing stem cells accumulate more variants than few rarely dividing stem cells

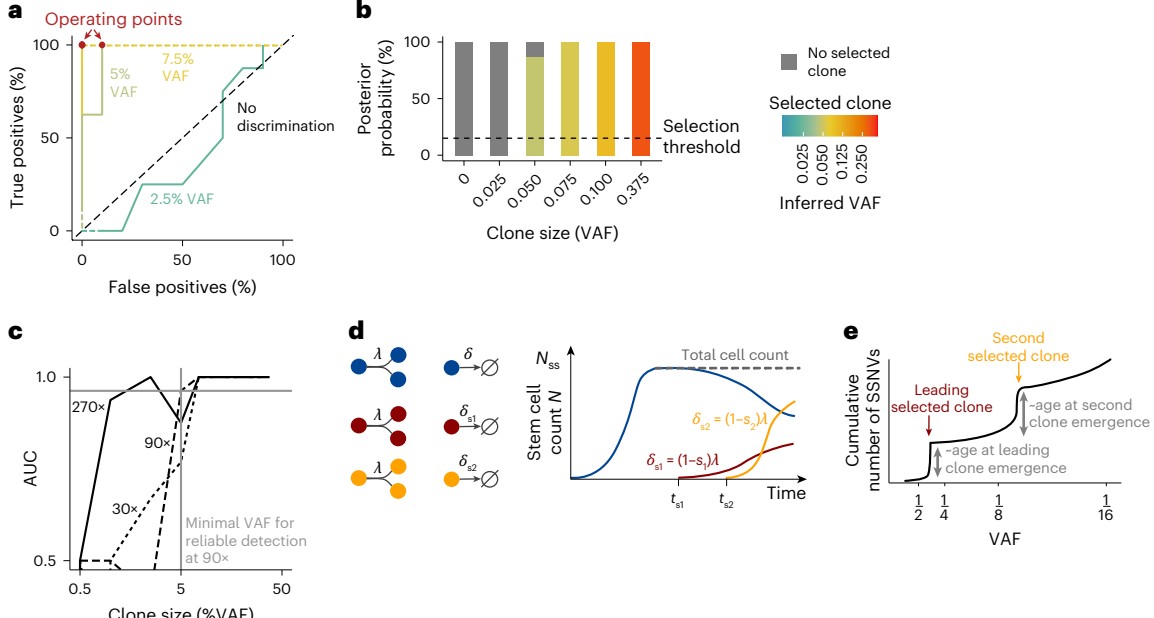

**Fig. 2 | Benchmarking SCIFER with simulated data. a**, ROCs quantifying the detection of clonal selection by SCIFER for different clone sizes (color-encoded). ROCs were generated by applying SCIFER to simulated data, generated with a stochastic birth–death process with ($s$ = 0.02, corresponding to a selective advantage of 2% increase in birth versus death) or without ($s$ = 0) selection of a clone initiated at 20 years of life ($\lambda$ = 10 per year, $\mu$ = 1 per division, $N_{ss}$ = 25,000, and assuming sequencing with an average coverage of 90×). In total 63 cases, with selected clone sizes of VAF 0%, 2.5%, 5%, 7.5%, 10%, 25% and 37.5%, were generated. Models with, or without, clonal selection were fit to the data using ABC. True positives and false positives were evaluated for varying posterior probability thresholds of clonal selection. For selected clones with VAF ≥ 5%, the difference between true positives and false positives was maximal for a selection threshold of 15% (operating points, shown in red). **b**, Posterior probability for clonal selection (colored bars) and neutral evolution (gray bars) conditioned on selected clones with VAF ≥ 5% for six simulated cases with varying clone size. The dashed line marks the selection threshold at 15% conditional posterior probability. **c**, Accuracy of SCIFER to distinguish clonal selection from genetic drift in simulated WGS data. Shown are AUC computed from the ROCs shown in **a** and from ROCs obtained in analogy for simulated sequencing depths of 30× and 270×. The simulated data were generated as in **a**. For 270× sequencing depth, an additional 17 cases with selected clone sizes of 0.5% VAF and 1% VAF were used for model evaluation. **d**, Model scheme for two selected clones (red and orange) that compete with normal cells (blue). The two selected clones are born at times $t_{s1}$ and $t_{s2}$, and expand due to decreased loss rates ($\delta_{s1}$ and $\delta_{s2}$); the total cell count remains constant over time. **e**, Selection of two sequential clones manifests itself in two subclonal shoulders whose heights scale with the time points of driver acquisition.

(although the ratio of stem cell number and self-renewal rate may remain unchanged[2,15]). Thus, SCIFER detected neutral evolution and quantified HSC number and division rate from a single pseudo-bulk sample.

To further validate SCIFER, we generated additional pseudo-bulk data from three neutrally evolving, published human HSPC single-cell WGS datasets (AX001, KX001 and KX002)[13] (Extended Data Fig. 3g,h). In all cases, SCIFER detected neutral evolution (Fig. 3c) and inferred similar HSC parameters (Fig. 3d,f). Interestingly, for the oldest individual (AX001, 63 years old), the inferred HSC division rate was higher than for the other individuals (29 and 38 years old) (Fig. 3e).

Next, we asked whether SCIFER detects clonal selection in pseudo-bulk WGS data from single HSPC clones from two published samples, KX0004 (ref. 13) (*DNMT3A* mutation, ~6%–8% VAF; Fig. 3g) and id2259 (ref. 12) (*SF3B1* mutation, nearly 50% VAF; Extended Data Fig. 3i). SCIFER identified the leading *DNMT3A* (VAF 6%) and *SF3B1* (VAF 49%) clones (Fig. 3h and Extended Data Fig. 3i) as selected (Fig. 3i). Moreover, using the two-clone model, SCIFER also detected the second largest clone in KX004 (Extended Data Fig. 3j,k). The original studies estimated that the *SF3B1* mutation arose 25–30 years before tissue sampling, and the *DNMT3A* mutation 45–50 years before tissue sampling. SCIFER determined very similar clone ages (Fig. 3j). The estimated clonal growth rates (Fig. 3k) were concordant with previous data suggesting stronger selection of *SF3B1* mutant clones[42]. Inferred stem cell parameters (Fig. 3l) were similar to individuals without CH (Fig. 3d–f). Finally, we applied SCIFER to a published sample without known CH driver, KX003 (ref. 13), having the largest expanded clone with ~17% VAF (Fig. 3m). SCIFER detected a leading selected clone at this VAF (Fig. 3n,o). Stem cell and selection parameters were similar to the samples above with known CH driver mutations (Fig. 3p, compare with Fig. 3j–l).

In summary, SCIFER robustly identified and quantified neutral evolution and selection in pseudo-bulk WGS data of human HSPCs, without previous knowledge of a driver event.

## SCIFER uncovers clonal complexity in human BM

Next, we generated genuine bulk WGS data from BM HSPCs of 22 humans, aged 30–89 years (Supplementary Table 2). Based on targeted sequencing (with VAF ≥ 1% sensitivity; Supplementary Table 3)[27], we selected 12 individuals with known CH drivers (6 with *DNMT3A*, 4 *TET2*, 1 *ASXL1* and 1 with both *TET2* and *ASXL1* mutations), and 10 individuals without known drivers. The samples are labeled 1–22, followed by letters for the clones detected (D, T, A or U, for, respectively, mutations in *DNMT3A*, *TET2* or *ASXL1*, or unknown drivers; N for neutral evolution). If SCIFER detected two clones, the larger clone is indicated first. All individuals had normal blood counts and blood films, without an antecedent history of blood or inflammatory disorder[27].

We performed WGS of Lin⁻CD34⁺ HSPCs (Extended Data Fig. 4a–f) at 90× for all cases, and for 19 cases with sufficient DNA at 270×; hair follicle DNA at 30× WGS served as a germline control. A comparison with WGS data from 14 BM MNC samples is given in Supplementary Note 3 and Supplementary Fig. 3. WGS data were concordant with panel sequencing. Somatic single base-pair substitution profiles indicated similar age-related variant accumulation across all individuals (Extended Data Fig. 5)[2,13,43]. We found no copy number aberrations (CNAs) in driver genes (Extended Data Fig. 6a,b and Supplementary Table 4).

We first applied SCIFER to the 12 individuals with mutations in known CH drivers (in total 14 such mutations, with 2 samples containing two, 17-TT and 12-AT). SCIFER identified a selected clone in all samples (both at 90× and 270×), associated with a shoulder in the VAF histogram (Fig. 4a–c and Extended Data Fig. 7a,b). In all but one case, the clone detected by SCIFER, without knowledge of a putative driver, agreed with the VAF of the CH driver detected by panel sequencing (Fig. 4d). Only for 21-DU, SCIFER did not detect the *DNMT3A* clone, but instead a subclone without a known CH driver (Extended Data Fig. 7b). The *DNMT3A* clone originated from a founding cell with only six SNVs (Extended Data Fig. 7b), likely during embryogenesis. This case indicates a practical detection limit for clones of very early origin. Taken together, SCIFER reliably detected selected clones with known drivers originating in postnatal life.

Applied to the 270× data, SCIFER found second clones in seven of ten samples, associated with a second shoulder in the VAF histogram (Fig. 4e,f; Extended Data Fig. 7c). Two secondary clones were associated with mutations in *TET2* (included in Fig. 4d), whereas five had no known driver (Supplementary Table 5 and Methods). Thus, clones without known drivers were abundant and, in all but one case (20-UT), had lower VAF than clones with known drivers.

We next asked whether HSC dynamics varied with mutations in different driver genes. The total number of HSCs, their division rate and the rate of SSNV acquisition were in similar ranges across the three common CH driver genes (Fig. 4g). Allowing for *TET2* mutations to increase the murine HSPC pool[44,45] did not change the inferred parameters (Extended Data Fig. 7d–f). Moreover, parameter estimates did not depend on sequencing coverage (Fig. 4g), or whether the leading-clone or two-clone SCIFER models were used (Extended Data Fig. 7g). For 5-DU and 10-D, HSC numbers were inferred with 90× data to be smaller than for the other cases. For 10-D, the selected clone was at the lower end of identifiable VAFs, and deeper sequencing made its stem cell number agree with the majority of cases. The 5-DU parameter estimates, by contrast, were consistent outliers for 90× and 270× and also had a consistently smaller $N$ to $\lambda$ ratio (Fig. 4h). Apart from this case, we inferred that $1.6 \times 10^5$ to $1 \times 10^7$ HSCs divided 1.4–14 times per year, acquiring 1–5 SSNVs per division. Moreover, the $N$ to $\lambda$ ratio, giving the timescale over which neutral evolution will cause a variant to reach fixation, ranged from $1.4 \times 10^4$ to $22 \times 10^6$ years (Fig. 4h), confirming that large clones arose by selection, not drift. Taken together, SCIFER identifies up to two selected clones and quantifies the stem cell parameters from bulk WGS data.

### Clonal selection without known CH drivers

Next we studied the ten cases in which targeted sequencing did not detect known CH mutations, at 90× and 270× (Fig. 5a) (30–76 years of age).

At 90×, SCIFER found clear evidence for clonal selection in three cases, and borderline evidence in one sample (3-N; Fig. 5b and Extended

Data Fig. 8a). Probing at 270× ruled out selection in 3-N (Fig. 5c), and identified three additional samples with selection (11-UU, 13-U, 14-U), as well as additional smaller clones in 8-UU, 11-UU and 16-UU (Fig. 5c,d and Extended Data Fig. 8b). None of the leading and secondary clones had known drivers (Supplementary Table 5). Thus, in six of ten cases without known CH drivers, SCIFER discovered one or more selected clones.

Although all three samples with a large SSNV count (11-UU, 13-U, 16-UU; 500–800 SSNVs) contained selected clones, several samples with one or two selected clones had SSNV counts indistinguishable from those of the four neutrally evolving samples (1-N, 3-N, 4-N, 15-N). This suggests that using bulk SSNV count alone is insufficient to detect CH.

For all neutral samples, SCIFER inferred the $N$-to-$\lambda$ ratio to be ~$5 \times 10^4$ to $10^6$ years, further supporting the notion that clones do not drift to large size in normal human hematopoiesis[15] (Fig. 5e). Moreover, we inferred $10^5$ to $3 \times 10^6$ HSCs, an HSC division rate of 1–14 times a year and a rate of acquired SSNV per cell division of 1–4 (Fig. 5f). These parameter estimates were insensitive to sequencing depth and consistent with our inferences from neutral phylogenetic trees (Fig. 3d–f), and from the cases with selection of unknown drivers (Fig. 5g,h). However, the division rate of the whole HSC population showed more variation in cases with selection (compare the middle panels in Fig. 5f and h). Hence, we compared HSC parameters in all samples with selection (including the cases in Fig. 4) versus neutrally evolving samples. The rate of SSNV acquisition and HSC numbers were overall very similar (Fig. 5i,j). The HSC division rates were more varied, and 11 of 18 individuals with CH had a high inferred division rate (Fig. 5k). Hence, HSC division rate appears to be more variable between individuals and, in some cases, clonal selection was associated with high division rate.

### Selection across life

Phylogenetic reconstruction has previously suggested that expanded clones originate before 40 years of age[13]. SCIFER estimated that clones typically emerge several decades before BM sampling (Extended Data Fig. 9a,b) with no significant differences between clones with and without known drivers (Fig. 6a). Secondary (smaller) clones generally emerged later (Fig. 6b). The time-averaged clonal growth rates showed substantial inter-individual variation, especially in cases with unknown drivers (Fig. 6c and Extended Data Fig. 9c,d). Combining all individuals with selection, we found that selected clones were born at approximately constant rate from early childhood to about 50 years of age (Fig. 6d; the leveling-off at 50 years likely reflects a technical limit of clone detection), suggesting that selection events in HSCs occur uniformly across life.

### Pervasive selection in the juvenile human brain

We applied SCIFER to bulk WGS data (average coverage 175× ± 70) for 185 human brain samples from 131 individuals aged 4–95 years[17]. This cohort comprised samples from different brain regions (cortex, striatum, hippocampus) from 44 normal (neurotypical) individuals and 87

**Fig. 3 | Benchmarking SCIFER with published pseudo-bulk data.**
**a**, Reconstructed single-cell phylogenies after re-calling SSNVs and indels from single-cell WGS data[2]. **b**, Left, VAF distribution of SSNVs shown in **a**, truncated at 1%. Right, model fit to the cumulative $\frac{1}{\text{VAF}}$ distribution (points and error bars, measured data and their standard deviation, which, assuming Poisson-distributed measurements, is the square root of the measured data; red area, 95% posterior probabilities of the model fit computed from simulations using 100 posterior samples). **c**, Posterior probability for neutral evolution for pseudo-bulk WGS data from ref. 2 (labeled Lee-Six) and three samples from ref. 13. SCIFER was applied twice to the data from ref. 2, using the SSNV counts obtained with Caveman or with Mutect2 and Strelka. **d–f**, Inferred HSC number (**d**), division rate (**e**) and number of SSNVs per division (**f**) for the cases shown in **c** (median and 80% credible intervals for each sample, estimated from 1,000 posterior samples; gray areas, 95% confidence band for the five estimates obtained with SCIFER). Estimates from ref. 2 are given for comparison. **g**, Single-cell phylogeny of

published sample KX004 (ref. 13). **h**, As **b**, but for the sample shown in **g** (gray area in right panel, 80% credible interval of the estimated clone size computed from 1,000 posterior samples). **i**, Posterior probabilities for selection (conditioned on clones with VAF ≥ 5%) and neutral evolution for samples introduced in **g** and Extended Data Fig. 3i. Dashed line, 15% selection threshold. **j**, Age of leading selected clones in the cases shown in **i**, estimated by SCIFER and by phylodynamic modeling in the original publications (points, median; error bars, 80% credible intervals, estimated from 1,000 posterior samples). **k,l**, Estimated clonal growth rates (**k**) and stem cell parameters (**l**) for the samples shown in **i** (points, median; error bars, 80% credible intervals, estimated from 1,000 posterior samples). **m**, Single-cell phylogenies of published sample KX003 (ref. 13). **n**, As **h** but for the sample shown in **m**. **o**, As **i**, but for sample KX003 (ref. 13). **p**, Estimated stem cell and selection parameters for the sample shown in **m**. Shown are median and 80% credible intervals, estimated from 1,000 posterior samples.

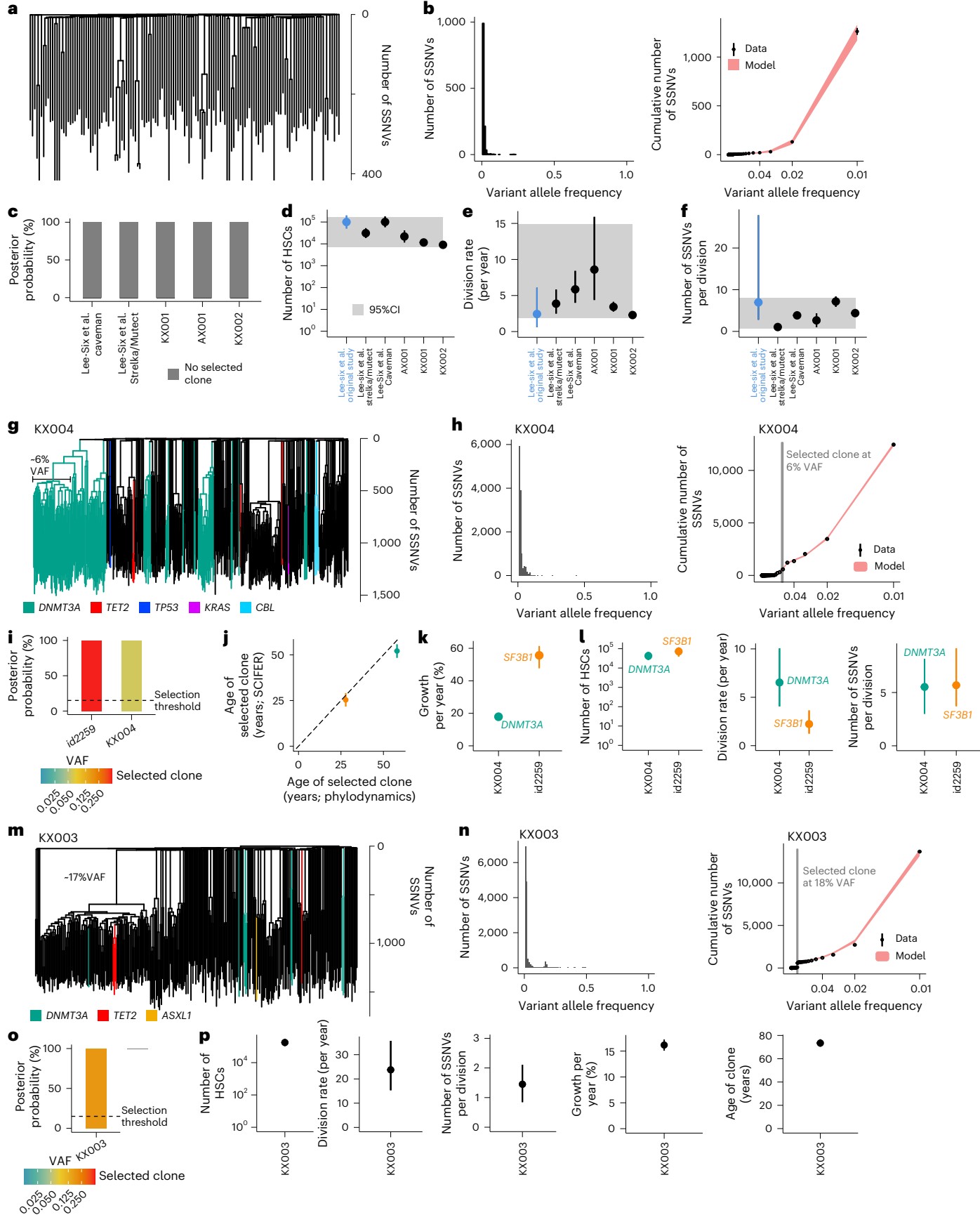

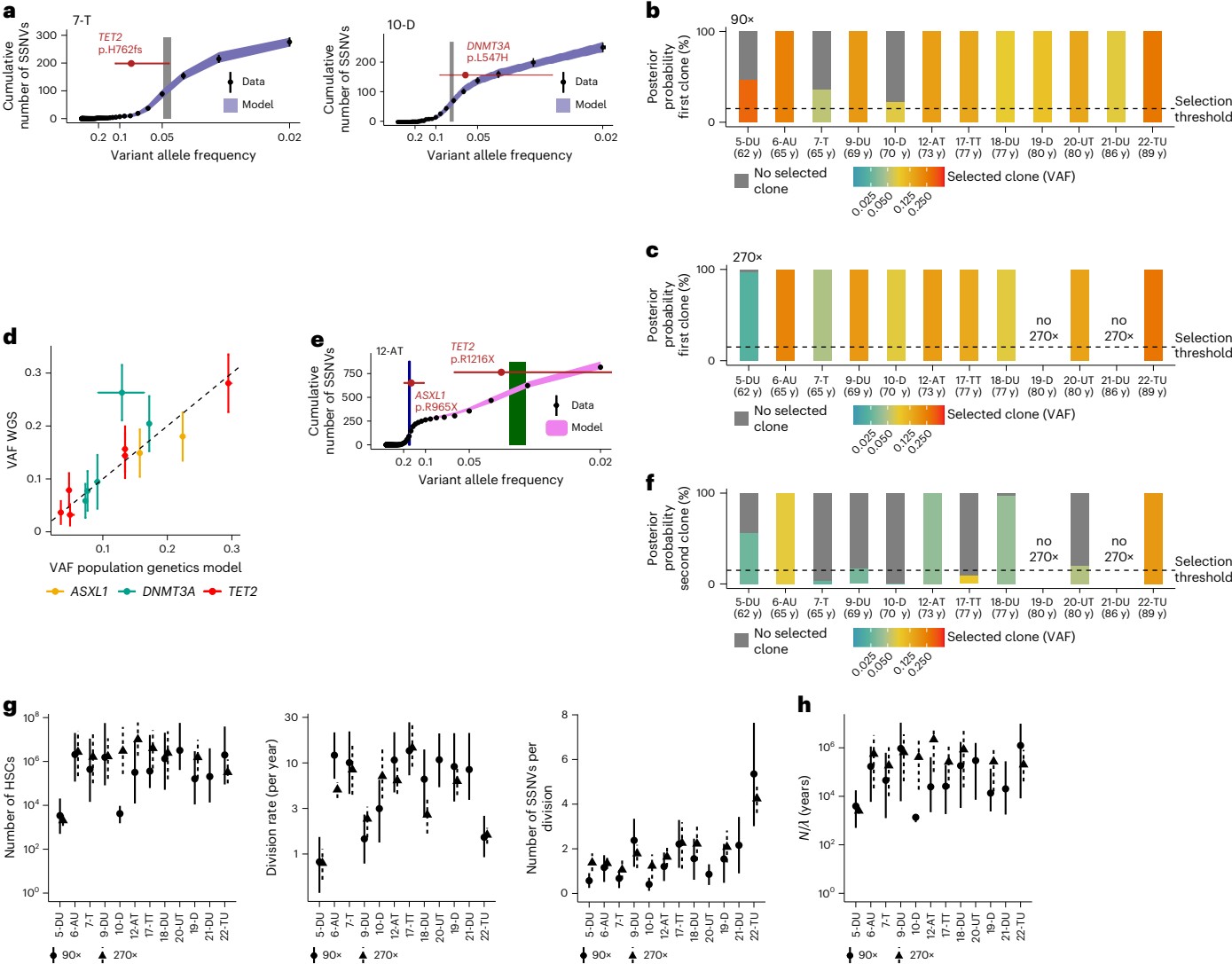

**Fig. 4 | Clonal selection for known CH drivers. a**, Model fit to the cumulative VAF distribution measured in CD34⁺ HSPCs of samples 7-T and 10-D with 270× WGS (points and error bars, measured data and their standard deviation, which, assuming Poisson-distributed measurements, is the square root of the measured data; purple area, 95% posterior probabilities of the model fit, estimated from simulations using 100 posterior samples; gray area, 80% credible interval of the clone size, estimated from 1,000 posterior samples; red points and error bars, mean and 95% confidence interval (CI) of the VAF of known CH drivers, based on binomial distributions with sample size and success probability values of 267 and 0.08, and 205 and 0.06, corresponding to read coverage and measured VAF in 7-T and 10-D, respectively). **b**, Model support for clonal selection (conditioned on clones ≥5% VAF) and neutral evolution based on 90× WGS data in 12 cases with selection and with at least one CH driver mutation in *AXSL1*, *DNMT3A* and *TET2*. Dashed line, 15% selection threshold. **c**, As **b**, but based on 270× bulk WGS data, where available (model support for selection conditioned on clones ≥2% VAF).

**d**, Estimated sizes of the selected clones (median and 80% credible intervals, computed from 1,000 posterior samples, for 270× WGS, where available, and 90× WGS else) versus measured VAF of known CH driver (mean and 95% CI according to binomial distributions with sample size taken as read coverage and success probability taken as measured VAF; for the 13 mutations, read coverage and VAFs are as follows: 249 and 0.03, 246 and 0.04, 205 and 0.06, 194 and 0.08, 267 and 0.08, 127 and 0.09, 264 and 0.14, 242 and 0.15, 275 and 0.16, 272 and 0.18, 230 and 0.20, 270 and 0.26, and 260 and 0.28). **e**, As in **a**, but for sample 12-AT (selected clones in blue and green; 95% CIs of the VAFs of mutations in *ASXL1* and *TET2* based on binomial distributions with sample size and success probability of 242 and 0.15 and 246 and 0.04, respectively). **f**, As in **b**, but for a second selected clone. **g**, Estimated stem cell parameters for the cases shown in **c**, showing median and 80% credible intervals for each sample, based on 1,000 posterior samples. **h**, As in **g**, but showing the ratio between stem cell number and division rate. y, years.

individuals with neurological disorders (autism spectrum disorder, schizophrenia, Tourette syndrome; Extended Data Fig. 10a).

First, we applied SCIFER to two cases (LIBD82 and NC7-CX-OLI) in which brain tumor-initiating driver mutations had been identified, despite no histological indication of malignancy[17]. In LIBD82, trisomy 7 and monosomy 10, characteristic of IDH-wild type glioblastoma[40], were detected in the hippocampus, but not cortex. SCIFER detected a selected clone of a size concordant with the CNA allele frequency (Fig. 7a). In NC7-CX-OLI, the R140Q *IDH2* hotspot mutation associated with *IDH*-mutant glioma[46,47] was detected in cortical oligodendrocytes

(NeuN⁻/Sox10⁺) and striatal interneurons (NeuN⁺/CITP2⁻). In both samples, SCIFER inferred selected clones, consistent with the VAFs of the *IDH2* mutation (Fig. 7b). The striatal clone contained additional cancer-associated driver mutations (*NRAS* G12D and *DNMT3A* splice donor variants). Interestingly, SCIFER inferred that the striatal clone was both larger (~15% VAF) and younger (acquired at age 55 years) than the cortical clone (~7.5% VAF and acquired at age 35 years) and had a faster growth rate. Although SCIFER cannot distinguish whether all driver mutations co-occurred in the same cells, or whether different clones with indistinguishable sizes coexisted, the large clone

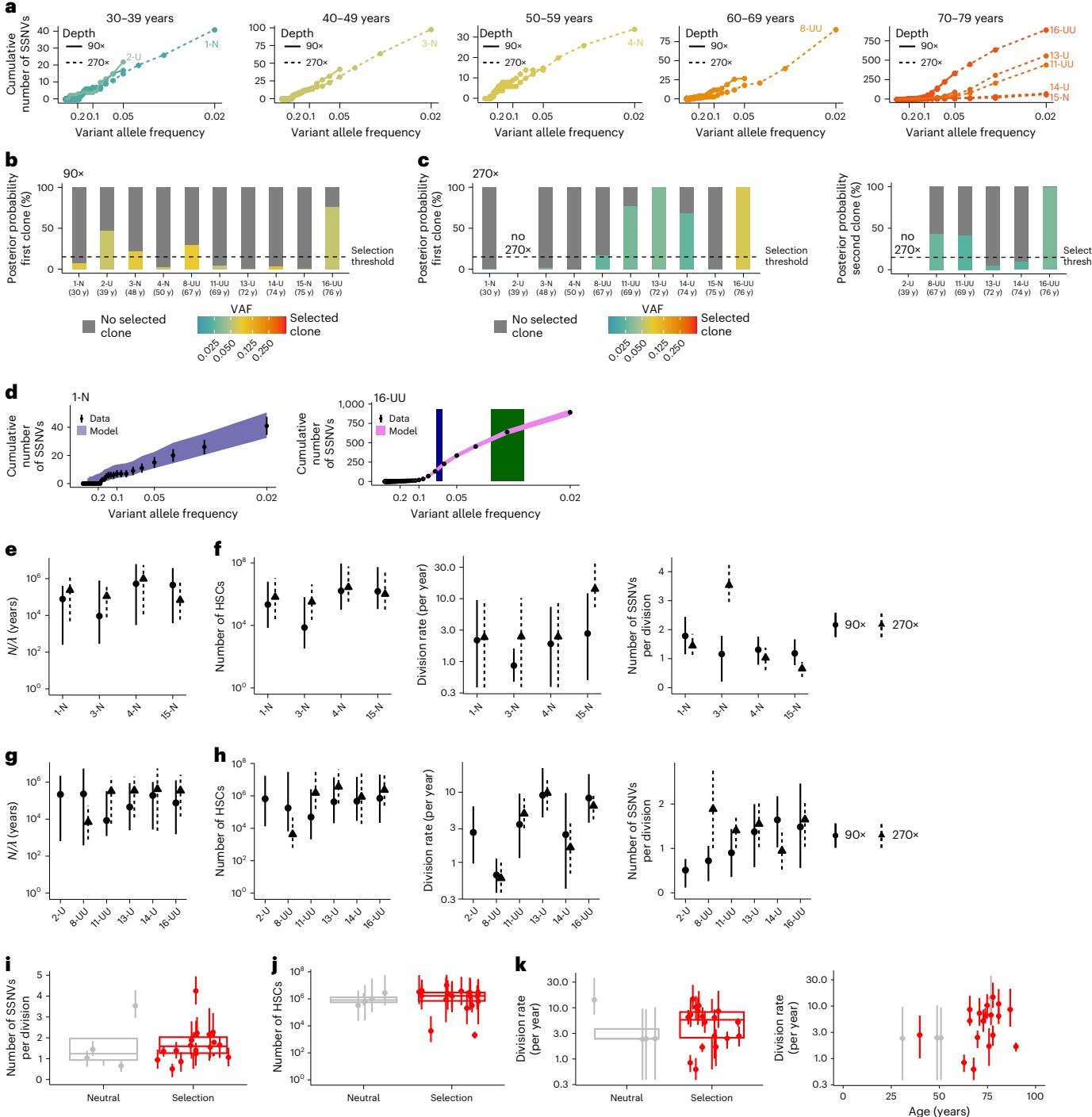

**Fig. 5 | Clonal selection for unknown CH drivers. a**, Cumulative $\frac{1}{VAF}$ distributions for samples without a known CH driver at 90× or 270× WGS coverage. **b**, Model support for clonal selection (conditioned on clones with VAF ≥ 5%) and neutral evolution across samples introduced in **a** at 90× WGS. Dashed line, 15% selection threshold. **c**, As in **b**, but for the leading (left) or second selected clone (right) at 270× WGS, where available (posterior probabilities conditioned on clones with VAF ≥ 2%). **d**, Model fit to the cumulative $\frac{1}{VAF}$ distribution for samples 1-N and 16-UU at 270× WGS (points and error bars, measured data and their standard deviation, which, assuming Poisson-distributed measurements, is the square root of the measured data; purple area, 95% posterior probabilities of the model fit computed from simulations using 100 posterior samples; blue and green areas, 80% credible intervals of the estimated sizes of the selected clones, computed from 1,000 posterior samples). **e,f**, Inferred ratio between HSC number and division rate (**e**) and stem cell parameters (**f**) for neutrally evolving

samples. Shown are median and 80% credible intervals for each sample, computed from 1,000 posterior samples; estimates obtained from 90× and 270× WGS are shown side by side. **g,h**, as in **e** and **f**, but for samples with unknown drivers. **i,j**, Estimated number of newly acquired SSNVs per HSC division (**i**) and number of HSCs contributing to hematopoiesis (**j**) in the 4 neutrally evolving cases compared with the 12 cases with selection for a known CH driver (introduced in Fig. 4) and the 6 cases with selection for an unknown driver (introduced in this figure; points and error bars, median and 80% credible intervals for each sample, estimated from 1,000 posterior samples for 270× WGS data, where available, and 90× WGS data else; boxplots, median and interquartile range, whiskers extend to the largest and smallest value no further than 1.5 times the interquartile range). **k**, Left, as in **j**, but showing the estimated HSC division rate. Right, estimated division rate versus age at sampling. Gray, neutral evolution; red, clonal selection.

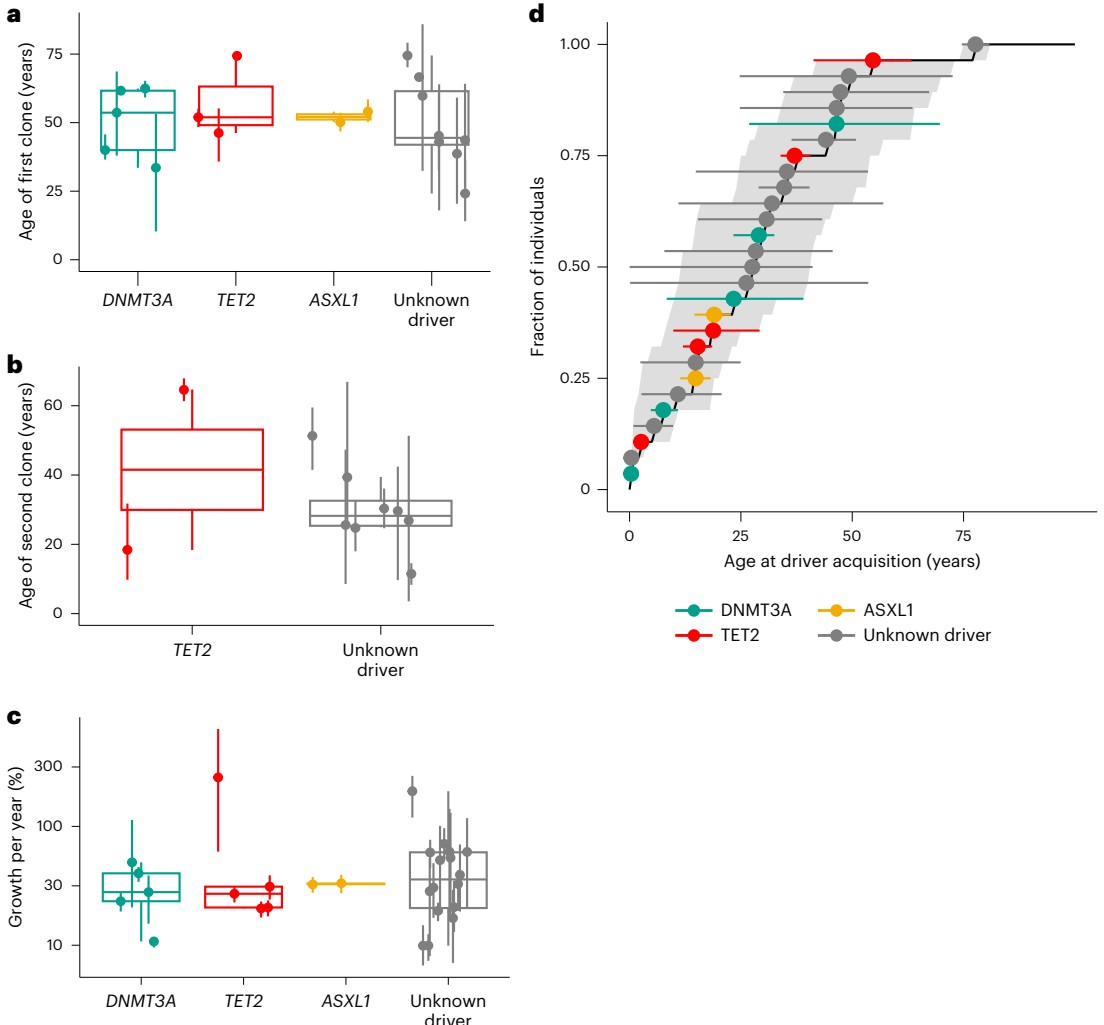

**Fig. 6 | Selection dynamics with and without known drivers. a,** Estimated age of the selected clone (for the 12 cases introduced in Fig. 4 and the 6 cases introduced in Fig. 5; points, median; error bars, 80% credible intervals estimated from 1,000 posterior samples; parameters were estimated with the two-clone model from 270× WGS data, where available, and from 90× WGS data else; boxplots, median and interquartile range, whiskers extend to the largest and smallest value no further than 1.5× the interquartile range). **b,** Age of the second selected clone estimated with the two-clone model (for the ten cases introduced in Figs. 4 and 5; points, median; error bars, 80% credible intervals, estimated from 1,000 posterior samples; boxplots, median and interquartile range, whiskers extend to the largest and smallest value no further than 1.5× the interquartile range). **c,** Estimated growth rates for the 28 selected clones introduced in **a** and

**b** (points, median; error bars, 80% credible intervals, estimated from 1,000 posterior samples; estimates were obtained with the two-clone model from 270× WGS data, where available, and from 90× WGS data else; boxplots, median and interquartile range, whiskers extend to the largest and smallest value no further than 1.5× the interquartile range). **d,** Cumulative distribution of estimated age at CH driver acquisition (median); shaded area, lower and upper bounds of the cumulative distribution of estimated age at CH driver acquisition based on 80% credible intervals of the estimated parameters; points and error bars, median and 80% credible intervals for the per-sample estimates, estimated from 1,000 posterior samples. Data are from the 28 selected clones introduced in **a** and **b** (estimates were obtained with the two-clone model from 270× WGS data, where available, and from 90× WGS data else).

size in striatal interneurons suggests that a subclone with additional driver mutations replaced the ancestral clone. In summary, SCIFER detected selected clones in brain samples associated with known driver mutations.

Across the cohort overall, SCIFER detected selected clones in 24% (44 of 185) of all samples (36 of 131 individuals; Fig. 7c,d). To independently validate these results, we compared the sizes of selected clones with the VAFs of CNAs reported in the original publication[17] (which are not used by SCIFER). Three individuals harbored duplications on chromosome 2 (AN09412; chr. 2: 96200001–102400000) or chromosome 3 (TS1, chr. 3: 113070000–113240000 and NC6, chr. 3: 60800001–61300000); their VAFs agreed with the selected clone detected by SCIFER, thus corroborating the SCIFER results. The presence of selected clones increased with age (Fig. 7e) and was not significantly associated with clinical phenotype, sex or sample location in the

brain (based on a generalized linear model taking into account age, location and phenotype; Extended Data Fig. 10b). SCIFER inferred ~10⁵ stem cells, in cortex, striatum or hippocampus, that divide on average two or three times per year (Fig. 7f and Extended Data Fig. 10c). Because human brain consists of hardly renewing neurons and glia cells that renew and have a shorter lifespan[48] we performed stochastic simulations to mirror this situation. These simulations suggest that VAFs measured in unsorted brain tissue report on the average renewal rate of these two subpopulations (Supplementary Note 1). Interestingly, the division rate of neural stem cells appears to decrease in the first 25 years of life, after which it remains constant (Fig. 7g). Between two divisions, they acquired typically fewer than two SSNVs (Extended Data Fig. 10c). SCIFER estimates that the vast majority of selected clones began to grow in the first 25 years of life (Fig. 7h). Consequently, elderly individuals had older clones (Extended Data Fig. 10d). Moreover, the

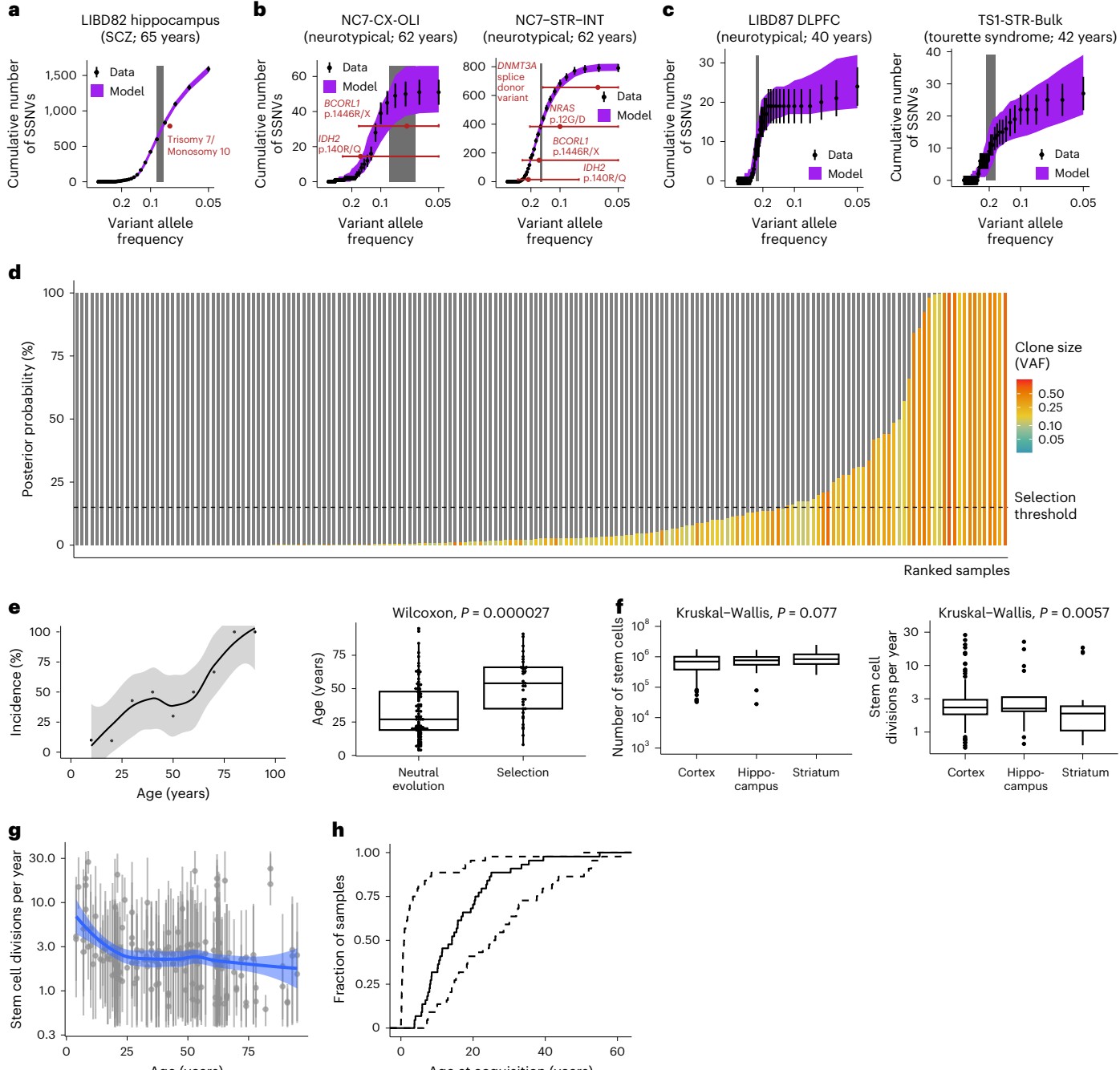

**Fig. 7 | Clonal selection in the human brain. a**, SCIFER fit to SSNVs measured in the hippocampus of LIBD82 (points and error bars, measured data and standard deviation, which, assuming Poisson-distributed measurements, is the square root of the measured data; purple area, 95% posterior probabilities of the model fit computed from simulations using 100 posterior samples; gray area, 80% credible interval of the clone size, estimated from 1,000 posterior samples; SCZ, schizophrenia). Red point, reported VAF of trisomy 7 and monosomy 10 (ref. 17). **b**, As in **a**, but for cortical oligodendrocytes and striatal interneurons of NC7 (red points, mean VAF; red lines, 95% CI based on binomial distributions with sample size and success probability of 26 and 0.15 and 29 and 0.07 (NC7-CX-OLI); 30 and 0.1, 33 and 0.06, 32 and 0.22, and 32 and 0.16 (NC7-STR-INT), corresponding to read coverage and measured VAF, respectively)[17]. **c**, As in **a**, but for LIBD87 (cortex) and TS1 (striatum). **d**, Model support for clonal selection (conditioned on clones with ≥5% VAF for <150× WGS and ≥2% VAF for ≥150×) and neutral evolution across 185 brain samples from 131 individuals. Dashed line, 15% selection threshold. **e**, Left, average incidence of clonal selection versus age (summarizing ages into 10 year-bins; in total, 36 of 131 individuals with selection; line and shaded area, LOESS regression with 95% CI). Right, age of individuals with (n = 36) and without selection (n = 95; boxplots, median and interquartile range, whiskers, largest and smallest value no further than 1.5× the interquartile range; P = 0.00002662, Wilcoxon test statistic W = 2,525, two-sided Wilcoxon rank sum test). **f**, Median posterior stem cell parameter values by location (cortex, n = 128; hippocampus, n = 17; striatum, n = 40; boxplots, median and interquartile range, whiskers, largest and smallest value no further than 1.5× the interquartile range; points, outliers; P values, one-way analysis of variance with Kruskal–Wallis test). **g**, Estimated stem cell division rate versus age for 185 samples (median and 80% credible intervals computed from 1,000 posterior samples each; blue line and shaded area, LOESS regression with 95% CI). **h**, Cumulative distribution of estimated age at driver acquisition (median values of 44 samples with selection); lower and upper bounds based on 80% credible intervals (based on 1,000 posterior samples for each sample).

clonal growth rate decreased with age (Extended Data Fig. 10d). The focused age incidence of clonal selection in the brain contrasts with the constant rate in hematopoiesis.

## Discussion

Here, we present a population genetics approach to detect clonal selection and quantify tissue stem cell dynamics from snapshot WGS data. Akin to recent approaches to cancer evolution[37,38,40,49–51], our theory delineates how neutral evolution versus selection shape the somatic VAF spectra. Compared with previous work in tumors[38,49], SCIFER explicitly treats genetic drift and its interplay with clonal selection, which is key for nonmalignant, homeostatic tissues. SCIFER identifies selection without knowledge of the underlying driver, detects whether more than one selected clone is present, and quantifies their age and selective advantage(s).

At 270×, SCIFER has a sensitivity of detecting clones at 2% VAF in a single bulk sample at a fraction of the cost and effort of single-cell colony-derived phylogenetic reconstruction. Recently, analysis of bulk WGS of blood samples aimed to identify CH, assuming selection increases total detected somatic variants[52]. Here, we found that this is not necessarily the case and hence would like to caution against using large SSNV count in a WGS sample as a sign of selection.

Agreement of HSC parameters inferred from one phylogenetic tree with neutral evolution[2] and SCIFER from the 22 BM samples studied here is remarkable, because SCIFER strongly relies on variants with higher VAF emerging relatively early, whereas the phylodynamic inference uses low-VAF SSNVs generated later. This may indicate that HSC dynamics are rather uniform across time. Nevertheless, it is important to note that our inferred parameters are time-averaged. Because analysis of driver VAF in serial blood samples suggests that expansion rates may change over time[12], it would be valuable to apply SCIFER to serial samples to quantify stem cell dynamics over time.

Our parameter inference suggests variation in HSC division rate between individuals. Indeed, mean telomere length has been shown to decrease faster in individuals with CH[53], and longer telomere length is a causative risk factor for CH[54,55], suggesting that long telomeres allow CH HSCs to undergo enhanced cell division. It should be kept in mind that our parameter estimates rely on the assumption of a constant mutation rate in HSCs along the human lifespan. During adult life, HSC mutation rate appears constant[56]; whether this is true in fetal development is unclear.

Consistent with previous studies[12–14], we identified pervasive clonal selection in the absence of known drivers. However, in three cases selected clones were at the limit of detection, and were not found in the 90× data. Such clones just below the detection limit may affect the measured VAF distribution at low frequencies, and bias SCIFER to show an erroneously increased drift effect. This potential problem can be addressed by deeper sequencing, and potentially extending SCIFER to more than two clones (for which the online code provides an option).

Although the statistical power of the current study is limited, we observed several structural variants (SVs) previously described as recurrent somatic mutations[57,58], associated with increased risk for hematological cancer[21,58] (Supplementary Tables 6 and 7). Moreover, in-depth analyses of the genome-wide SSNV profiles revealed potential candidates for driver mutations (Supplementary Table 5), one of which had significantly higher ratios of nonsynonymous-to-synonymous SSNVs in ~200,000 UK Biobank blood samples[25]. By applying SCIFER to large cohorts it may be possible to more accurately determine the frequency of individuals with unknown driver events, and their clinical phenotype.

Applying SCIFER to brain tissue, we found clonal selection in ~25% of the samples. The prevalence of unknown drivers was even more pronounced than in hematopoiesis. Moreover, in two individuals, we found pre-malignant selected clones with characteristic brain tumor-initiating driver mutations without histological evidence of malignancy. This raises the question of how frequent pre-malignancy

in the human brain is and whether the situation is similar to blood[19,20]. Selected clones in the brain were largely born in the first 25 years of life, coinciding with the period of extensive postnatal brain development[59]. Brain cell-type heterogeneity (neurons and glia cells) precludes a more accurate quantitation of the stem cell dynamics, which will require analysis of purified cell populations. Nevertheless, stochastic simulations suggest that VAFs in tissues composed of heterogeneous cell types reflect on average cell turnover (Supplementary Note 1). We envisage that SCIFER will be useful to study clonal selection in other solid organs in humans.

## Online content

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

## Methods

### Ethical approval

Patient samples were collected with informed consent under the Mechanisms of Age-Related Clonal Haematopoiesis (MARCH) Study. Written informed consent was obtained from all participants in accordance with the Declaration of Helsinki. This study was approved by the Yorkshire & The Humber – Bradford Leeds Research Ethics Committee (REC Ref17:/YH/0382).

### Study samples

Participants were recruited from individuals undergoing elective total hip replacement surgery at the Nuffield Orthopaedic Centre, Oxford. Exclusion criteria were history of rheumatoid arthritis or other inflammatory arthritis, history of septic arthritis in the limb undergoing surgery, history of hematological cancer, bisphosphonate use and oral steroid use. Patient characteristics are summarized in Supplementary Table 2. At the time of surgery, trabecular bone fragments and BM aspirates were obtained from the femoral canal and collected in anticoagulated buffer containing acid–citrate–dextrose, heparin sodium and DNase. Samples of peripheral blood were collected in EDTA vacutainers. Hair follicle samples were collected from participants as a germline control. Peripheral blood and BM mononuclear cells (MNCs) were isolated by Ficoll density gradient centrifugation and viably frozen. Peripheral blood granulocytic cell pellets were frozen for later DNA extraction. Genomic DNA was extracted from BM MNCs, peripheral blood granulocytes and hair follicles using a DNeasy Blood & Tissue Kit (Qiagen).

### Cell sorting

Thawing media was prepared with IMDM medium (Gibco) supplemented with 20% fetal bovine serum (FBS) and 110 µg ml$^{-1}$ DNase. BM samples were thawed at 37 °C in a water bath, 1 ml of warm FBS was added and the suspension was then diluted by dropwise addition of 8 ml of thawing media. The suspension was centrifuged at 400 g for 10 min, cells were resuspended in flow cytometry staining medium (IMDM with 10% FBS and 10 µg ml$^{-1}$ DNase), filtered through a 35-µm cell strainer and placed on ice.

Cells were stained with the following antibodies: anti-CD34-PE (1:160, BioLegend, clone 581), anti-CD3-PE/Cy7 (1:100, BioLegend, clone HIT3a), anti-CD2-PE/Cy5 (1:160, BioLegend, clone RPA-2.10), anti-CD4-PE/Cy5 (1:160, BioLegend, clone RPA-T4), anti-CD7-PE/Cy5 (1:160, BioLegend, clone CD7-6B7), anti-CD8a-PE/Cy5 (1:320, BioLegend, clone RPA-T8), anti-CD11b-PE/Cy5 (1:160, BioLegend, clone ICRF44), anti-CD14-PE/Cy5 (1:160, eBioscience, clone 61D3), anti-CD19-PE/Cy5 (1:160, BioLegend, clone HIB19), anti-CD20-PE/Cy5 (1:160, BioLegend, clone 2H7), anti-CD56-PE/Cy5 (1:80, BioLegend, clone MEM188) and anti-CD235ab-PE/Cy5 (1:320, BioLegend, clone HIR2). Following antibody incubations, cells were washed with 1 ml of flow cytometry staining buffer, centrifuged at 350 g for 5 min and resuspended in flow cytometry staining buffer containing 1:10,000 Hoechst 33342 live–dead stain.

Cell sorting was performed on a BD FACSAria Fusion or Sony MA900 equipped with a 100-µm nozzle or sorting chip. Unstained, single-stained and fluorescence-minus-one controls were used to determine background staining and compensation in each channel. Doublets and dead cells were excluded. The following populations were sorted with a mean purity >95%: Lin$^-$CD34$^+$ HSPCs, Lin$^+$CD3$^+$ T cells and Lin$^{+/-}$CD34$^-$CD3$^-$ MNCs. Genomic DNA from sorted cell populations was extracted using a QIAamp DNA Micro Kit (Qiagen).

### Whole-genome sequencing

#### Library preparation and sequencing.

Sequencing libraries were prepared using the Illumina DNA Prep Kit (cat. no. 20018704) and Nextera DNA CD Indexes (cat. no. 20018707) according to manufacturer's guidelines. The Qubit HS DNA Assay Kit (Invitrogen, cat. no. Q32854)

and a tape station run with the Agilent HS D5000 Assay Kit (cat. no. 5067-5593) were used for quality control. Thereafter, libraries were diluted to 10 nM, pooled and sequenced on either HiSeq X PE150 or on NovaSeq 6000 PE150.

Low-quality bases on raw sequencing reads were trimmed using Trim Galore v.0.5.0 (https://www.bioinformatics.babraham.ac.uk/projects/trim_galore/) with cutadapt v.2.8 (ref. [60]) with the following settings: --quality 30, --illumina --length 32 --trim-n --clir_R1 2 --clip_R2 2 --three_prime_clip_R1 2 --three_prime_clip_R2 2. Trimmed read pairs were mapped to the human reference genome (build 37 with the GRCh37.75 genome annotation) using bwa mem v.0.7.12 (ref. [61]). Mapped reads were coordinate-sorted with samtools sort v.1.5 (ref. [61]), followed by marking duplicate read pairs with gatk MarkDuplicates v.4.0.9.0 (ref. [62]) and indexing with samtools index.

**Detection of SSNVs and insertions–deletions.** Somatic SNVs and small insertions–deletions (indels) in BM and blood samples were called with Strelka v.2.9.2 (ref. [63]) and Mutect2, GATK v.4.2.0.0 (ref. [62]), using the matched hair follicle samples as the germline control. Variants in repeat regions and simple repeat regions (downloaded from UCSC table browser, setting the assembly to hg19, the track to 'RepeatMasker' or 'Simple Repeats'; accession date: 5 December 2018) were filtered using bedtools intersect v.2.24.0 (ref. [64]). Only variants that passed the default filters of both Strelka and Mutect2 were retained. To identify remaining germline variants, variants were looked up in dbSNP (v.150) and the population frequency of reported variants was annotated based on the Genome Aggregation Database (gnomAD v.2.1.1; https://gnomad.broadinstitute.org/). Variants were then filtered to exclude potential contamination of germline variants based on the global allele frequency (AF) of gnomAD (retaining variants with AF < 0.001). In a few cases, a CH driver was identified by panel sequencing at low VAF, but not recovered by Strelka and Mutect; here, we re-examined the respective position using bcftools mpileup v.1.10.2 (ref. [65]). All variants were annotated with ANNOVAR (v.May2018; http://annovar.openbioinformatics.org/)[66] according to the human genome build v.19. For SSNVs, VAFs were recalculated directly from the BAM files using alleleCounter v.4.0.2 (https://github.com/cancerit/alleleCount; with default base and mapping quality thresholds) and MaC (https://github.com/nansari-pour/MaC).

**Detection of copy number variants.** Genome-wide subclonal CNAs were identified with Battenberg (v.2.2.10; https://github.com/Wedge-lab/battenberg), which has been described in detail previously[67].

**Detection of structural variants.** SVs from whole-genome data were identified using Manta v.1.6.0 (ref. [68]) with default tumor–normal pair settings comparing samples with the germline controls. We kept variants that passed Manta's default filter, had a minimal SOMATICSCORE of 30, were not classified as IMPRECISE, had at least three variants and a VAF of 5% in the blood sample, and at most two variant reads and VAF < 5% in the control sample. SVs previously described in somatic tissues[57,58] and associated with hematological cancer[21,58] were classified as putative drivers (Supplementary Table 7).

**Driver analysis.** To comprehensively search for candidate driver mutations, we concatenated published lists of putative driver genes in CH and leukemia from Intogen[69] (subsetting genes described in 'ALL', 'AML', 'CLL', 'CML' or 'MM' from release 2020.02.01, and the CH gene list from ref. [70]), Cosmic[71] v.94 (subsetting on genes with annotation in leukemia), ref. [13], and a curated list of CH-associated mutations compiled from five large studies[19,20,27,30,72,73].

Mutations identified by Strelka and targeting any of these genes were looked up in ClinVar[74] (v.20221231) and annotated accordingly. Using the R package drawProteins[75] v.1.18.0 we collected information

on the protein domain targeted by each variant. Moreover, we manually annotated variant information (involvement in disease, functional evidence for mutations at this site, SNP identifier (SNPID), association with genetic disorders and whether the variant targets a functional domain) using the manually curated variant information 'homo_sapiens_variation.txt' from Uniprot[76] (downloaded 4 April 2023) and variant information provided on www.uniprot.org. Thereafter, we computed SIFT[77] and Revel[78] prediction scores for each substitution. We kept variants causing a 'stopgain', 'stoploss', 'frameshift_insertion' or 'frameshift_deletion', or with 'Conflicting interpretations of pathogenicity', 'Likely pathogenic', 'Pathogenic/Likely pathogenic' or of 'uncertain significance' according to ClinVar, or annotated to a related disease (based on uniprot.org) or annotated as pathogenic (uniprot.org) or with a REVEL score ≥0.75 or targeting either *ASXL1*, or *DNMT3A* or *TET2*.

This yielded a set of variant positions, which we merged with curated positions in known CH drivers[27]. Based on this set, we re-called variants in all blood and control samples with bcftools mpileup[65] v.1.10.2, using the option -p, and bcftools call, using the option -mA. Variant calls were performed in batches and merged using bcftools merge. In the final set, we kept variants targeting any of *ASXL1*, *DNMT3A* or *TET2* and variants that were found on at least three reads in the sample and on fewer than five reads in the controls.

### Analysis of single-cell WGS data

We tested our population genetics model on published single-cell WGS data from refs. 2,12,13. To this end, we computed pseudo-bulk VAFs from the single-cell phylogenies as

$$\text{VAF} = \frac{n_{\text{variant cells}}}{2n_{\text{cells}}}$$

where $n_{\text{variant cells}}$ is the number of cells harboring a variant and $n_{\text{cells}}$ is the total number of sequenced cells; the factor 2 accounts for diploidy.

**Parameter estimation.** We fit our population genetics model to the cumulative VAF distribution truncated at 0.01, as detailed below and using the prior probabilities outlined in Supplementary Table 7.

**Reanalysis of SSNV and indels.** To assess differences in variant calling pipelines, we re-called SSNVs and indels in the data from ref. 2. To this end, we intersected the results between Mutect2 (ref. 62) and Strelka2 (ref. 63), filtered remaining germline variants using gnomAD (retaining variants with AF <0.001) and recalculated VAFs directly from the BAM files using alleleCounter v.4.0.2 (https://github.com/cancerit/allele-Count; with default base and mapping quality thresholds) and MaC (https://github.com/nansari-pour/MaC). We filtered the remaining variants as stated in ref. 2. In brief, we removed variants occurring in >120 of the 140 colonies, variants that fell within 10 bp of each other and variants with a coverage <6 on autosomes or <3 on sex chromosomes in more than five samples. Moreover, we retained only variants with a mean VAF > 0.3 across all samples with at least one mutant read. Finally, we excluded sites at which >10% of the samples with at least one mutant read had a VAF < 0.1.

**Phylogenetic reconstruction.** We reconstructed single-cell phylogenies based on the re-called SSNVs and indels from ref. 2 by converting the mutation table into a fasta file, learning the phylogenetic tree using MPBoot v.1.1.0 (ref. 79) and plotting the tree using custom scripts from ref. 13 (https://github.com/emily-mitchell/normal_haematopoiesis/).

### Analysis of brain samples

SSNVs of 457 human brain samples were downloaded from ref. 17. Of these, we analyzed 177 samples with an average coverage >100×, as well as 8 samples from individual NC7 (NC7-CX-ASTMIG, NC7-CX-INT, NC7-CX-OLI, NC7-CX-PYR, NC7-STR-ASTMIG, NC7-STR-INT, NC7-STR-

MSN and NC7-STR-OLI) that contained multiple known driver mutations but had average coverage between 32× and 40× (Supplementary Table 8). Both tier1 and tier2 variant calls were used for analysis.

### Population genetics model

**Theory.** We modeled the evolution of VAFs mechanistically, accounting for accumulation, drift and selection of somatic variants in a homeostatic tissue. The model is parametrized with the rate at which HSCs divide during adulthood, $\lambda$, the number of SSNVs acquired between two cell divisions, $\mu$, the number of HSCs during adulthood, $N_{\text{ss}}$, as well as the time of origin of the selected clone, $t_{\text{s}}$, and its selective advantage expressed by $r$. We bundled the model functions in an R package, SCIFER, available on https://github.com/VerenaK90/SCIFER.

*Modeling the site frequency spectrum of somatic variants generated by neutral evolution.* The time-dependent site frequency spectrum $S_i(t)$ gives the number of variants with clone size $i$, where $i$ ranges from some minimally observable clone size (owing to WGS sequencing depth) up to the total number of stem cells $N(t)$. We derive analytical expressions for the site frequency spectra (SFS) resulting from neutral evolution and clonal selection and compare these with measured VAF histograms.

To begin with neutral evolution, we develop a stochastic model for accumulation and drift of neutral somatic variants during developmental expansion and subsequent homeostasis of the stem cell pool. The model assumes that stem cells proliferate via symmetric self-renewing divisions, with rate $\lambda$, and are lost by differentiation and cell death, with rate $\delta$. Between two subsequent stem cell divisions, on average $\mu$ new variants are introduced. These variants are inherited to daughter cells and, depending on the dynamics of the corresponding stem cell clone, may either go extinct or drift to variable frequencies. The SFS generated by these dynamics is:

$$S_i(t) = \int_0^t \lambda \mu N(t') P_{1,i}(t, t') \mathrm{d}t'$$

describing the generation of new variants at time $t'$, in a population of size $N(t')$, and their drift to clone size $i$ up to the time of measurement $t$ with probability $P_{1,i}(t, t')$. To compare $S_i(t)$ with measured VAF histograms, we transform from clone size to VAF:

$$\text{VAF} = \frac{i}{2N}$$

To describe developmental expansion of the stem cell pool followed by homeostasis, we concatenated linear birth–death processes. During development, the division rate will exceed the loss rate, $\lambda_{\text{exp}} > \delta_{\text{exp}}$, hence defining a supercritical birth–death process. At time $t_1$, the system reaches its steady-state with a constant number of active stem cells, $N_{\text{ss}}$. The cellular dynamics are now appropriately described by a critical birth–death process with steady-state rate $\lambda_{\text{ss}} = \delta_{\text{ss}}$.

To compute the SFS during developmental expansion, we consider the probability that a cell acquiring a new variant will expand to a clone of size $a$ in time $t$ (ref. 80):

$$P_{\text{exp},1,a}(t) = \begin{cases} x(t), & \text{if } a = 0 \\ (1 - x(t))(1 - y(t)) y(t)^{a-1}, & \text{if } a \geq 1 \end{cases} \quad (1)$$

with

$$x(t) = \frac{\delta_{\text{exp}} e^{(\lambda_{\text{exp}} - \delta_{\text{exp}})t} - \delta_{\text{exp}}}{\lambda_{\text{exp}} e^{(\lambda_{\text{exp}} - \delta_{\text{exp}})t} - \delta_{\text{exp}}}, \quad y(t) = \frac{\lambda_{\text{exp}} e^{(\lambda_{\text{exp}} - \delta_{\text{exp}})t} - \lambda_{\text{exp}}}{\lambda_{\text{exp}} e^{(\lambda_{\text{exp}} - \delta_{\text{exp}})t} - \delta_{\text{exp}}} \quad (2)$$

The measured VAF histograms report on the number of variants with a given frequency. To calculate variant number in the model,

we note that between time $t'$ and $t' + dt'$ on average $\mu\lambda_{\exp}N(t')dt'$ variants are generated. Hence,

$$\mu\lambda_{\exp}N(t')\,dt' \times P_{\exp,1,a}(t - t') \tag{3}$$

variants introduced at time $t'$ each occur in $a$ cells at time $t$.

During tissue homeostasis, when stem cell division and loss will occur both at steady-state rate $\lambda_{ss}$, drift is described by the critical birth–death process. A variant in $a$ cells at $t_1$ (when the homeostatic stem cell number is reached) will drift to occur in $b$ cells within time $t$ with probability[80]

$$P_{ss,a,b}(t) = \begin{cases} p(t)^a, & \text{if } b = 0 \\ \sum_{j=0}^{b} \frac{j}{b}\binom{a}{j}p(t)^{a-j}(1-p(t))^j\binom{b}{j}p(t)^{b-j}(1-p(t))^j, & \text{otherwise} \end{cases} \tag{4}$$

with

$$p(t) = \frac{\lambda_{ss}t}{1 + \lambda_{ss}t} \tag{5}$$

Thus, at $t \geq t_1$, the number of variants occurring exactly in $b$ cells is

$$\sum_{a=1}^{N_{ss}} \mu\lambda_{\exp}N(t')P_{\exp,1,a}(t_1 - t')P_{ss,a,b}(t - t_1)\,dt' \tag{6}$$

for variants generated during developmental expansion ($t' < t_1$).

Finally, we consider variants acquired during homeostasis, which evolve according to the critical birth–death process entirely. Hence, the number of such variants occurring exactly in $b$ cells is

$$\mu\lambda_{ss}N_{ss} \times P_{ss,1,b}(t - t')\,dt' \tag{7}$$

where $t' \geq t_1$. Combining the contribution of both phases, we arrive at the SFS of neutral variants in a homeostatic tissue without selection:

$$S_i(t) = \underbrace{\int_0^{t_1} \sum_{a=1}^{N_{ss}} \mu\lambda_{\exp}N(t')P_{\exp,1,a}(t_1 - t')P_{ss,a,i}(t - t_1)\,dt'}_{\text{variants generated in development}} \\ + \underbrace{\int_{t_1}^{t} \mu\lambda_{ss}N_{ss}P_{ss,1,i}(t - t')\,dt'}_{\text{variants generated in homeostasis}} \tag{8}$$

Equation (8) generalizes a result by Ohtsuki and Innan[81] for expanding tissues.

*Modeling the site frequency spectrum of somatic variants under selection.* **Modeling a single selected clone in a homeostatic tissue.** Clonal selection will modify the SFS. Consider that a positively selected mutation is acquired at time $t_s$ and reduces the rate of stem cell loss by the factor $r < 1$ (the alternative for imparting selective advantage, an increase in the rate of stem cell division, yields very similar results). The cell number of mutated stem cells, $n_2(t)$, expands at the expense of the number of normal stem cells, $n_1(t) = N_{ss} - n_2(t)$, according to the competition model

$$\frac{dn_1}{dt} = \lambda_{ss}n_1(t)\left(1 - \rho_{n_1,n_1}\frac{n_1(t)}{N_{ss}} - \rho_{n_1,n_2}\frac{n_2(t)}{N_{ss}}\right) = g(n_1, n_2),$$
$$\frac{dn_2}{dt} = \lambda_{ss}n_2(t)\left(1 - \rho_{n_2,n_2}\frac{n_2(t)}{N_{ss}} - \rho_{n_2,n_1}\frac{n_1(t)}{N_{ss}}\right) = h(n_1, n_2) \tag{9}$$

The $\rho$ parameters denote phenomenological competition coefficients between and in the mutant clone and the normal stem cells, maintaining homeostasis. We have no further growth if either normal or mutated stem cells fill the entire compartment, $g(N_{ss}, 0) = 0 = h(0, N_{ss})$ and $g(0, N_{ss}) = 0 = h(N_{ss}, 0)$, implying that $\rho_{n_1,n_1} = 1 = \rho_{n_2,n_2}$. Moreover, $\rho_{n_2,n_1} = r$ and hence

$$\frac{dn_1}{dt} = \lambda_{ss}n_1(t)\left(1 - \frac{n_1(t)}{N_{ss}} - (2 - r)\frac{n_2(t)}{N_{ss}}\right) \tag{10}$$

$$\frac{dn_2}{dt} = \lambda_{ss}n_2(t)\left(1 - \frac{n_2(t)}{N_{ss}} - r\frac{n_1(t)}{N_{ss}}\right) \tag{11}$$

The number of mutant stem cells $n_2(t)$ will remain much smaller than the number of normal stem cells for extended periods, and hence we approximate equation (11) by

$$\frac{dn_2}{dt} = \lambda_{ss}n_2(t)(1 - r) \tag{12}$$

Equations (10) and (12) will be used when computing the SFS.

With clonal selection, the SFS has three principal contributions for somatic variants originating: (1) before the positively selected mutation occurred, $S_{i,1}$; (2) after this mutation occurred and happening in the mutant clone, $S_{i,2}$; and (3) after the mutation occurred but happening in normal stem cells, $S_{i,3}$:

$$S_i(t) = S_{i,1}(t) + S_{i,2}(t) + S_{i,3}(t) \tag{13}$$

We specify each contribution in turn. The SFS in the mutant clone, $S_{i,2}$, is

$$S_{i,2}(t) = \int_{t_s}^{t} \mu\lambda_{ss}n_2(t')P_{\exp,1,i}(t - t'|\lambda_{ss}, r\lambda_{ss})\,dt' \tag{14}$$

where $P_{\exp,1,i}$ is given by equation (1).

The SFS in normal stem cells, $S_{i,1}$ and $S_{i,3}$, are shaped by the decline in the number of normal stem cells after $t_s$ (equation (10)). We approximate this process with a subcritical birth–death process with division rate $\lambda_{ss}$ and effective loss rate $\delta_{\text{eff}}(t)$. The loss rate is chosen such that the expectation of the decline of normal stem cells in the linear birth–death process equals the number of normal stem cells lost by the competition dynamics, $D$. Equation (10) implies that

$$D = \int_{t_s}^{t} \lambda_{ss}\left(\frac{n_1(t)}{N_{ss}} + (2 - r)\frac{n_2(t)}{N_{ss}}\right)n_1(t)\,dt \tag{15}$$

Therefore, the effective loss rate $\delta_{\text{eff}}(t)$ is defined by

$$\int_{t_s}^{t} \delta_{\text{eff}}N_{ss}\exp\left((\lambda_{ss} - \delta_{\text{eff}})(t - t_s)\right)dt = D \tag{16}$$

The SFS of variants acquired in normal stem cells after $t_s$, $S_{i,3}$, is the superposition of $(N_{ss} - 1)$ independent linear subcritical birth–death processes and is given by

$$S_{i,3}(t) = (N_{ss} - 1)\int_{t_s}^{t} \mu\lambda_{ss}e^{(\lambda_{ss} - \delta_{\text{eff}})(t' - t_s)}P_{\exp,1,i}(t - t'|\lambda_{ss}, \delta_{\text{eff}})\,dt' \tag{17}$$

Finally, we compute the SFS of variants acquired before $t_s$, $S_{i,1}$. These variants may be inherited to the selected clone, in which case they will be present in all selected cells and, additionally, in some of the normal cells. Alternatively, they may be present in normal cells only. To distinguish the two cases, we consider a variant that was acquired before $t_s$ and is present in $k$ cells at $t_s$. Assuming that the driver mutation is acquired in a random cell, the probability of this variant being inherited to the selected clone is $\frac{k}{N_{ss}}$, while the probability of this variant being exclusively present in normal cells is $\frac{1-k}{N_{ss}}$. In the former case, all $n_2(t)$

selected cells will harbor the variant at the time of measurement, $t$; in addition, the $k-1$ normal stem cells harboring the variant at time $t_s$ may reach a clone size between 0 and $N_{ss} - n_2(t)$, according to the drift dynamics of normal stem cells. In the alternative case, the variant does not end up in the selected clone and hence solely drifts in the normal cells. Taken together, this yields

$$S_{i,1}(t) = \begin{cases} \sum_{k=1}^{N_{ss}} S_k(t_s) \left[ \overbrace{\frac{k}{N_{ss}} P_{exp,k-1,i-n_2(t)} (t - t_s | \lambda_{ss}, \delta_{eff})}^{\text{variants drift in normal cells and present in all selected cells}} \right. \\ \left. + \overbrace{\left(1 - \frac{k}{N_{ss}}\right) P_{exp,k,i} (t - t_s | \lambda_{ss}, \delta_{eff})}^{\text{variants not present in selected cells}} \right], i \geq n_2(t), \\ \sum_{k=1}^{N_{ss}} \left(1 - \frac{k}{N_{ss}}\right) S_k(t_s) P_{exp,k,i} (t - t_s | \lambda_{ss}, \delta_{eff}), i < n_2(t), \end{cases}$$

(18)

where $S_k(t_s)$ is the SFS at $t_s$ and $P_{exp,a,b}$ is the clone size distribution generated by a subcritical birth–death process initiated by $a$ cells[80]:

$$P_{exp,a,b}(t) = \begin{cases} x(t)^a, \text{if } b = 0, \\ \sum_{j=0}^{\min(a,b)} \binom{a}{j} \binom{a+b-j-1}{a-1} x(t)^{a-j} y(t)^{b-j} (1 - x(t) - y(t))^j, \text{if } b \geq 1 \end{cases}$$

(19)

Thus, we have specified the contributions to the SFS of a stem cell population containing a selected clone, $S_i(t) = S_{i,1}(t) + S_{i,2}(t) + S_{i,3}(t)$.

**Modeling a single selected clone without size compensation.** We here develop a version of our model in which the selected clone expands unrestrictedly. We assume a constant number $N_{ss}$ of normal stem cells, whereas the selected clone exponentially expands. Hence, the population dynamics of normal ($n_1$) and mutant ($n_2$) stem cells now read

$$n_1(t) = N_{ss},$$
$$n_2(t) = e^{\lambda_{ss}(1-r)(t-t_s)}$$

(20)

As before, the SFS comprises variants that originated (1) before the positively selected mutation occurred, $S_{i,1}$; (2) after this mutation occurred and happening in the mutant clone, $S_{i,2}$; and (iii) after the mutation occurred but happening in normal stem cells, $S_{i,3}$. $S_{i,2}$ is the same as in the competition model and is modeled by equation (14). $S_{i,3}$ is now the superposition of $N_{ss}$ independent linear critical birth–death processes and reads

$$S_{i,3}(t) = N_{ss} \int_{t_s}^{t} \mu \lambda_{ss} P_{ss,1,i}(t - t' | \lambda_{ss}) \, dt'$$

(21)

Likewise, $S_{i,1}$ is now governed by genetic drift according to a critical birth–death process in the founder cell population, in addition to selection of the mutant clone. It reads

$$S_{i,1}(t) = $$
$$\begin{cases} \sum_{k=1}^{N_{ss}} S_k(t_s) \left[ \frac{k}{N_{ss}} P_{ss,k-1,i-C(t)}(t - t_s | \lambda_{ss}) \right. \\ \left. + \left(1 - \frac{k}{N_{ss}}\right) P_{ss,k,i}(t - t_s | \lambda_{ss}) \right], i \geq n_2(t), \\ \sum_{k=1}^{N_{ss}} \left(1 - \frac{k}{N_{ss}}\right) S_k(t_s) P_{ss,k,i}(t - t_s | \lambda_{ss}), i < n_2(t). \end{cases}$$

(22)

**Modeling multiple selected clones in a homeostatic tissue.** In the following, we generalize our model to account for the selection of

multiple competing clones in a homeostatic tissue. We use $\tau_c$ to denote the time at which a particular clone $c$ was born, where $c = 1$ identifies normal cells and $1 < c \leq C$ identifies the $C - 1$ selected clones. For each clone the variable $\upsilon_c$ reports the identity of its mother clone. We assume that all clones compete for a limited space with carrying capacity $N_{ss}$. As with the one-clone model, we implement clonal selection as a reduction in the loss rates, while leaving division rates unchanged. Denoting the competition coefficient between clone $c$ and other clones in the tissue with $\rho_{c,}$, we model the expected number of cells in clone $c$, $n_c$, with a system of ordinary differential equations,

$$\frac{dn_c}{dt} = \lambda_{ss} n_c \left(1 - \sum_{j;1 \leq j \leq C} \rho_{jc} \frac{n_j}{N_{ss}}\right)$$

(23)

with conditions

$$n_c(\tau_c) = 1,$$
$$n_c(t < \tau_c) = 0,$$
$$n_{\upsilon_c}(\tau_c) = n_{\upsilon_c}(\tau_c - dt) - 1.$$

(24)

The competition matrix $\rho$ is defined by the selective advantages, and, requiring $\Sigma_j n_j = N_{ss}$ at all times, is given by

$$\rho = \begin{pmatrix} 1 & 2 - r_2 & 2 - r_3 & \dots \\ r_2 & 1 & 2 - r_3/r_2 & \dots \\ r_3 & r_3/r_2 & 1 & \dots \\ \dots & \dots & \dots & \dots \end{pmatrix}$$

where $r_c$ models a reduction of cell loss in clone $c$ at $\tau_c$ and takes values $0 < r_c < 1$. Ultimately, we are interested in $S_i(t)$, the expected number of variants that are found in a clone of size $i$ at the time of measurement, $t$. As in the one-clone model, $S_i(t)$ has contributions from variants acquired in normal cells and variants acquired in the selected clones. In the following, we denote the SFS of variants acquired in a particular clone with $S_{i,c}(t)$. Hence,

$$S_i(t) = \sum_{c=1}^{C} S_{i,c}(t)$$

(25)

To evaluate $S_{i,c}(t)$, we denote with $\phi_c$ the set of subclones generated by a mutation in clone $c$, in the following termed 'daughters' of clone $c$, and with $\varphi \subset \phi_c$ the set of all possible combinations of $\phi_c$. Finally, the set $\Psi_c$ denotes the entire progeny of clone $c$, including daughters, granddaughters and so on. Consider a variant that was acquired in clone $c$ and is present in $k$ cells at time $t'$. This variant will, on average, be inherited to the daughter clone $d, d \in \phi_c$, with probability

$$\kappa(c, d, k, t') = H(\tau_d - t') \frac{k}{n_c(t')}$$

(26)

where $H$ is the heavyside step function and $\tau_d$ is the birth date of clone $d$. Considering a particular subset of daughters of clone $c$, $\varphi_j \in \varphi$, the probability of inheriting the variant to all daughters in this subset is, on average,

$$K(c, \varphi_j, k, t') = \overbrace{\prod_{l \in \varphi_j, \varphi_j \in \varphi} \kappa(c, l, k, t')}^{\text{presence in the subset of daughters } \varphi_j \in \varphi}$$
$$\times \underbrace{\prod_{l \in \{\phi_c \setminus \varphi_j\}, \varphi_j \in \varphi} (1 - \kappa(c, l, k, t'))}_{\text{absence in the remaining daughters}}$$

(27)

We compute $S_{i,c}(t)$ by considering all possible combinations of daughters that may inherit variants acquired in clone $c$. This yields

$$S_{i,c}(t)$$

$$= \int_{\tau_c}^{t} \overbrace{\mu\lambda_{ss}n_c(t')}^{\text{acquisition of variants}} \underbrace{\sum_{\varphi_j \in \varphi} K(c,\varphi_j,1,t')P_c\left(1,i-\sum_{k\in\varphi_j}n_k(t),t',t\right)}_{\text{all possible combinations of daugthers}}dt' +$$

*variants acquired in clone c and inherited to at least 1 daughter of clone c*

$$\int_{\tau_c}^{t} \overbrace{\mu\lambda_{ss}n_c(t')}^{\text{acquisition of variants}}\left(1-\sum_{\varphi_j\in\varphi}K(c,\varphi_j,1,t')\right)P_c(1,i,t',t)dt'$$

*variants acquired in clone c and not inherited to any daughter of clone c*

(28)

where $P_c(1,i,t',t)$ is the probability that a variant acquired in clone $c$ has drifted to $i$ cells within a time span $t-t'$. Specifically, $P_c(1,i,t',t)$ is defined by nonlinear birth–death processes, because the division and loss rates in clone $c$ change with time, subject to the dynamics in equation (23). In analogy to the one-clone model, we approximate this nonlinearity by modeling $S_{i,c}(t)$ with linear birth–death processes in a step-wise fashion, considering time intervals between clonal birth dates, $\tau$ turning points in $n(t)$ (as determined by numeric evaluation of equation (23)) and the end point as time points of interest. Specifically, we modeled the drift of somatic variants in each time interval using linear birth–death processes, which we parametrized with an effective loss rate of cells in clone $c$, $\delta_{\text{eff},c,t_a\le t<t_b}$, and which we evaluated for an effective time span $\Delta_{\text{eff},c,t_a\le t<t_b}$, where $t_a$ and $t_b$ denote the start and end point of the interval. To define $\delta_{\text{eff},c,t_a\le t<t_b}$ and $\Delta_{c,\text{eff},t_a\le t<t_b}$, we distinguished the case when the number of cells in clone $c$ increased ($n_c(t_b) > n_c(t_a)$) from the case when it decreased ($n_c(t_b) < n_c(t_a)$) during the time interval $(t_a,t_b)$. If $n_c(t_b) > n_c(t_a)$, we defined $\delta_{\text{eff},c,(t_a,t_b)} = r_c\lambda_{ss}$ and, to avoid overshooting the actual clone size, $\Delta_{\text{eff},c,(t_a,t_b)} = \frac{\log n_c(t_b)/n_c(t_a)}{\lambda_{ss}(1-r_c)}$. By contrast, if $n_c(t_b) < n_c(t_a)$, we parametrized $\delta_{\text{eff},c,(t_a,t_b)}$ such that the expected decline in a linear birth–death process equals the number of cells lost by competition dynamics. Hence, $\delta_{\text{eff},c,(t_a,t_b)}$ is defined by

$$\int_{t_a}^{t_b}\delta_{\text{eff},c,(t_a,t_b)}n_c(t_a)\exp((\lambda_{ss}-\delta_{\text{eff},c,(t_a,t_b)})t')dt'$$
$$= \int_{t_a}^{t_b}\lambda_{ss}n_c(t')\sum_{1\le j\le C}\rho_{jc}\frac{n_j(t')}{N_{ss}}dt' \quad (29)$$

where the right-hand side gives the number of death events in clone $c$ according to the competition dynamics in equation (23) and the left-hand side gives the number of death events according to a linear birth–death process. Moreover, if $n_c(t_b) < n_c(t_a)$, we defined $\Delta_{c,\text{eff},(t_a,t_b)} = t_b - t_a$. We then approximated $S_{i,c}(t)$ by recursively evaluating

$$S_{i,c}(t_b) = \sum_{k=1}^{N_{ss}}S_{k,c}(t_a)$$

$$\left[\overbrace{\sum_{\varphi_j\in\varphi}K(c,\varphi_j,k,t_a)P_{\text{exp},k,i-\sum_{l\in\varphi_j}n_l(t)}\left(\Delta_{\text{eff},c,(t_a,t_b)}|\lambda_{ss},\delta_{\text{eff},c,(t_a,t_b)}\right)}^{A}\right.$$

$$\left.+\overbrace{\left(1-\sum_{\varphi_j\in\varphi}K(c,\varphi_j k,t_a)\right)P_{\text{exp},k,i}\left(\Delta_{\text{eff},c,(t_a,t_b)}|\lambda_{ss},\delta_{\text{eff},c,(t_a,t_b)}\right)}^{B}\right]$$

$$+\overbrace{n_c(t_a)\int_{t_a}^{t_b}\mu\lambda_{ss}e^{(\lambda_{ss}-\delta_{\text{eff},c,(t_a,t_b)})(\Delta_{c,\text{eff},(t_a,t_b)})}P_{\text{exp},1,i}\left(\Delta_{\text{eff},c,(t_a,t_b)}|\lambda_{ss},\delta_{\text{eff},c,(t_a,t_b)}\right)}^{C}$$

(30)

until $t_b = t$. Here, terms (A) and (B) compute the drift of variants acquired before $t_a$, of which some (A) will be inherited to subclones of clone $c$, whereas others (B) will be found exclusively in clone $c$. Finally, term (C) computes the drift of variants newly acquired in the time interval $(t_a,t_b)$.

**Parameter estimation.** Model fits were obtained for each donor separately. In a first step, clonal selection was distinguished from genetic drift by applying the one-clone model of SCIFER to the data. For blood samples classified as selected and sequenced at high resolution (≥150× WGS), subsequent refinement was achieved by applying the two-clone model in a second step. Here, both possible topologies between two selected clones (linear and branched evolution) were tested, using prior distributions that were informed by the initial one-clone model fits (Supplementary Table 10; note that constraining the prior distributions based on the one-clone model fits facilitates the detection of additional subclones). All model fits were performed using ABC based on sequential Monte Carlo as implemented in pyABC[82] (posterior sample size of 1,000 and termination criterion $\varepsilon_{\min} = 0.05$). Briefly, ABC samples parameter sets from user-defined prior distributions (Supplementary Tables 9 and 10) and simulates the expected VAF distribution for each parameter set. The simulated VAF distributions are compared with the measured ones and the 1,000 parameter sets by minimal distance between measured and simulated data are selected. In each of the following iterations, the parameter set is expanded by adding random noise to the current parameter set, and the distance between model and data is reassessed. ABC is terminated once the distance between model and data is smaller than a defined threshold, $\varepsilon_{\min}$, yielding an estimate for the posterior distributions of the model parameters. We performed the following steps in each ABC iteration:

(1) [Experimental data]: Determine the cumulative VAF distribution at time $t$ for the experimental data. For 90× bulk WGS data from human BM or peripheral blood samples, include variants if they are supported by at least 3 reads, absent in the germline control sample, and if the locus is covered by at least 10 and at most 300 reads. For 270× bulk WGS data from human CD34+ HSPCs, include variants if they are supported by at least 3 reads. For pseudo-bulk WGS data, include all variants. For bulk WGS data from human brain, include all tier1–tier2 variants from ref. 17. Compute the cumulative number of VAFs $\ge f$, defined as $M_{\text{experimental},f} = \sum_{\text{VAF}=f}^{1}S_{\text{experimental,VAF}}$, where $S_{\text{experimental}}$ denotes the measured SFS and $S_{\text{experimental,VAF}}$ is the number of variants with a particular VAF. We evaluated $M_{\text{experimental},f}$ for bins with width 0.01, spanning $0.05 \le f \le 1$ for bulk WGS data with an average coverage <150×, $0.02 \le f \le 1$ for bulk WGS data with an average coverage ≥150×, and $0.01 \le f \le 1$ for pseudo-bulk data.

(2) [Model]: Sample parameters from their prior distributions (Supplementary Table 9, one-clone model; Supplementary Table 10, two-clone model).

(3) [Model]: Simulate the expected cumulative VAF distribution, $M_{\text{sim},f}(t_{\text{data}})$, where $t_{\text{data}}$ is the age of the patient and $f$ the minimal VAF (for numerical implementation of the model see Supplementary Note 4). The bins are as with the experimental data. Depending on the following criterion, based on the prior parameter sample, the cumulative VAF histogram is simulated with either the neutral model or the selection model: if a selected clone could grow above the detection limit with the prior parameter sample, the selection model is used; otherwise the neutral model is used. Formally, if $\frac{1}{2}e^{\lambda_{ss}^{\text{prior}}(1-r^{\text{prior}})(t_{\text{data}}-t_s^{\text{prior}})} \ge \gamma N_{SS}^{\text{prior}}$ (one-clone model) or if any $\frac{1}{2}n_{c,c>1}\left(t_{\text{data}}|\lambda_{SS}^{\text{prior}},\tau^{\text{prior}},r^{\text{prior}},\phi,\psi,N_{SS}^{\text{prior}}\right) \ge \gamma N_{SS}^{\text{prior}}$ (two-clone model) then the selection model is used. The detection limit is set to $\gamma = 0.025$ for 90× and 30× WGS data, and $\gamma = 0.005$ for pseudo-bulk and 270× bulk WGS data. Importantly, note that the selection model can return a posterior corresponding to neutral evolution.

(4) [Model]: Add the number of variants present in the founder cell of the hematopoietic system, $\Delta_{\text{clonal}}^{\text{prior}}$, to $M_{\text{sim},0.5}$ (corresponding to mutations acquired during early development).

(5) [Model]: Simulate experimental error of sequencing. To this end, generate the expected SFS for the $j$th frequency bin,

$S_{sim}(f_j) = M_{sim}(f_j) - M_{sim}(f_{j+1})$. For bulk WGS data, sample for each simulated variant a sequencing coverage $\vartheta$ from a Poisson distribution with mean $\hat{\vartheta}$ corresponding to the average sequencing depth. Thereafter, sample VAFs for each variant according to the $\frac{B(f,\vartheta)}{\vartheta}$, where $B$ denotes the binomial distribution and $f$ is the true VAF in the tissue. Discard variants supported by less than 3 reads. For pseudo-bulk WGS data, sample VAFs for each variant according to $\frac{B(2f,n_{cells})}{(2n_{cells})}$, where $n_{cells}$ is the number of sequenced single-cell clones. Compute the sampled cumulative VAF distribution, $M_{sim,f,sampled}$.

(6) [Model versus Experimental data]: Determine the distance function for ABC

$$d = \sum_f \left( M_{sim,f,sampled} - M_{experimental,f} \right)^2$$

**Classification of cases as neutrally evolving or as selected.** We classified pseudo-bulks and samples sequenced at <150× as selected if at least 15% of the posterior samples report a selected clone size ≥0.1. These thresholds were defined based on in silico generated test data, and correspond to the WGS depth of 90× (Fig. 2). WGS data with ≥150× coverage allow for higher resolution and hence, we classified cases as selected if at least 15% of the posterior samples report a selected clone size ≥0.04 according to the one-clone model, and as harboring two selected clones if at least 15% of the posterior samples report a size ≥0.04 for both selected clones. Upon sample classification, we computed the 80% highest density intervals for each parameter on the parameter subsets supporting neutral evolution or clonal selection, respectively.

## Simulation of phylogenetic trees and in silico evaluation of model performance

We validated the population genetics model with simulated data, generated according to stochastic birth–death processes (see Supplementary Note 5 for a description of the simulations and https://github.com/VerenaK90/SCIFER for their computational implementation).

Simulations used to evaluate model performance (Fig. 2) were run for stem cells only. Simulations of neutral evolution were parametrized with $N_{ss,S} = 25{,}000$, $\mu = 1$, $\lambda_{ss,S} = 10$ per year, $\delta_{ss,S} = 10$ per year and with $\tau = 250$ (that is, summarizing 1% of the reactions occurring in 25,000 stem cells). Simulations of clonal selection were parametrized with $N_{ss,S} = 25{,}000$, $\mu = 1$, $\lambda_{ss,S} = 10$ per year, $\delta_{ss,S} = 10$ per year, $t_s = 20$ years, $s = 0.02$ and with $\tau = 2{,}500$.

Simulations of neutral evolution used to assess the effect of differentiation into a single progenitor cell population on the VAF histogram were parametrized with $N_{ss,S} = 1{,}000$, $\mu = 1$, $\lambda_{ss,S} = 1$ per year, $\delta_{ss,S} = 1$ per year, $N_{ss,P} = 2{,}500$, $\lambda_{ss,P} = 4.6$ per year, $\delta_{ss,S} = 5$ per year and $\tau = 2{,}500$.

Simulations of neutral evolution in a heterogenous tissue where stem cells differentiate into two progenitor cell types during development, but continue to produce only one of them during adulthood were parametrized with $N_{ss,S} = 1{,}000$, $\mu = 1$, $\lambda_{ss,S} = 1$ per year, $\delta_{ss,S>P_1} = 1$ per year, $\delta_{ss,S>P_2} = 0$ per year, $N_{ss,P_1} = 6{,}000$ $\lambda_{ss,P_1} = 3$ per year, $\delta_{ss,P_1} = 3.167$ per year, $N_{ss,P_2} = 4{,}000$, $\lambda_{ss,P_2} = 0$ per year, $\delta_{ss,P_2} = 0$ per year and $\tau = 1{,}000$.

**Parameter inference.** To infer the dynamic stem cell parameters ($\mu$, $\delta_{exp}$, $\lambda_{ss,S}$, $N_{ss,S}$, $t_s$, $r$) (Fig. 2), we subsampled 10,000 cells from the simulated trees at time points specified in Supplementary Table 1 and computed the simulated VAF distribution from subsampled trees. To account for technical noise, we simulated sequencing by sampling for each variant a sequencing coverage $\vartheta \propto Pois(\hat{\vartheta})$, where $\vartheta$ is the average sequencing coverage and thereafter sampling mutant reads according to $B(VAF, \vartheta)$, where $B$ is the binomial distribution. We simulated VAFs for average sequencing coverages of 30×, 90× and 270×, and fitted the population genetics model to the simulated bulk WGS data as described above for the real data.

**Sensitivity and specificity of detecting selected clones.** To evaluate the sensitivity and specificity of the out model, we analyzed the posterior probability of clonal selection. In accordance with the minimal clone sizes used in the inference setup (see above), we computed the probability of clonal selection as $P(\text{selection}) = \frac{\sum_i \frac{n_2(t_{data}|\theta_i)}{N_{ss,i}} \geq 0.05}{\sum_i 1}$ for 30× and 90× sequencing depths, and as $P(\text{selection}) = \frac{\sum_i \frac{n_2(t_{data}|\theta_i)}{N_{ss,i}} \geq 0.01}{\sum_i 1}$ for 270× sequencing depth, where $n_2(t_{data}|\theta_i)$ is the size of the selected clone at the patient age, $t_{data}$, given the $i$th parameter sample, $\theta_i$, and $N_{ss,i}$ is the $i$th estimate of the stem cell number. Among all parameter sets reporting a clone size ≥0.05 (30× and 90× coverage) or ≥0.01 (270× coverage), we determined median clone size and, if they were at least 5% in size, rounded it with 5% accuracy, or else rounded them with 1% accuracy. Thereafter, we computed the sensitivity and specificity of our approach, by classifying cases as selected if $P(\text{selection}) \geq \beta$, varying the threshold $\beta$ between 1% and 100%. For each of the true clone sizes (1%, 2%, 5%, 10%, 15%, 20%, 25%, 50%, 75%; Supplementary Table 1), we computed the number of true positives as the number of cases where the actual clone size was correctly inferred (we classified inferred clone sizes between 0.5 and 1.5 of the actual clone size as correct). Conversely, we computed the number of false positives as the number of cases in which SCIFER erroneously reported a particular clone size, albeit the actual clone size was zero.

The difference between true positives and false positives was maximal for $\beta = 15\%$. At this threshold, clones of size ≥5% VAF were reliably inferred (90×).

## Statistics and reproducibility

Bayesian parameter inference and statistical analyses were conducted with pyABC[82] v.0.12.6, using python v.3.10.1 and R (v.4.2.0 and v.4.2.1). We used the following R packages: ape[83] v.5.6-2, phytools[84] v.1.2-0, phangorn[85] v.2.10.0, castor[86] v.1.7.5, TreeTools v.1.8.0, deSolve[87] v.1.33, openxlsx v.4.2.5, cdata v.1.2.0, ggpubr v.0.4.0, RRphylo[88] v.2.7.0, ggplot2 (ref. [89]) v.3.4.2, cgwtools v.3.3, ggVennDiagram v.1.2.2, ggbeeswarm v.0.6.0, ggsci v.2.9, Hmisc v.4.7.1, lemon v.0.4.5, data.table v.1.14.2, RColorBrewer v.1.1.3, ggridges v.0.5.4, doParallel v.1.0.17, foreach v.1.5.2, parallel v.2.1, wesanderson v.0.3.6, bedr v.1.0.7, ggformula v.0.10.2, HDInterval v.0.2.2, reshape2 v.1.4.4 (ref. [90]), dplyr v.1.0.9 and scales v.1.2.1. Flow cytometry data was analyzed with FlowJo v.10.8.1.

## Reporting summary

Further information on research design is available in the Nature Portfolio Reporting Summary linked to this article.

## Data availability

Single-cell WGS data were part of previously published studies[2,12,13]. WGS data from these studies are deposited at the European Genome-Phenome Archive (https://www.ebi.ac.uk/ega/) under accession nos. EGAD00001004086, EGAD00001007851 and EGAD00001007684. Substitution calls from these studies are deposited on Mendeley Data (https://doi.org/10.17632/yzjw2stk7f.1; ref. [91]), (https://doi.org/10.17632/np54zjkvxr.2; ref. [92]) and on figshare (https://doi.org/10.6084/m9.figshare.15029118; ref. [93]). WGS data generated in this study (aligned bam files) are available at the European Genome-Phenome Archive under accession no. EGAS00001007558. The bam files contain all relevant meta data for back conversion into fastq files and realignment. Variant calls and model fits have been made available on Mendeley data[94] (https://doi.org/10.17632/gkz-vmg5f6z.1). Patient information and driver mutations are available as Supplementary Information to this manuscript. We also used the following publicly available datasets: hg19 reference genome (https://ftp.ensembl.org/pub/grch37/release-99/fasta/homo_sapiens/dna/Homo_sapiens.GRCh37.dna.primary_assembly.fa.gz), gnomAD

v.2.1.1. (https://storage.googleapis.com/gcp-public-data–gnomad/release/2.1.1/vcf/genomes/gnomad.genomes.r2.1.1.sites.vcf.bgz), repeat regions and simple repeat regions (downloaded from UCSC table browser, setting the assembly to hg19, the track to 'RepeatMasker' or 'Simple Repeats'), annovar (version May2018; http://annovar.openbioinformatics.org/), dbSNP v.150 (https://ftp.ncbi.nlm.nih.gov/snp/organisms/human_9606_b150_GRCh37p13/VCF/00-All.vcf.gz), Clinvar (version 20221231, https://ftp.ncbi.nlm.nih.gov/pub/clinvar/vcf_GRCh37/archive_2.0/2023/clinvar_20221231.vcf.gz) and manually curated variants from Uniprot (https://ftp.uniprot.org/pub/databases/uniprot/current_release/knowledgebase/variants/homo_sapiens_variation.txt.gz).

## Code availability
Code to reproduce the analysis is available via GitHub at https://github.com/VerenaK90/SCIFER (ref. 95) and https://github.com/VerenaK90/Clonal_hematopoiesis (ref. 96).

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

## Acknowledgements

We thank the participants and allied healthcare professionals who supported collection of bone marrow and blood samples. V.K. acknowledges funding by the Deutsche Forschungsgemeinschaft (DFG, German Research Foundation – project no. 526169089). T.H. was supported by the Deutsche Forschungsgemeinschaft (project no. 497777320), the German Federal Ministry of Education and Research (BMBF; project no. 01KD2206B – SATURN3), the DKFZ-MOST cooperation program, Project Ca 211 and DKFZ core funding. P.V. acknowledges funding from the Medical Research Council Molecular Haematology Unit Programme Grant (MC_UU_00029/8), Blood Cancer UK Programme Continuity Grant 13008, NIHR Senior Fellowship and the Oxford Biomedical Research Centre Haematology Theme. N.A.J. was supported by a Medical Research Council and Leukaemia UK Clinical Research Training Fellowship (MR/R002258/1). M.M., B.U. and M.A.S. were funded by the Haematology Theme of the Oxford NIHR Biomedical Research Centre. Research by T.H. was supported in part by grant NSF PHY-2309135, the Gordon and Betty Moore Foundation grant no. 2919.02, and the Chan Zuckerberg Initiative DAF grant to the Kavli Institute for Theoretical Physics (KITP).

We thank D. Cruz Hernandez, A. Groom, N. Becker, M. Günther and all members of the Höfer and Vyas groups for discussions. The authors are grateful for support by the MRC WIMM Flow Cytometry and Single Cell Facilities and the DKFZ NGS core facility.

## Author contributions

V.K., N.A.J., P.V. and T.H. conceived the project. S.N., B.J.L.K., A.H.T., R.A.-L., R.G., B.W., K.W., D.B., S.G.D., A.J.C. and A.P. collected bone marrow and blood samples. N.A.J. performed flow cytometry. N.A.J., M.M., R.M., B.U. and M.A.S. performed DNA extraction. N.C. prepared DNA sequencing libraries. V.K. and N.A.-P. analyzed whole-genome sequencing data. E.T. helped to analyze data for the revision. F.E. helped with phylogenetic inference. V.K. developed the mathematical model, with input from T.H., performed parameter estimation and all computations. V.K., T.H. and P.V. wrote the manuscript with input from all coauthors. T.H. and P.V. acquired funding and administered the project.

## Funding

## Competing interests

The authors declare no competing interests.

## Additional information

**Extended data** is available for this paper at https://doi.org/10.1038/s41588-025-02217-y.

**Correspondence and requests for materials** should be addressed to Verena Körber, Paresh Vyas or Thomas Höfer.

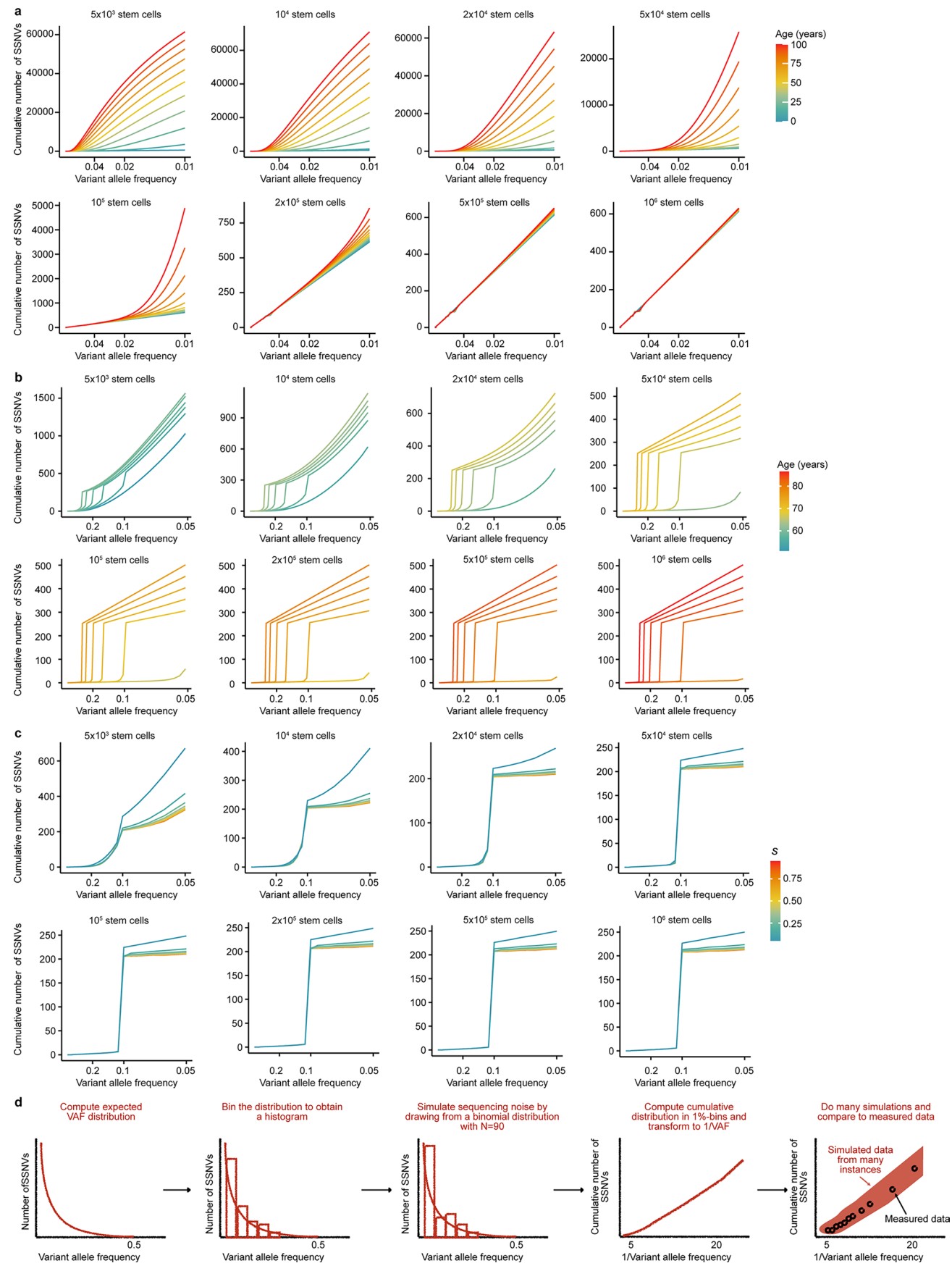

**Extended Data Fig. 1 | See next page for caption.**

**Extended Data Fig. 1 | Quantifying selection and drift from bulk whole genome sequencing data. a**, Simulated cumulative variant allele frequency (VAF) distribution of somatic single nucleotide variants (SSNVs) at selected ages between 0 and 100 years for stem cell numbers ranging between 5,000 and 1 M; $\lambda$=5/year, $\mu$=10/division. **b**, Simulated cumulative VAF distribution of SSNVs when a driver is acquired at 20 years of age, and the selected clone grows by 22% per year ($s$ = 0.02, $\lambda$=10/year, $\mu$=1/division, $N_{ss}$ = 25,000). Shown are the time points at which the clone sizes reached 5%, 10%, 15%, 20%, 25%, or 30% VAF for stem cell numbers ranging between 5,000 and 1 M. **c**, Simulated cumulative VAF distribution of somatic SSNVs when a driver is acquired at 20 years of age and has reached a clone size of 10% VAF ($\lambda$=10/year, $\mu$=1/division, $N_{ss}$ = 25,000). Shown are the VAF distributions for selective advantages ($s$) ranging between 0.05 and 0.95, and stem cell numbers ranging between 5,000 and 1 M. **d**, Parameter estimation with approximate Bayesian computation. First, the expected variant allele frequency histogram is analytically computed. Then, sequencing noise is simulated by drawing from a binomial distribution with average 90x coverage. The modelled cumulative distribution is compared to the measured data for varying VAFs (1% step size).

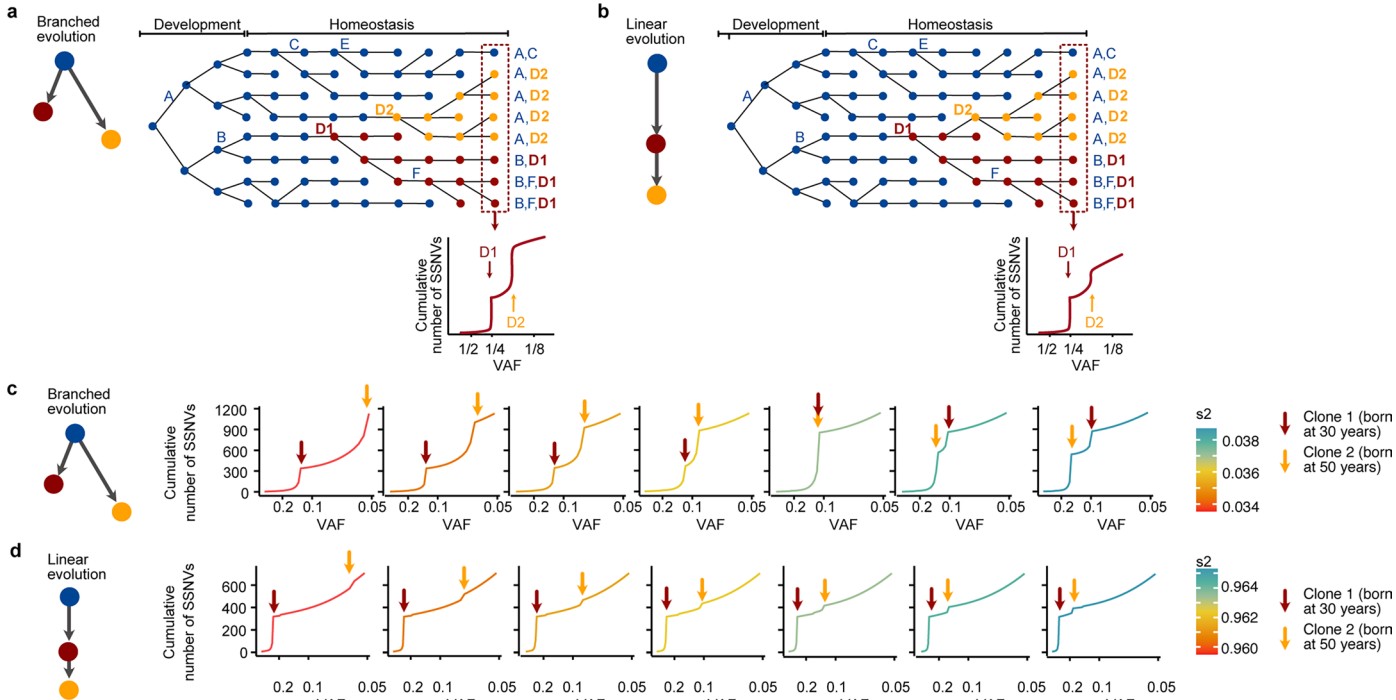

**Extended Data Fig. 2 | Modeling two selected clones in stem cell homeostasis.**
**a**, Schematic illustrating variant accumulation during development and
subsequent homeostasis, with two selected clones (red and orange) emerging
by branched evolution during homeostasis. **b**, Schematic illustrating variant
accumulation during development and subsequent homeostasis, with
two selected clones (red and orange) emerging by linear evolution during
homeostasis. **c**, Simulated cumulative variant allele frequency (VAF) distribution
of somatic single nucleotide variants (SSNVs) when two selected clones, founded
at 30 years and 50 years, evolve in parallel. The selective advantage of the

first clone is fixed at 0.018, while the selective advantages of the second clone
varies between 0.033 and 0.038 (color-encoded; 25,000 stem cells; $\lambda$=10/year,
$\mu$=1/division; arrows highlight positions of the selected clones). **d**, Simulated
cumulative VAF distribution of SSNVs when two selected clones, founded at 30
years and 50 years, evolve linearly. The selective advantage of the first clone is
fixed at 0.018, while the selective advantages of the second clone varies between
0.041 and 0.035 (color-encoded; 25,000 stem cells; $\lambda$=10/year, $\mu$=1/division;
arrows highlight positions of the selected clones).

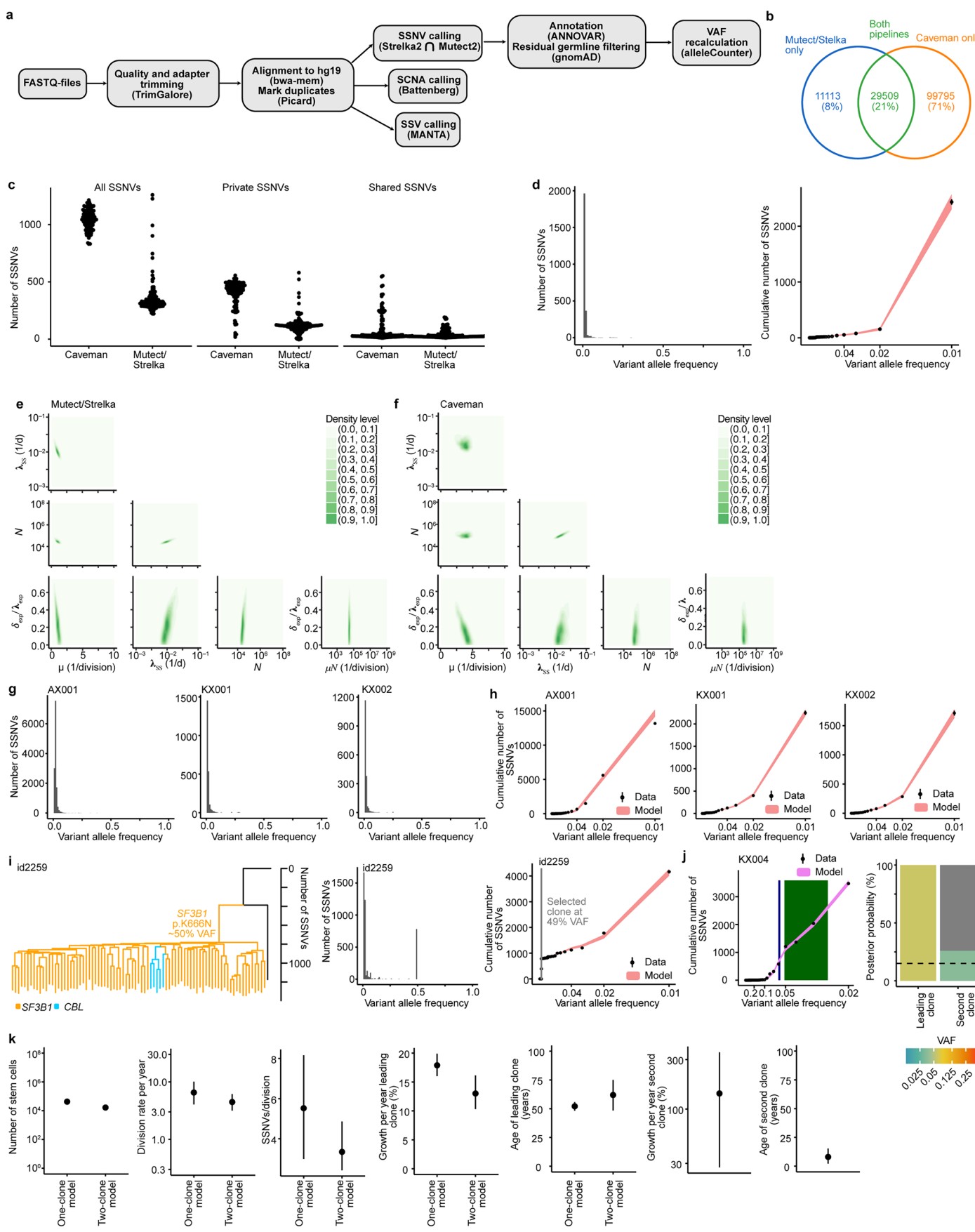

**Extended Data Fig. 3 | See next page for caption.**

**Extended Data Fig. 3 | Quantifying drift and selection in published data.**
**a**, Somatic variant calling pipeline (using hg19). SSNV, somatic single nucleotide variant. SCNA, somatic copy number abnormality. SSV, somatic structural variant. **b**, Number of shared and unique SSNVs in 140 whole genomes from hematopoietic stem cand progenitor cell (HSPC) clones from published data[2] identified by Caveman (original publication) and our pipeline (shown in **a**). **c**, Number of SSNVs of different classes identified by each variant caller. **d**, Left, variant allele frequency (VAF) distribution (truncated at 0.01 VAF) of SSNVs identified by Caveman[2]. Right, model fit to the cumulative 1/VAF distribution shown in left (points and error bars, measured data and their standard deviation, which, assuming Poisson-distributed measurements, is the square root of the measured data; red area, 95% posterior probabilities of the model fit, estimated from simulations using 100 posterior samples). **e**, Two-dimensional posterior probability distributions for the parameters estimated by SCIFER using SSNVs identified with Mutect2 and Strelka in published data[2]. Axis limits, range of prior distributions. **f**, As in **e**, but using SSNVs identified with Caveman. Note that the posterior for the fraction of cell loss,

$\delta_{exp}/\lambda_{exp}$, is relatively broad, but has little effect on the mutation rate and other parameters (see Supplementary Note 2). **g**, As in **d** (left panel) but for samples AX001, KX001 and KX002 without clonal selection (SSNVs from published data[13]). **h**, As in **d** (right panel) but for the data shown in **g**. **i**, Left, phylogenetic tree of published sample id2259[12] (reconstruction from the original publication). Middle, VAF distribution (truncated at 0.01 VAF) of SSNVs (taken from[12]). Right, as in **d**, but for id2259 (grey area, 80% credible interval of the clone size, estimated from 1,000 posterior samples). **j**, Left, as in **d** but for sample KX004 when allowing for two selected subclones (based on the phylogeny, we assumed branched evolution, c.f. Figure 3g; blue and green areas, 80% credible intervals of the size of leading and second selected clone, estimated from 1,000 posterior samples). Right, posterior probability for clonal selection obtained with the 2-clone model (conditioned on clones with VAF ≥ 2%). **k**, Median and 80% credible interval (estimated from 1,000 posterior samples) for stem cell and selection parameters for KX004, obtained with a one-clone and two-clone model.

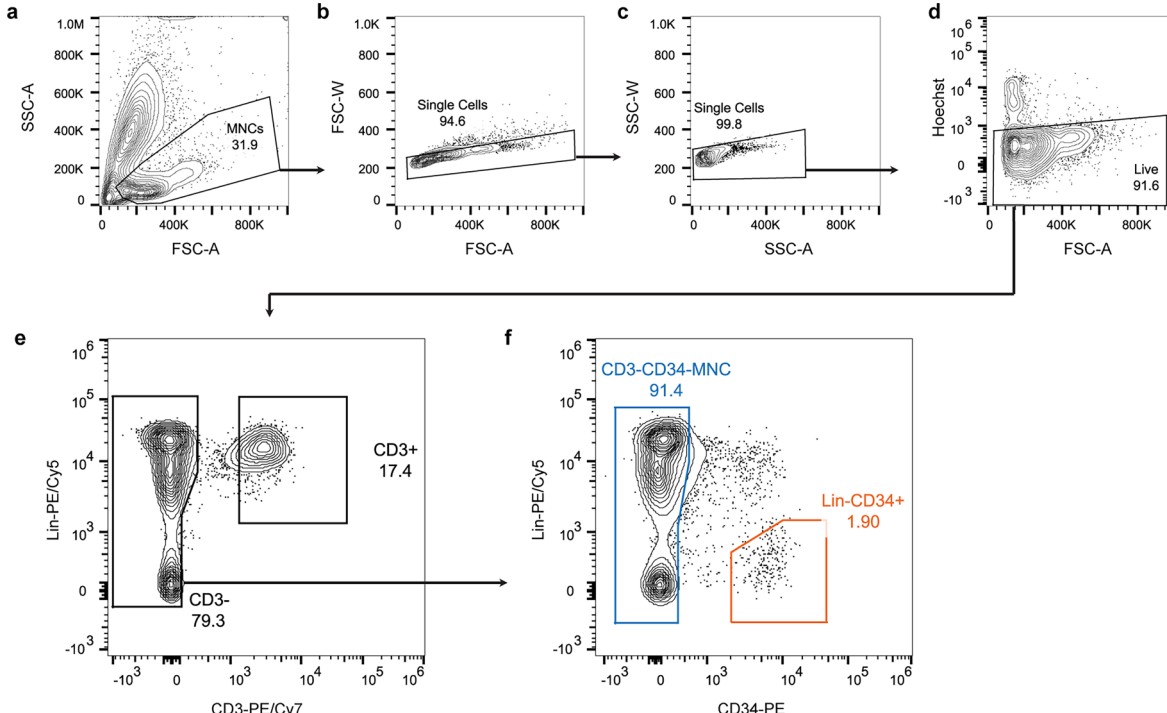

**Extended Data Fig. 4 | FACS gating for hematopoietic stem and progenitor cells. a-f**, Cells were stained with a panel of antibodies (Methods), then single and live mononuclear cells were sorted (MNCs, **a-d**). T cell depleted MNCs (MNC(–T)) were sorted from the CD3-, CD34+ cell fraction (**e** and **f**), and, hematopoietic stem and progenitor cells were sorted from the CD3-Lin-CD34+ cell fraction (**f**). Image created with FlowJo v10.8.1; values in **a-f** give percentages.

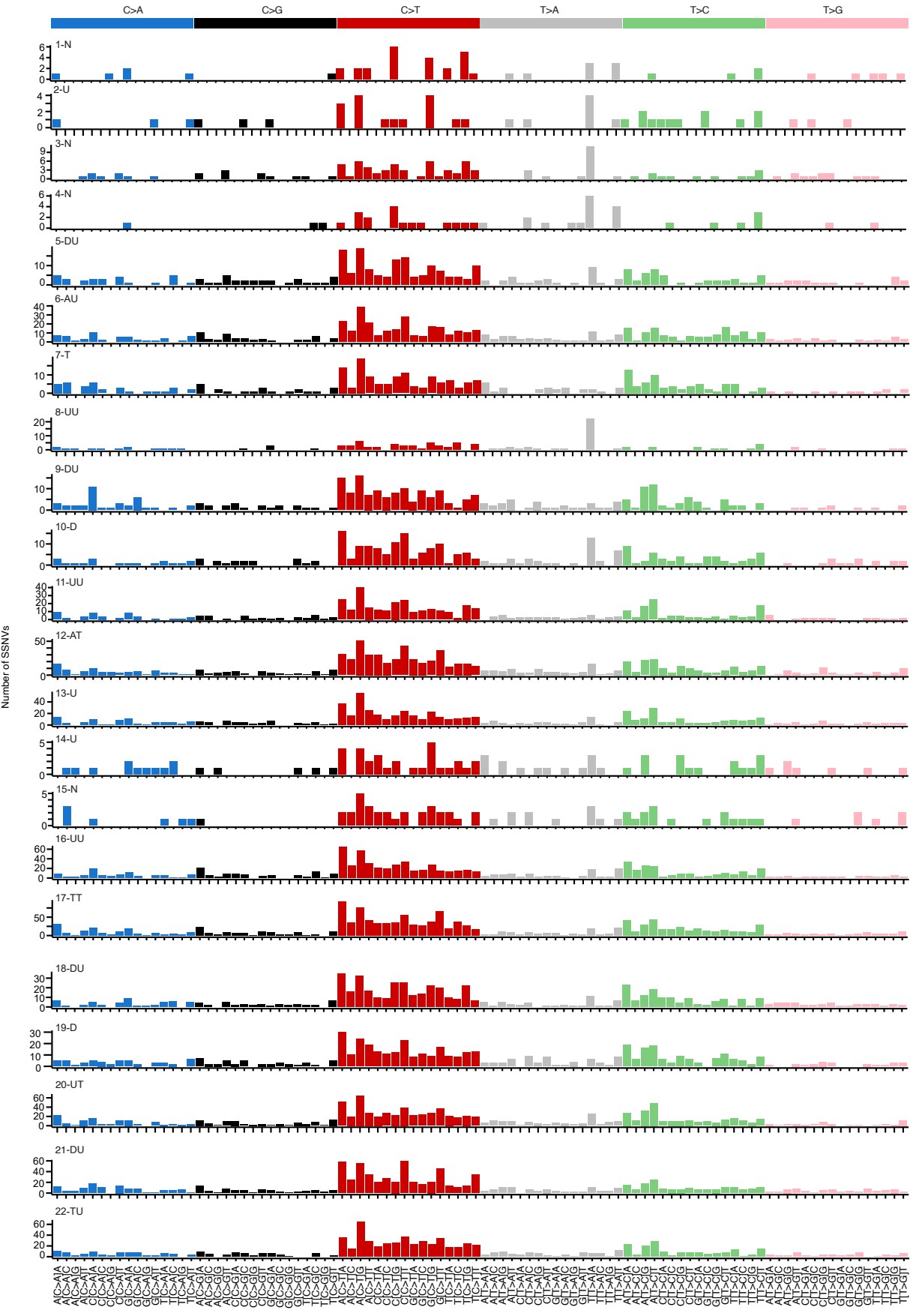

**Extended Data Fig. 5 | Somatic variants in hematopoietic stem and progenitor cells.** Genome-wide profile of trinucleotide substitution patterns (x-axis) (either from 270x whole genome sequencing (WGS) data, where available, or from 90x WGS data) across the genome in CD34+ hematopoietic stem and progenitor cells from the samples indicated.

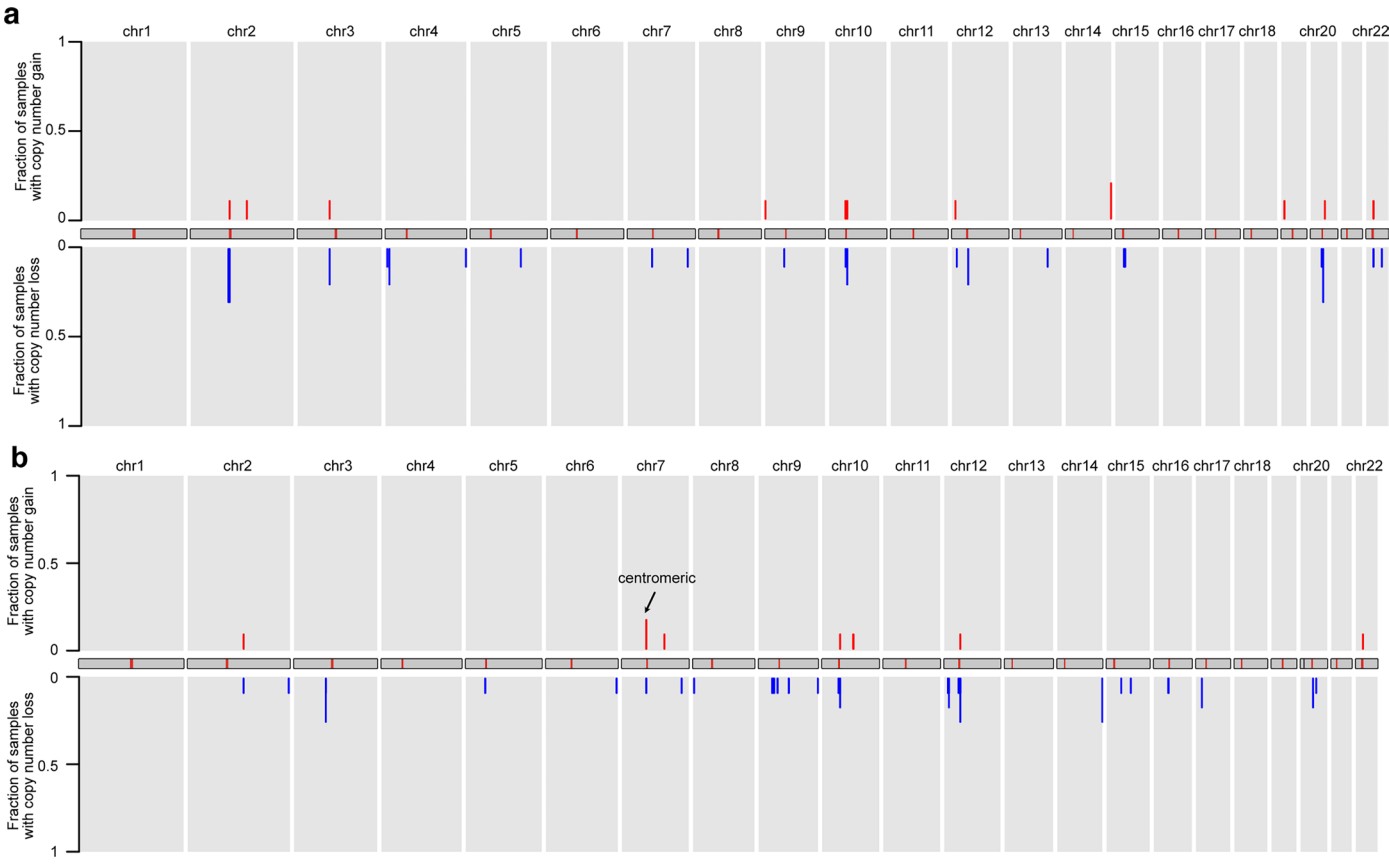

**Extended Data Fig. 6 | Copy number variants in hematopoietic stem and progenitor cells. a**, Genome-wide (chromosomes indicated and arrayed across x-axis) profiles of copy number gains (red) and losses (blue) in CD34⁺ hematopoietic stem and progenitor cells (HSPCs) across the 10 samples without known clonal hematopoiesis (CH) driver. **b**, Genome-wide (chromosomes indicated and arrayed across x-axis) profiles of copy number gains (red) and losses (blue) in CD34⁺ HSPC across the 12 samples with known CH driver.

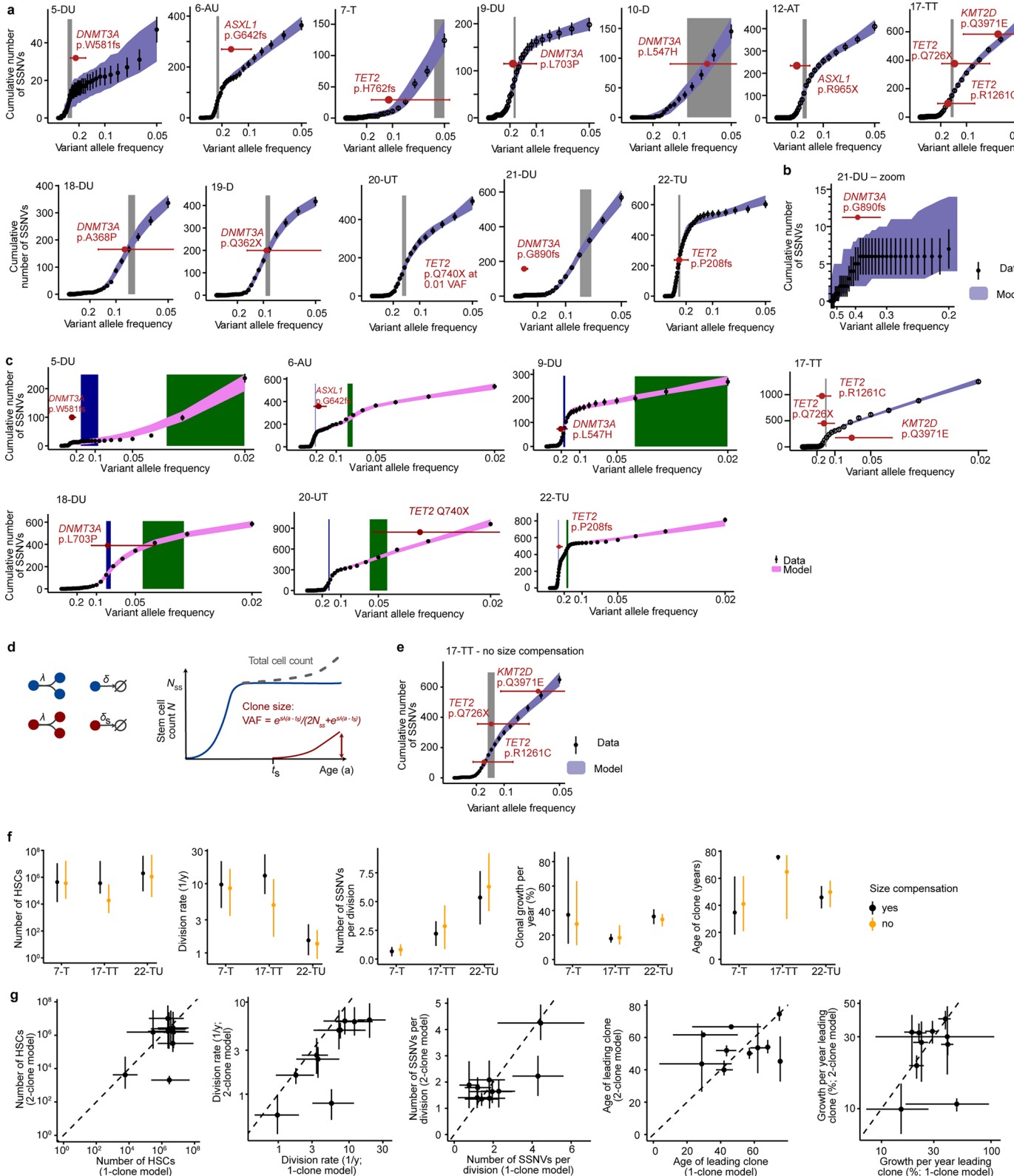

**Extended Data Fig. 7 | See next page for caption.**

**Extended Data Fig. 7 | Quantifying clonal selection from bulk whole genome sequencing. a**, Model fits to cumulative variant allele frequency (VAF) distributions measured by 90x whole genome sequencing (WGS) of CD34[+] hematopoietic stem and progenitor cells (HSPCs) from twelve individuals with known clonal hematopoiesis (CH) driver mutations (points and error bars, measured data and standard deviation, which, assuming Poisson-distributed measurements, is the square root of the measured data). Violet areas, 95% posterior probabilities of model fit, estimated from simulations using 100 posterior samples. Grey areas, 80% credible interval for the clone size, estimated from 1,000 posterior samples; red points and error bars, mean and 95% confidence interval of the VAF of known CH drivers, based on binomial distributions with sample size/success probability of 140/0.23 (5-DU), 168/0.15 (6-AU), 102/0.11(7-T), 89/0.18 (9-DU), 77/0.06 (10-D), 126/0.21(12-AT), 204/0.16 (17-TT), 230/0.13 (17-TT), 226/0.06 (17-TT), 70/0.09 (18-DU), 127/0.09 (19-D), 55/0.05 (20-UT), 156/0.39 (21-DU), 103/0.27 (22-TU), corresponding to read coverage and measured VAF, respectively. **b**, As in **a**, but showing an expanded view (VAF ≥ 0.2) for sample 21-DU (90x WGS). Selection (inferred at 7% VAF) was not associated with the *DNMT3A* mutation (43% VAF). At most 6 variants were

acquired prior to the *DNMT3A* mutation, suggesting acquisition during early development. **c**, As in **a**, but for 270x WGS from seven individuals with known CH driver mutations. Colored rectangles, 80% credible interval for the clone size of leading and second selected clone, estimated from 1,000 posterior samples (where applicable); red points and error bars, mean and 95% confidence interval of the VAF of known CH drivers, based on binomial distributions with sample size/success probability of 270/0.26 (5-DU), 272/0.18 (6-AU), 230/0.2 (9-DU), 264/0.14 (17-TT), 275/0.16 (17-TT), 279/0.07 (17-TT), 194/0.08 (18-DU), 249/0.03 (20-UT), 260/0.28 (22-TU), corresponding to read coverage and measured VAF, respectively. **d**, Scheme for a model variant where a selected clone (red) expands without replacing normal cells (blue). **e**, As in **a**, but using the model variant introduced in **d**. **f**, Estimated model parameters obtained with SCIFER and the model variant introduced in **c** for three cases with a mutation in *TET2* (points and error bars, median and 80% credible intervals, estimated from 1,000 posterior samples). **g**, Estimated model parameters obtained with the one-clone model and the two-clone model. Points and error bars, median and 80% credible estimates, estimated from 1,000 posterior samples; dashed line, bisectrix.

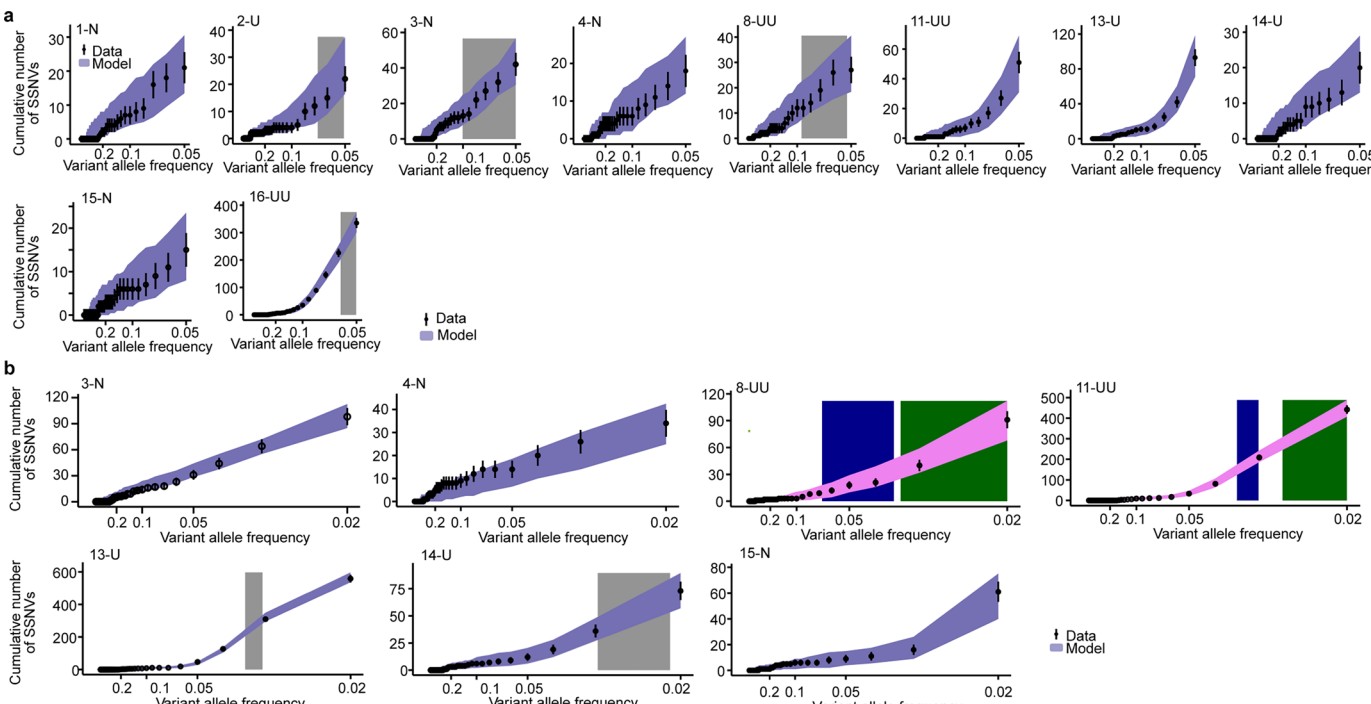

**Extended Data Fig. 8 | Quantifying neutral drift and clonal selection in cases without known driver from bulk whole genome sequencing. a**, Model fits to the cumulative variant allele frequency (VAF) distributions measured by 90x whole genome sequencing (WGS) of CD34+ hematopoietic stem and progenitor cell (HSPC) samples from ten individuals (1-N, 2-U, 3-N, 4-N, 8-UU, 11-UU, 13-U, 14-U, 15-N, 16-UU) without known clonal hematopoiesis (CH) driver mutation (points and error bars, mean and standard deviation of the measured data, where standard deviations were computed based on Poisson distributions with mean corresponding to the respective somatic single nucleotide variant (SSNV) count; purple areas show the 95% posterior probabilities of the model fit, estimated from simulations using 100 posterior samples). The model fits of 1-N, 4-N and 15-N show no evidence of selection; the model fits of 3-N shows weak evidence of selection, which is, however, invalidated when probed with 270x WGS

(shown in **b**). **b**, Model fits to the cumulative VAF distributions measured by 270x WGS of CD34+ HSPC samples from seven individuals (3-N, 4-N, 8-UU, 11-UU, 13-U, 14-U, 15-N, 16-UU) without known CH driver mutation (points and error bars, mean and standard deviation of the measured data, where standard deviations were computed based on Poisson distributions with mean corresponding to the respective SSNV count; purple areas show the 95% posterior probabilities of the model fit, estimated from simulations using 100 posterior samples). Colored rectangles, 80% credible interval for the estimated clone size of the leading and second selected clone inferred with SCIFER (where applicable). The model fits show no evidence of selection for 3-N, 4-N and 15-N but evidence for one selected clone in 13-U and 14-U (grey rectangles) and for two selected clones in 8-UU and 11-UU (blue and green rectangles; no 270x WGS was available for 2-U).

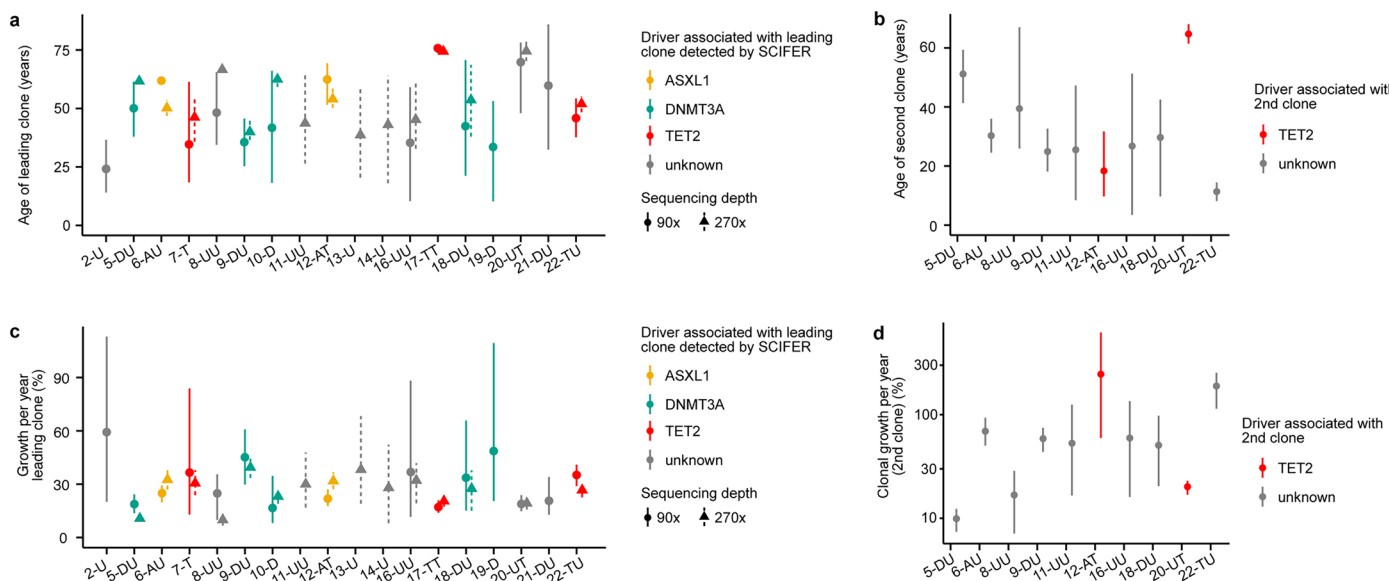

**Extended Data Fig. 9 | Selection dynamics of leading and subsequent clones.** **a**, Estimated age of the leading selected clone (for the twelve cases with known clonal hematopoiesis (CH) drivers and the six cases with unknown drivers introduced in Figs. 4 and 5; points and error bars, median and 80% credible intervals, estimated from 1,000 posterior samples; parameter estimates are compared between model fits obtained from 90x and 270x whole genome sequencing (WGS) data). **b**, Estimated age of the second selected clone (for the ten cases introduced in Figs. 4 and 5; points and error bars, median and 80% credible intervals, estimated from 1,000 posterior samples). **c**, Estimated clonal growth rate of the leading selected clone (for the twelve cases with known drivers and the six cases with unknown drivers introduced in Figs. 4 and 5; points and error bars, median and 80% credible intervals, estimated from 1,000 posterior samples; parameter estimates are compared between model fits obtained from 90x and 270x WGS data). **d**, Estimated clonal growth rate of the second selected clone (for the ten cases introduced in Figs. 4 and 5; points and error bars, median and 80% credible intervals, estimated from 1,000 posterior samples).

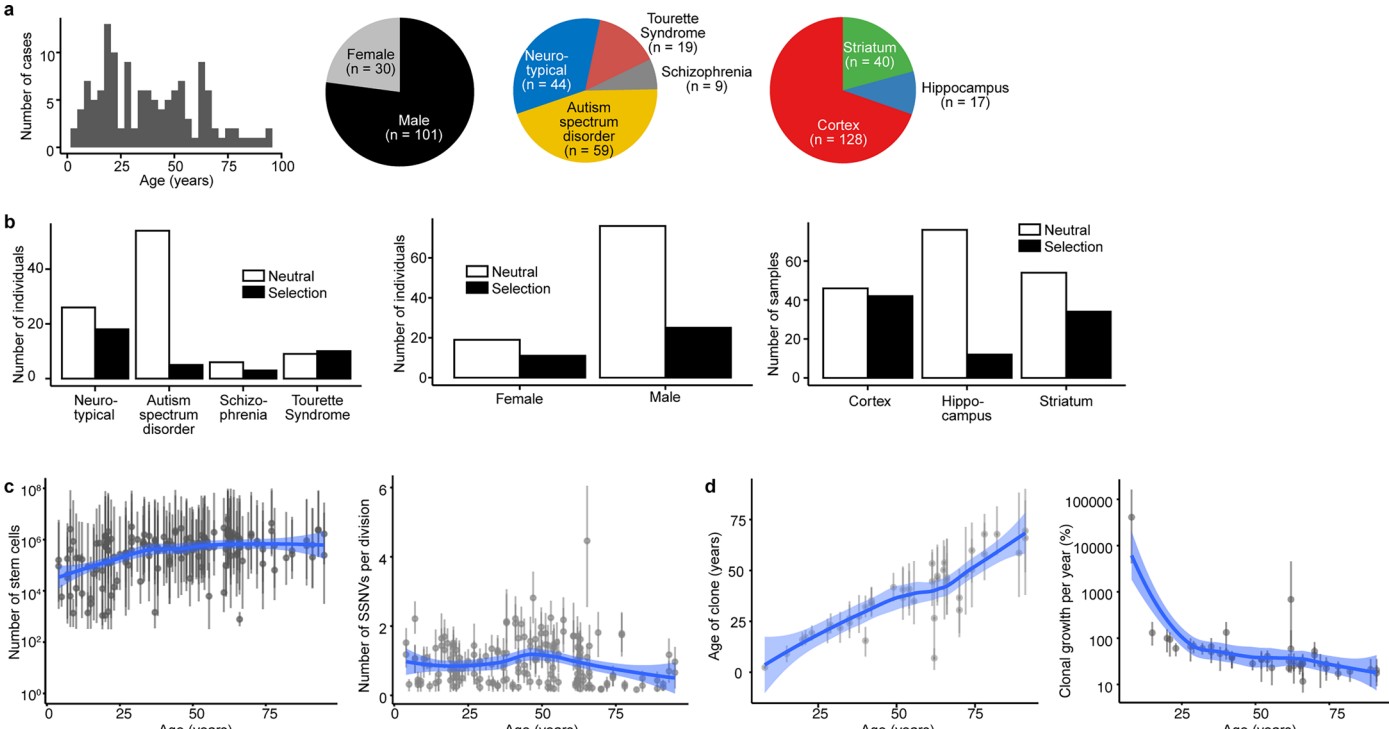

**Extended Data Fig. 10 | Quantifying clonal selection in human brain samples.** **a**, Cohort characteristics, showing age, sex and phenotype of the profiled individuals, as well as location of the analyzed samples. **b**, Number of individuals with evidence for clonal selection or neutral evolution stratified by phenotype (left), sex (middle), and location (right). **c**, Estimated number of stem cells (left) and number of somatic single nucleotide variants (SSNVs) per division (right) plotted against age for the 185 analyzed brain samples (points and error bars, median and 80% credible intervals, estimated from 1,000 posterior samples; blue line and shaded area, LOESS regression and 95% confidence interval). **d**, Estimated age of the selected clone (left) and clonal growth rate (right) plotted against age for the 44 analyzed brain samples with evidence for clonal selection (points and error bars, median and 80% credible intervals, estimated from 1,000 posterior samples; blue line and shaded area, LOESS regression and 95% confidence interval).

# Reporting Summary

## Statistics

For all statistical analyses, confirm that the following items are present in the figure legend, table legend, main text, or Methods section.

| n/a | Confirmed | |
|---|---|---|
| ☐ | ☒ | The exact sample size ($n$) for each experimental group/condition, given as a discrete number and unit of measurement |
| ☐ | ☒ | A statement on whether measurements were taken from distinct samples or whether the same sample was measured repeatedly |
| ☐ | ☒ | The statistical test(s) used AND whether they are one- or two-sided<br>*Only common tests should be described solely by name; describe more complex techniques in the Methods section.* |
| ☐ | ☒ | A description of all covariates tested |
| ☒ | ☐ | A description of any assumptions or corrections, such as tests of normality and adjustment for multiple comparisons |
| ☐ | ☒ | A full description of the statistical parameters including central tendency (e.g. means) or other basic estimates (e.g. regression coefficient) AND variation (e.g. standard deviation) or associated estimates of uncertainty (e.g. confidence intervals) |
| ☐ | ☒ | For null hypothesis testing, the test statistic (e.g. $F$, $t$, $r$) with confidence intervals, effect sizes, degrees of freedom and $P$ value noted<br>*Give P values as exact values whenever suitable.* |
| ☐ | ☒ | For Bayesian analysis, information on the choice of priors and Markov chain Monte Carlo settings |
| ☒ | ☐ | For hierarchical and complex designs, identification of the appropriate level for tests and full reporting of outcomes |
| ☒ | ☐ | Estimates of effect sizes (e.g. Cohen's $d$, Pearson's $r$), indicating how they were calculated |

*Our web collection on statistics for biologists contains articles on many of the points above.*

## Software and code

Policy information about availability of computer code

| Data collection | n/a |
|---|---|
| Data analysis | Sequencing reads were mapped with bwa mem v0.7.12, coordinate-sorted with samtools v1.5 and duplicates were marked with GATK MarkDuplicates v4.0.9.0.<br><br>Variant calling:<br>Strelka v2.9.2<br>Mutect2, GATK v4.2.0.0<br>bedtools v2.24.0<br>dbSNP v150<br>gnomAD v2.1.1<br>bcftools v1.10.2<br>alleleCounter v4.0.2<br>MaC v0 (https://github.com/nansari-pour/MaC)<br>Battenberg v2.2.10<br>Manta v1.6.0<br><br>Data analysis was performed with R (v4.2.0 and v4.2.1) and python v3.10.1 using the following packages:<br>SCIFER v2.0.2 (https://doi.org/10.5281/zenodo.14507248)<br>pyABC v0.12.6<br>ape v5.6-2 |

phytools v1.2-0
phangorn v2.10.0
castor v1.7.5
TreeTools v1.8.0
deSolve v1.33
openxlsx v4.2.5
cdata v1.2.0
ggpubr v0.4.0
RRphylo v2.7.0
ggplot2 v3.4.2
cgwtools v3.3
ggVennDiagram v1.2.2
ggbeeswarm v0.6.0
ggsci v2.9
Hmisc v4.7.1
lemon v0.4.5
data.table v1.14.2
RColorBrewer v1.1.3
ggridges v0.5.4
doParallel v1.0.17
foreach v1.5.2
parallel v2.1
wesanderson v0.3.6
bedr v1.0.7
ggformula v0.10.2
HDInterval v0.2.2
reshape2 v1.4.4
dplyr v1.0.9
scales v1.2.1

Custom code used for data analysis (https://doi.org/10.5281/zenodo.14627371).

Flow cytometry data was analyzed with FlowJo v10.8.1.

For manuscripts utilizing custom algorithms or software that are central to the research but not yet described in published literature, software must be made available to editors and reviewers. We strongly encourage code deposition in a community repository (e.g. GitHub). See the Nature Portfolio guidelines for submitting code & software for further information.

## Data

Policy information about availability of data

All manuscripts must include a data availability statement. This statement should provide the following information, where applicable:
- Accession codes, unique identifiers, or web links for publicly available datasets
- A description of any restrictions on data availability
- For clinical datasets or third party data, please ensure that the statement adheres to our policy

Single-cell WGS data were part of previously published studies.2,12,13 WGS data from these studies are deposited at the European Genome-Phenome Archive (https://www.ebi.ac.uk/ega/) under accession nos. EGAD00001004086, EGAD00001007851 and EGAD00001007684. Substitution calls from these studies are deposited on Mendeley Data (https://doi.org/10.17632/yzjw2stk7f.191, https://doi.org/10.17632/np54zjkvxr.292) and on figshare (https://doi.org/10.6084/m9.figshare.1502911893). WGS data generated in this study (aligned bam files) are available at the European Genome-Phenome Archive under accession no. EGAS00001007558. The bam files contain all relevant meta data for back conversion into fastq files and re-alignment. In accordance with the laws of data protection, data are deposited under controlled access. Access can be granted by contacting Paresh Vyas (paresh.vyas@imm.ox.ac.uk) and requires a data access agreement; requests will be replied to within 4 weeks. Variant calls and model fits have been made available on Mendeley data94 (https://doi.org/10.17632/gkzvmg5f6z.1). Patient information and driver mutations are available as Supplementary Data to this manuscript. We also used the following publicly available datasets: hg19 reference genome (https://ftp.ensembl.org/pub/grch37/release-99/fasta/homo_sapiens/dna/Homo_sapiens.GRCh37.dna.primary_assembly.fa.gz) , gnomAD v2.1.1. (https://storage.googleapis.com/gcp-public-data--gnomad/release/2.1.1/vcf/genomes/gnomad.genomes.r2.1.1.sites.vcf.bgz), repeat regions and simple repeat regions (downloaded from UCSC table browser, setting the assembly to hg19, the track to "RepeatMasker" or "Simple Repeats") , annovar (version May2018; http://annovar.openbioinformatics.org/), dbSNP v150 (https://ftp.ncbi.nlm.nih.gov/snp/organisms/human_9606_b150_GRCh37p13/VCF/00-All.vcf.gz), Clinvar (version 20221231, https://ftp.ncbi.nlm.nih.gov/pub/clinvar/vcf_GRCh37/archive_2.0/2023/clinvar_20221231.vcf.gz), manually curated variants from Uniprot (https://ftp.uniprot.org/pub/databases/uniprot/current_release/knowledgebase/variants/homo_sapiens_variation.txt.gz)

## Research involving human participants, their data, or biological material

Policy information about studies with human participants or human data. See also policy information about sex, gender (identity/presentation), and sexual orientation and race, ethnicity and racism.

| Reporting on sex and gender | Patients' sex is reported in Supplementary Table 2. |
| Reporting on race, ethnicity, or other socially relevant groupings | n/a |
| Population characteristics | Patient characteristics (age, clinical information, etc.) are reported in Supplementary Table 2. |

| Recruitment | All eligible subjects were approached by GCP trained clinical staff involved in the routine clinical care of the subjects. We selected individuals to span an age range between 30 and 89 years, with a balanced representation of both genders. We selected 12 individuals with known CH drivers (at least one driver with VAF>3%) and 10 individuals without known CH drivers to get a balanced representation of individuals with and without known CH drivers. |
|---|---|
| Ethics oversight | This study was approved by the Yorkshire & The Humber - Bradford Leeds Research Ethics Committee (REC Ref: 17/YH/0382). |

Note that full information on the approval of the study protocol must also be provided in the manuscript.

# Field-specific reporting

Please select the one below that is the best fit for your research. If you are not sure, read the appropriate sections before making your selection.

☒ Life sciences　　☐ Behavioural & social sciences　　☐ Ecological, evolutionary & environmental sciences

For a reference copy of the document with all sections, see nature.com/documents/nr-reporting-summary-flat.pdf

# Life sciences study design

All studies must disclose on these points even when the disclosure is negative.

| Sample size | 12 individuals with known CH drivers >=5% VAF and 10 individuals without known CH drivers >=1% VAF were selected based on prior characterization of CH status with targeted deep sequencing. This is an individual-based study, aiming at in-depth characterization of individual cases rather than population-wide statistics. To obtain good statistical power for this approach, we sequenced whole genomes of these samples at high coverage (90x). For 19 of the 22 individuals with sufficient DNA available, we resequenced libraries to a total coverage of 270x. |
|---|---|
| Data exclusions | No data were excluded |
| Replication | Individual bone marrow samples were sequenced and their genome-wide somatic variant profile was analyzed with mathematical modelling. This is an individual-based study and hence no replication was performed. |
| Randomization | This is not an individual-based study and not a case-control study. Hence no replication was performed. |
| Blinding | This is not an individual-based study and not a case-control study. Hence no blinding was performed. |

# Reporting for specific materials, systems and methods

We require information from authors about some types of materials, experimental systems and methods used in many studies. Here, indicate whether each material, system or method listed is relevant to your study. If you are not sure if a list item applies to your research, read the appropriate section before selecting a response.

| Materials & experimental systems | | Methods | |
|---|---|---|---|
| n/a | Involved in the study | n/a | Involved in the study |
| ☐ | ☒ Antibodies | ☒ | ☐ ChIP-seq |
| ☒ | ☐ Eukaryotic cell lines | ☐ | ☒ Flow cytometry |
| ☒ | ☐ Palaeontology and archaeology | ☒ | ☐ MRI-based neuroimaging |
| ☒ | ☐ Animals and other organisms | | |
| ☐ | ☒ Clinical data | | |
| ☒ | ☐ Dual use research of concern | | |
| ☒ | ☐ Plants | | |

# Antibodies

| Antibodies used | mouse anti-human CD34-PE (Biolegend, clone 581, #343505, FC)<br>mouse anti-human CD3-PE/Cy7 (Biolegend, clone HIT3a, #300316, FC)<br>mouse anti-human CD2-PE/Cy5 (Biolegend, clone RPA-2.10, #300209, FC)<br>mouse anti-human CD4-PE/Cy5 (Biolegend, clone RPA-T4, #300509, FC)<br>mouse anti-human CD7-PE/Cy5 (Biolegend, clone CD7-6B7, #343110, FC)<br>mouse anti-human CD8a-PE/Cy5 (Biolegend, clone RPA-T8, #301009, FC)<br>mouse anti-human CD11b-PE/Cy5 (Biolegend, clone ICRF44, #301307, FC)<br>mouse anti-human CD14-PE/Cy5 (eBioscience, clone 61D3, #15-0149-42, FC)<br>mouse anti-human CD19-PE/Cy5 (Biolegend, clone HIB19, #302209,FC )<br>mouse anti-human CD20-PE/Cy5 (Biolegend, clone 2H7, #302307, FC)<br>mouse anti-human CD56-PE/Cy5 (Biolegend, clone MEM188, #304607, FC) |
|---|---|

mouse anti-human CD235ab-PE/Cy5 (Biolegend, clone HIR2, #306605, FC)

| Validation | anti-CD34-PE was validated for immunofluorescent staining with flow cytometry by staining human peripheral blood mononuclear cells with 581 PE or PE mouse IgG1 isotype control and CD45 (HI30) PerCP (gated on CD14- population) on the manufacture's webpage |
|---|---|

anti-CD3-PE/Cy7 was validated for immunofluorescent staining with flow cytometry by staining human peripheral blood lymphocytes with HIT3a PE/Cyanine7 on the manufacturer's webpage
anti-CD2-PE/Cy5 was validated for immunofluorescent staining with flow cytometry by staining human peripheral blood lymphocytes with RPA-2.10 PE/Cyanine4 on the manufacturer's webpage
anti-CD4-PE/Cy5 was validated for immunofluorescent staining with flow cytometry by staining human peripheral blood lymphocytes with RPA-T4 PE/Cyanine5 on the manufacturer's webpage
anti-CD7-PE/Cy5 was validated for immunofluorescent staining with flow cytometry by staining human peripheral blood lymphocytes with CD7-6B7 PE/Cyanine5 on the manufacturer's webpage
anti-CD19-PE/Cy5 was validated for immunofluorescent staining with flow cytometry by staining human peripheral blood lymphocytes with HIB19 PE/Cyanine5 on the manufacturer's webpage
anti-CD20-PE/Cy5 was valiated for immunofluorescent staining with flow cytometry by staining human peripheral blood lymphocytes with anti-CD20 (clone 2H7) PE/Cyanine5 or mouse IgG2b, Kappa PE/Cyanine5 on the manufacturer's webpage.
anti-CD56-PE/Cy was validated for immunofluorescent staining with flow cytometry by staining human peripheral blood lymphocytes with MEM-188 PE/Cyanine5 on the manufacturer's webpage.
anti-CD8a-PE/Cy5 was validated for immunofluorescent staining with flow cytometry by staining human whole blood on the manufacturer's webpage.
anti-CD11b-PE/Cy5 was validated for immunofluorescent staining with flow cytometry by staining human peripheral blood lymphocytes, monocytes, and granulocytes with ICRF44 PE/Cyanine5 on the manufacturer's webpage
anti-CD14-PE/Cy5 was validated for immunofluorescent staining with flow cytometry by staining normal human peripheral blood cells with Mouse IgG1 K Isotype Control PE-Cyanine5 or Anti-Human CD14 PE-Cyanine5 on the manufacturer's webpage. Cells in the monocyte gate were used for analysis.
anti-CD235ab-PE/Cy5 was validated for immunofluorescent staining with flow cytometry by staining human red blood cells with HIR2 PE/Cyanine5 on the manufacturer's webpage

## Clinical data

Policy information about clinical studies
All manuscripts should comply with the ICMJE guidelines for publication of clinical research and a completed CONSORT checklist must be included with all submissions.

| Clinical trial registration | n/a this is a retrospective study of individual bone marrow samples without clinical intervention. The study was approved by the Yorkshire & The Humber - Bradford Leeds Research Ethics Committee (REC Ref: 17/YH/0382). |
|---|---|
| Study protocol | n/a this is a retrospective study of individual bone marrow samples without clinical intervention. The study was approved by the Yorkshire & The Humber - Bradford Leeds Research Ethics Committee (REC Ref: 17/YH/0382). |
| Data collection | All eligible subjects were approached by GCP trained clinical staff involved in the routine clinical care of the subjects. The study was approved by the Yorkshire & The Humber - Bradford Leeds Research Ethics Committee (REC Ref: 17/YH/0382). |
| Outcomes | n/a this is a retrospective study of individual bone marrow samples without clinical intervention. The study was approved by the Yorkshire & The Humber - Bradford Leeds Research Ethics Committee (REC Ref: 17/YH/0382). |

## Flow Cytometry

### Plots

Confirm that:

☒ The axis labels state the marker and fluorochrome used (e.g. CD4-FITC).

☒ The axis scales are clearly visible. Include numbers along axes only for bottom left plot of group (a 'group' is an analysis of identical markers).

☒ All plots are contour plots with outliers or pseudocolor plots.

☒ A numerical value for number of cells or percentage (with statistics) is provided.

### Methodology

| Sample preparation | Patient samples were collected from individuals undergoing elective total hip replacement surgery. At the time of surgery, trabecular bone fragments and bone marrow (BM) aspirates were obtained from the femoral canal and collected in anticoagulated buffer containing acid-citrate-dextrose, heparin sodium and DNase. BM mononuclear cells (MNCs) were isolated by Ficoll density gradient centrifugation and viably frozen.<br>For cell sorting, thawing media was prepared with IMDM medium (Gibco) supplemented with 20% fetal bovine serum (FBS) and 110 μg/mL DNase. BM samples were thawed at 37°C in a water bath, 1 mL warm FBS was added, and the suspension then diluted by dropwise addition of 8 mL thawing media. The suspension was centrifuged at 400 g for 10 mins, cells were resuspended in flow cytometry staining medium (IMDM with 10% FBS and 10 μg/mL DNase), filtered through a 35 μm cell strainer, and placed on ice.<br>Cells were stained with the following antibodies: anti-CD34-PE (1:160, Biolegend, clone 581), anti-CD3-PE/Cy7 (1:100, Biolegend, clone HIT3a), anti-CD2-PE/Cy5 (1:160, Biolegend, clone RPA-2.10), anti-CD4-PE/Cy5 (1:160, Biolegend, clone RPA- |
|---|---|

T4), anti-CD7-PE/Cy5 (1:160, Biolegend, clone CD7-6B7), anti-CD8a-PE/Cy5 (1:320, Biolegend, clone RPA-T8), anti-CD11b-PE/Cy5 (1:160, Biolegend, clone ICRF44), anti-CD14-PE/Cy5 (1:160, eBioscience, clone 61D3), anti-CD19-PE/Cy5 (1:160, Biolegend, clone HIB19), anti-CD20-PE/Cy5 (1:160, Biolegend, clone 2H7), anti-CD56-PE/Cy5 (1:80, Biolegend, clone MEM188), and anti-CD235ab-PE/Cy5 (1:320, Biolegend, clone HIR2). Following antibody incubations, cells were washed with 1 mL flow cytometry staining buffer, centrifuged at 350 g for 5 min and resuspended in flow cytometry staining buffer containing 1:10,000 Hoechst 33342 live/dead stain.

| | |
|---|---|
| Instrument | BD FACSAria Fusion or Sony MA900 equipped with a 100 µm nozzle or sorting chip |
| Software | Acquisition: Sony Cell Sorter Software; Analysis: FlowJo v10.8.1 |
| Cell population abundance | BM cell populations (Lin–CD34+ HSPCs and CD34–CD3– MNCs) were sorted with a mean purity > 95% for DNA extraction and whole genome sequencing. Purity was determined by flow cytometry of post-sort fractions. |
| Gating strategy | Unstained, single stained and Fluorescence Minus One (FMO) controls were used to determine background staining and compensation in each channel. Extended Data Fig. 3 shows full gating strategy. Gating was on live cells (FSC-A vs. SSC-A), doublet exclusion (FSC-A vs. FSC-W, followed by SSC-A vs. SSC-W), dead cell exclusion (FSC-A vs. Hoechst), and then on immunophenotypic markers as described. |

☒ Tick this box to confirm that a figure exemplifying the gating strategy is provided in the Supplementary Information.

