## [Peer Review File · Nature Genetics]

Detecting and quantifying clonal selection in somatic stem cells

Corresponding Author: Professor Thomas Hofer

Version 0:

Decision Letter:

11th Dec 2023

Dear Thomas,

Firstly, thank you very much to you and your co-authors for your patience! We were holding out for a final reviewer with important expertise, but given the delay and a lack of response from them, we have decided to go ahead with a decision at this point.

Your Article, "Detecting and quantifying clonal selection in somatic stem cells" has now been seen by 2 referees. You will see from their comments copied below that while they find your work of considerable potential interest, they have raised quite substantial concerns that must be addressed. In light of these comments, we cannot accept the manuscript for publication, but would be very interested in considering a revised version that addresses these serious concerns.

In brief, the two referees express some support for the aims of SCIFER, but sound unconvinced that - in its current form - your work offers a Nature Genetics-level advance.

Referee #1 says that your study could be of interest, but makes a range of nuanced and insightful comments on the assumptions of the model, and how they relate to what is known about HSPC and CH biology.

Reviewer #2 suggests the technical basis of SCIFER is sound, but states that no new insights into CH are offered; they suggest that application of SCIFER to a broader range of WGS data is required to improve this biological novelty.

In our reading of these reviews, the clear message is that this biological novelty (with respect to CH, or more broadly) must be improved; we think there are some useful suggestions given (although we presume that application to relatively lower-coverage WGS data may not be the best use of SCIFER), but leave it up to you and your co-authors to decide which avenues to pursue.

Unfortunately, the referee we secured with expertise in population genetic analysis did not submit a review; as this is an important and fundamental part of your manuscript, we will be seeking to cover this in a review of a revision.

We hope you will find the referees' comments useful as you decide how to proceed. If you wish to submit a substantially revised manuscript, please bear in mind that we will be reluctant to approach the referees again in the absence of major revisions.

To guide the scope of the revisions, the editors discuss the referee reports in detail within the team, including with the chief editor, with a view to identifying key priorities that should be addressed in revision and sometimes overruling referee requests that are deemed beyond the scope of the current study. We hope that you will find the prioritised set of referee points to be useful when revising your study. Please do not hesitate to get in touch if you would like to discuss these issues further.

If you choose to revise your manuscript taking into account all reviewer and editor comments, please highlight all changes in the manuscript text file. At this stage we will need you to upload a copy of the manuscript in MS Word .docx or similar editable format.

*2) If you have not done so already please begin to revise your manuscript so that it conforms to our Article format instructions, available here. Refer also to any guidelines provided in this letter.

Please be aware of our guidelines on digital image standards.

Link Redacted

If you wish to submit a suitably revised manuscript we would hope to receive it within 6 months. If you cannot send it within this time, please let us know. We will be happy to consider your revision so long as nothing similar has been accepted for publication at Nature Genetics or published elsewhere. Should your manuscript be substantially delayed without notifying us in advance and your article is eventually published, the received date would be that of the revised, not the original, version.

Nature Genetics is committed to improving transparency in authorship. As part of our efforts in this direction, we are now requesting that all authors identified as 'corresponding author' on published papers create and link their Open Researcher and Contributor Identifier (ORCID) with their account on the Manuscript Tracking System (MTS), prior to acceptance. ORCID helps the scientific community achieve unambiguous attribution of all scholarly contributions. You can create and link your ORCID from the home page of the MTS by clicking on 'Modify my Springer Nature account'. For more information please visit please visit www.springernature.com/orcid.

Thank you for the opportunity to review your work.

Sincerely,

Michael Fletcher, PhD
Senior Editor, Nature Genetics

ORCID: 0000-0003-1589-7087

Referee expertise: blood cancers, including clonal haematopoiesis; cancer risk/genetics.

Reviewers' Comments:

Reviewer #1:

Remarks to the Author:

In their study, Körber et al., propose a unique method, SCIFER, to infer the population dynamics of tissue stem cells, discriminating between neutral evolution vs. positive selection, on the basis of the distribution of the cumulative number of SNVs according to VAF value in a given tissue sample using whole genome sequencing. The authors first modeled the clonal dynamics under neutrality and positive selection, to which the observed data in bone marrow CD34+ cells were fitted to investigate the dynamics of hematopoiesis with and without clonal hematopoiesis. After benchmarking SCIFER using published single colony sequencing data, the authors applied the method to newly obtained whole genome sequencing data of bulk bone marrow samples from 22 cases, including seven cases without evidence of clonal selection or known CH mutations, 10 with ≥ 1 CH mutations, and 5 showing clonal selection with no known CH-mutations. The authors found accelerated stem cell division of all stem cells in CH, compared to age-matched non-CH cases, suggesting altered bone marrow environment in CH-positive cases. Finally, they concluded that SCIFER can be applied to other renewing somatic tissues to detect and quantify clonal selection. The results are interesting and potentially merit publication. Meanwhile, this

reviewer raises several issues to be addressed before the decision is made.

Major issues:

- 1) The method to infer the hematopoietic stem cell (HSC) dynamics, SCIFER, is unique and of interest. Meanwhile, most of the result obtained using SCIFER have been previously described using colony sequencing combined with bulk sequencing (refs #2, #22, #26, #33). The increased cell division rate of all stem cells could be novel but needs further validation.
- 2) The mathematical formulations of the model $S_i(t; \mu, \delta, N, \lambda\mu N, r)$ and (8) and (14)-(18), are quite complex and difficult to comprehend. Thus, for better understanding of the model, several instances of the model need to be presented in an Extended Figure, by showing $S_i(t)$ for a wider range of parameters (N, μ, λ) than presented in Fig. 1e-l, in particular for a wider variety of stem cell numbers (N_{ss}) to clarify a big discontinuity between $N=50,000$ and $N=500,000$. For clonal selection, the instance should be presented for the terminal clone size of 5, 10, 15, 20, 25 and 30, and for a variety of N_{ss} , assuming the SSNVs are measured 10, 20, 40, and 50 years after the acquisition of a driver mutation.
- 3) As shown previously (for example, Moran-Cusio et al., Cancer Cell, 2011), TET2 deficiency resulted in an expanded stem cell pool. Thus, N_{ss} in CH(+) cases might not be constant but gradually increase over years. Does this model can lead to a different model and estimation of critical parameters?
- 4) There has been a debate regarding the dynamics of hematopoietic stem cells (HSC) vs. differentiated progenitors based on lineage-tracing study in the mouse model. Some studies suggest a relatively small efflux from HSCs to progenitors where the mature progenitors differentiated from HSC replenish hematopoiesis for long time, which could be replaced in a short period by the progenitors in the next generation (Sun et al., Nature, 2014). For example, the analysis of the life history of myeloproliferative neoplasms demonstrated that a large clade in MPN phylogenies could be easily disappear (Fig. 2 in #33). Alternatively, those progenitors long lived without achieving equilibrium within the mouse lifetime (Busch et al., Nature, 2015; Pei et al., Nature, 2017). In these cases, the measured SSNVs does not correctly reflect those in the stem cell compartment, where N_{ss} will be inflated because of the presence of differentiated progenitors.
 - How could these different underlying stem cell dynamics influence the interpretation of the current results?
 - Could the authors analyze samples from different time points from the same individuals?
- 5) The authors claim increased time-averaged HSC division rate for all stem cells in CH-positive patients. How the authors confirmed that the division rate of both mutated and unmutated stem cells increased compared with those cells from age-matched individuals?
- 6) The model of clonal selection seems to assume one selected clone, but some cases used for benchmarking had ≥ 2 major clones. How were these cases analyzed or detected by SCIFER. Did the authors find any cases in which ≥ 2 large clones were suspected? Did those cases create multiple shoulders on the graph?

Minor issues:

- 7) How were the parameters inferred by parameter fitting for those cases having multiple selected clones. The iterations in parameter estimation using ABC in page 45 are not detailed, where a minimum description of the ABC framework is needed for most readers to understanding how iteration works. Moreover, the expression of $M_i(t)$ in L1080 might not be correct, where no j explicitly appears in $S_{i,1}(t)$.
- 8) Derivation of several formula in the method section need more explanation to easy follow-up. For example, it might not be easy to follow the transformation to (18) (+1061). Every expression should be rigorously defined without any ambiguity. L1085-1088, f is not well-defined. Does it express the position of bin? $Pois(c)(L1109)$.
- 9) Legends in Fig. 3 are incorrect. For example, the legend for f should be that for h ?

Reviewer #2:

Remarks to the Author:

In their manuscript Korber et al. present a statistical method to study mutation dynamics using bulk whole genome sequencing data. Their method is similar to other previous work (e.g PACER). However in addition to leveraging somatic mutation burden to infer age at driver acquisition, they also use VAF distribution taken from WGS to infer mutation selection and division rate. They use simulations and publicly available data to show that application of SCIFER to bulk GWS is able to generate similar results to single cell sequencing data. While some of the math is beyond my comfort level, the overall approach seems reasonable with a few points that I think could be clarified that I detail below. The application of their method to existing data does not reveal new insights into clonal hematopoiesis. The impact of the method could come from application of the method to higher depth WGS datasets where their simulations suggest SCIFER could be used to infer clonality with a similar resolution to single cell sequencing.

A few additional specific comments below

1. Their method does not seem to take into account potential differences in selection of CH mutations over the life course of an individual. This has been suggested to occur on a gene specific-basis in CH.
2. In Figure 3l and p I am unsure what "stem cell number" represents and how this was calculated.
3. CH mutations including TET2 are known to impact self-renewal. This increased replicative capacity has been cited to explain why individuals with CH have shorter telomeres (yet longer telomeres predispose to CH). Thus, an increased HSC division rate would be expected.

Reviewer #3:

None

Version 1:

Decision Letter:

2nd Oct 2024

Dear Thomas,

Thank you to you and your co-authors for your patience during this prolonged review process. As previously discussed, we solicited a new Referee #3 with the specific population genetics/mathematical expertise we thought was missing from the original round of review. In this round, the original Referee #2 was unfortunately not timely in submitting a review, but we made an editorial check of your responses and given the other reports we are prepared to proceed without their feedback on the revision and your responses to their specific comments.

Your Article, "Detecting and quantifying clonal selection in somatic stem cells" has now been seen by 2 referees. You will see from their comments below that while they find your work of interest, some important points are raised. We are interested in the possibility of publishing your study in Nature Genetics, but would like to consider your response to these concerns in the form of a revised manuscript before we make a final decision on publication.

Briefly, Referee #1 appreciates the improvement and the application to the brain data; but they have a number of new comments, which help to clarify SCIFER's new two-clone model and the application/interpretation of non-blood data.

The new Referee #3 is strikingly positive, saying they "wholeheartedly recommend" publication. They have two major comments, but as stated in their review, these are primarily for clarification on SCIFER and could be addressed textually.

In our reading, all of this feedback is constructive and we think addressing them would further demonstrate the utility of SCIFER; hence we hope you and your co-authors will be able to respond to them in full.

To guide the scope of the revisions, the editors discuss the referee reports in detail within the team, including with the chief editor, with a view to identifying key priorities that should be addressed in revision and sometimes overruling referee requests that are deemed beyond the scope of the current study. We hope that you will find the prioritized set of referee points to be useful when revising your study. Please do not hesitate to get in touch if you would like to discuss these issues further.

We therefore invite you to revise your manuscript taking into account all reviewer and editor comments. Please highlight all changes in the manuscript text file. At this stage we will need you to upload a copy of the manuscript in MS Word .docx or similar editable format.

*2) If you have not done so already please begin to revise your manuscript so that it conforms to our Article format instructions, available

[here](http://www.nature.com/ng/authors/article_types/index.html).

*3) Include a revised version of any required Reporting Summary: <https://www.nature.com/documents/nr-reporting-summary.pdf>

Link Redacted

Note: This URL links to your confidential home page and associated information about manuscripts you may have submitted, or that you are reviewing for us. If you wish to forward this email to co-authors, please delete the link to your

homepage.

We hope to receive your revised manuscript within four to eight weeks. If you cannot send it within this time, please let us know.

Nature Genetics is committed to improving transparency in authorship. As part of our efforts in this direction, we are now requesting that all authors identified as 'corresponding author' on published papers create and link their Open Researcher and Contributor Identifier (ORCID) with their account on the Manuscript Tracking System (MTS), prior to acceptance. ORCID helps the scientific community achieve unambiguous attribution of all scholarly contributions. You can create and link your ORCID from the home page of the MTS by clicking on 'Modify my Springer Nature account'. For more information please visit please visit www.springernature.com/orcid.

Sincerely,

Michael Fletcher, PhD
Senior Editor, Nature Genetics
ORCID: 0000-0003-1589-7087

Referee expertise: the new Reviewer #3 has, as noted, population genetics expertise.

Reviewers' Comments:

Reviewer #1 (Remarks to the Author):

In their revised manuscript, Körber et al. fully addressed the comments and questions this reviewer raised. In addition, the authors newly applied SCIFER to a published dataset of deep WGS for 185 brain samples from 131 donors, including 44 normal donors and evaluated positive selection in the brain, describing the inferred clonal dynamics of IDH2-mutated clones in a 62-year-old female and the early initiation of clonal selection in the brain. This reviewer raises only a few further questions regarding the new results.

1. The authors have newly developed a two-clone model. Does it improve the detection of clonal selection in the presence of multiple clones compared to the original single-clone model? Is the improvement limited to the cases with deep WGS?

2. In contrast to hematopoietic samples, brain tissue, and most of other tissues, consist of heterogeneous cell components of different cell of origin, including astrocytes, oligodendrocytes, neurons, microglial cells, etc.. It is therefore informative to discuss how this heterogeneity in cell type affects the detection of clonal selection, the interpretation of cell division/mutation rates, and the number of stem cells. Do the inferred values for these variables make any sense? In other words, what cell types do these values refer to? This is a general problem to be answered when SCIFER is applied to solid tissues, which are typically composed of many cell components, immune cells, endothelial cells, blood components, fibroblasts, and epithelial cells. Does this cause any limitation?

3. In particular, multiple mutations were detected in NC7, including those affecting DNMT3A (1.91%), BCORL1 (2.99%), and NRAS (1.84%), in addition to an IDH2 (3.55%) mutation, but only in a small fraction of the sample with small VAFs (ref#17). Thus, it is impossible to determine if these mutations are in the same clone or not, whereas in Fig.7b, SCIFER analyzed the data as if they were found in the same clone. All of these mutations were common in myeloid neoplasms, while DNMT3A mutations were rarely found in glioblastomas. Thus, these mutations may not reside in the neural tissue but some of them might be present in contaminated blood cells.

Reviewer #3 (Remarks to the Author):

Korber et al. develop a method ("SCIFER") that uses relatively deep WGS data to detect the presence of a clonal expansion and infer important parameters governing the clonal evolution in blood including (i) the occurrence time and expansion rate of the positively selected clone (if detected) and (ii) parameters governing the neutral dynamics of the stem cells such as population size (N), symmetric division rate (λ), and mutation rate (μ). They then apply this method to various clinical samples to show that it recapitulates results from more comprehensive (and expensive!) phylogenetic approaches and apply it to new samples to distinguish between samples governed largely by neutral dynamics and those showing clear clonal expansions. They also use it to detect the presence of clonal expansions with no known driver (as previously described by Poon et al. 2021 and Mitchell et. al. 2022).

Overall this is beautiful work and outstanding in its quality and presentation. There are a number of original developments. The application of proper population genetic theory to the WGS data offers a rational approach to detecting clones under

positive selection in WGS and therefore improves significantly on previous approaches such as “PACER” (or the original Williams et al. Nature Genetics 2018 on which PACER is based). The key advance of this method in my view is the ability to infer parameters governing the clonal evolution of blood on an individual basis rather than using ensembles of individuals (as the data scale is provided by the breadth of WGS and does not require thousands of individuals to develop statistically meaningful distributions). The major biological results largely confirm previous estimates for parameters governing stem cell dynamics in blood (Lee-Six 2018, Watson 2020 and Mitchell 2022) and I see this as a good validation of the method. The method as applied to the blood data suggests a possibly interesting result linking selection to alterations in HSC division rates would be novel and very interesting — though as outlined below in more detail I think it is likely this may be explained by other effects (as should be mentioned in the discussion).

I wish to say at the outset that wholeheartedly recommend this be published. I have two comments on the method and its application to the data that I think would be good for the authors to consider mentioning in the revision. The authors have already attempted to address point 1 below in the revision, but not point 2. I suspect both of my points below could be addressed via the discussion to caution the reader against over-interpreting some of the results. Overall I commend the authors for their beautiful work.

The issue of oligoclonal expansion.

The biggest limitation of the method in my opinion is that it does not appear to handle oligoclonal expansion i.e. when there are multiple independent clones that are expanding in the blood (other reviewers pointed this out). As can be seen from almost all of the people in Mitchell 2022, when expansions occur they are almost always quite oligoclonal. As I understood it the method only fits a single dominant clonal expansion (note the the updated version now handles 2 clones). This is quite a limitation because the model presented here will mistake oligoclonal expansion — which lead to too many SSNVs at moderate VAFs but with no clear “jump” in the cumulative distribution — for stronger genetic drift. In samples where there is oligoclonal positive selection the model effectively fits a larger HSC division rate (or smaller N). I think this effect explains some of the reported association between samples with selection and increased HSC division rates. For example if we take the results in Figure 4c (N4 and N5). The authors interpret this “tick up” of the cumulative at lower VAF as being driven by drift and in order to explain why it occurs at VAFs of ~5% they infer a stem cell number (in units of the symmetric division time N/λ) to be about ~10-fold lower than other samples. An alternative explanation for this effect (and a more likely explanation in my opinion) is that this tick up is driven by positive selection on multiple clones, none of which causes a clean “step” in the cumulative. The reason the model prefers to fit a neutral theory to these data (with slightly wonky parameters) rather than positive selection is because the theory for positive selection does not include oligoclonal effects — hence it prefers to infer neutrality with anomalously small parameters for N/λ . I’m not suggesting the authors redo the theory to include this effect (though they now have tried this in the revision) but they should make it clear that in samples where there is oligoclonal expansions (which will be most samples!) that this effect could cause pretty large skews in their estimates for stem cell parameters and therefore those should be treated with caution.

This effect is also present in the pseudo-bulk analysis of the data from Lee-Six 2018 / Mitchell 2022 data in Figure 3. For example, the tree from Lee-Six et al. has a single 5-mer clone, a single 4-mer clone and 3 independent 3-mer clones that arose in adulthood (after 20-years of age). The original paper essentially only fitted to the 2-mers and therefore incorrectly concluded that the evolution in this person was entirely neutral. In fact it is very unlikely that the 5-mer or the 4-mer or 3x 3-mer clones can arise through drift using their own inferred parameters. In these cases it is more likely to be the beginning of an oligo clonal expansion. SCIFER will be quite sensitive to these effects and this is likely part of the reason why the inferences from SCIFER applied to this example infer N/λ to be about a factor of 4 lower than the inferences from the original paper.

This effect is also likely present in Figure 6. For example in U2 and U3 (samples where there is no clear “step” in the cumulative). In effect the model will have an instability where it goes from inferring a selected clone at VAFs close to the limit of detection (~5%) to a scenario where it says things are neutral, but with wonky parameters. This instability will derive from the fact the the underlying two hypotheses (one major selected clone vs entirely neutral) are not accounting for the actually most likely scenario which is oligoclonal positive selection.

Separation of the population size and division rate.

I am sorry to be a stick-in-the-mud but I remain unconvinced that the HSC symmetric division rate and the population size parameters can be robustly separated using the approach described. The arguments given were (i) use of absolute counts of variants and (ii) including the developmental phase in the model. I don’t think either of these can give an estimate of the symmetric HSC division rate in the homeostatic phase which is what is needed to separately infer N and λ . The first argument falls down because in adulthood the only robust estimate for mutation rates in HSCs comes from WGS of single cells from someone of a given age. This means we have very robust estimates for the mutation rate *per year* (~20 per cell per year) but no direct way of estimating the mutation rate per symmetric division. The second argument falls down because while there are good estimates of the mutation rate per symmetric division during development (2-3 per HSC per division) this does not tell one anything about the mutation rate per symmetric division in adulthood.

I am of course biased because I do not believe separating these parameters robustly is possible, but I am open to being proved wrong on this. However I did not find the arguments or data presented to be very compelling and it was hard to determine exactly how the inferences worked. I think the authors should make it clear in the discussion that the ability to

separate N and lambda replies on some assumptions which may or may not hold.

Best,

Jamie Blundell

Version 2:

Decision Letter:

Our ref: NG-A63657R1

3rd Dec 2024

Dear Thomas,

Thank you for submitting your revised manuscript "Detecting and quantifying clonal selection in somatic stem cells" (NG-A63657R1). It has now been seen by the original referees and their comments are below. The reviewers find that the paper has improved in revision, and therefore we'll be happy in principle to publish it in Nature Genetics, pending minor revisions to satisfy the referees' final requests and to comply with our editorial and formatting guidelines.

Sincerely,

Michael Fletcher, PhD
Senior Editor, Nature Genetics
ORCID: 0000-0003-1589-7087

Reviewer #1 (Remarks to the Author):

In their revised manuscript and rebuttal, the authors fully addressed the questions this reviewer raised, discussing the limitation of SCIFER in inferring dynamics of clones. I have no further concerns.

Reviewer #3 (Remarks to the Author):

The authors have addressed all my concerns. I congratulate them on their high quality science.

!

" H " "O

\$

&% % % ' % % B * , % \$
5# ,0 \$ 5" % % + \$+ + &
& % % ,

- \$ / % % % & \$. + +%" , -
+ % + % % 3 4 \$+ & % ,)
+ %

W %\$ 5! %1 0 2 : ' + 6+ % + ,
\$ B '+ ,
\$+ 5# & % \$ % % % % 1 0 2 + , % * %

0 \$ + & & %
1 0 2 6 % % ' + + ,

\$ % & % \$ + %
/ + % % % % % % U
, % &

) % % & % % % <1) , *
+ ,

) / % + % + & % % ,

) %

= 80 > ? \$ '+% \$ % ,0
& \$%

= 80 % + %
@ A \$ + , B B B C% B * , .D @D,
%

#80 % / 1 % % , , B B ' '
0 + + 7 \$ % 8 + &
, &

+ @ A \$, , B ' B ' B ' %D
, @D
&+ + %

&

+ \$ 9 & % ,0% ' + \$ % % & %
%

) ,3) 41 " !J 2 + % 8 & % > 2 ?7 + %
7 + 7 8 + % +) 41 / + %
% + 7 '1 * " !J\$) " " < " " " 8 0 ,
+ % Y + & 0 & & % 1 % + *
+ % 7 + 8 + + ,0
0 % 0 & ! + \$ + , ,0 + % , + 0 +
% + & ,H 0 +
* ,
+ % * + 7 % / 8 + , ,
< " " "\$ * 7 % Y % 113 /
+ % & + Y Y ,0 & * &
% 1 7 380 , * + %
+ 7 1 , * + &
O 70 318 ZIX % & ? + 7 % 7 & %
3B + 8 + + ZI ' & + % 7 \$ % & % 8
* ? , + % % 3B + , G & % &
> ? , + % % (8 8+ % & %
* \$ 7 '1 * , 8 I' \$ O % % !JB< " " # ,
,0 % % & % I' O #* # %
,0 & % + + %
* ,1 02 + / 3B + + + O
,
& % E, * 9" 9#7 > ?
80 ZIX 8 % + % \$+ & % 8 , + % %
& % ,
1 U ,
0 % + & ' ' + 0 1 % 78 + U
% 78 + ,0 G & % 3 + ,
% 1 + % + 1 3) 41
% 8+ % , % + % = % =Z"
% + % , 7' # 1 + 8
0 + + 0 + + % + \$+ 0 +
* % , 0 & ,0 & & + % + %
+ % 5

42\$&\$- * '#\$%&\$#+' /28\$/#-'#8* '<'+#"\$8* 4 "+ /#%&, "# &# , "+?#%-<#%#&\$"# *0#Cc76#C*88"1\$-, "8?9#\$/#/"#
 *) /"+, &\$- * '/# &+"#1* '/-/'\$# .-\$%# 8* '<#\$"8* 4 "+ /#) "-<# "+4-//-, "#+" 12+" 4 "'\$# 0*+# Cc9#
 &+<2&)8?#) ?# &88* .- '<#Cc# cBC/#\$*#2'3"<+*#" '%&'1"3#1"88#3-, -/-'6# ! "#%&, "# &33"3#\$/#/"#
 -45*+\$&'\$#- /-<#%#/#-'\$*#%#"#` -/12//-* '#=p14, lines 481-492>6#

#

Figure R1. Telomere lengths (bp) in CD34+ HSPCs and in PB granulocytes as inferred with Telomerehunter for the 14 cases where WGS was available for both populations.

#7&8 '9' (9, 1&, 4& : *5&9-0'1+&) %(3;0'9, 1&<9(3&=%00&+9>9-9, 17&

D' #&33-\$- *'9# . "#&\$" 45\$ "3\$*#-'0"+ /\$ " 4#1"88#3-, -/-' '#+&\$" /# . -\$%#0821\$2&\$- '<# 4 "\$%?8&\$- *' #18*1 (#
 &' &8?/-/8# K%-/# &55+* &1%# 12&' \$-0-"/# 3+0\$# &'3# /"#8"1\$- *' # -' # %* 4 " * /\$&\$-1# \$-//2" /# 0+* 4# \$%"#
 3-/\$+)2\$- *' #*0#4 "\$%?8&\$- *' #8", "8/#&\$#C5J#/\$- /#%&\$#/\$*1%&/\$-1&88?#/. -\$1%#) "\$. " " #2' 4 "\$%?8&\$"3#
 &'3# 4 "\$%?8&\$"3# /\$&\$" /6# c * . , "+9# . "#0*2'3# \$%&\$#0821\$2&\$- '<# 4 "\$%?8&\$- *' #18*1 (/#1&' '*\$#) "#
 &558- "3# \$*#-'0"+#3-, -/-' '#+&\$" /#*0#%" 4&\$*5*-" \$-1/\$" 4#1"88/#-' #*2+#/ &458" /#0+* 4# ' *+4&8#*+#18* ' &8#
 % " 4&\$*5*-" /-/#0*+#\$ "1%' -1&8#+' &/' /6#

E821\$2&\$- '<#4 "\$%?8&\$- *' #18*1 (/# . "+*#*+<-&88?#2/"3# \$*#12&' \$-0?# ' *+4&8#&'3#5+"W 4&8-<' &' \$#/" 4#
 1"88#3?' &4-1/# #/ 4&88#1"88#1 * 45&+\$4 " '\$/#. -\$%#%* 4 " * /\$&\$-1# \$2+ * , "+#- *+"# "1" '\$# ;\$' /-'9#
 FSQEiXj9#0*12/" /#* '# " *58&/\$-1#";5& /-' /#&'3#-/# *\$#&558-1&)8" # \$*#%* 4 " * /\$&\$-1# \$-//2" /#^9>6#
 7?#1' *\$+&/9#%24&' #%" 4&\$*5*-" /-/#-/#/2/\$&- '3#)?#&+<"#/\$" 4#1"88#5* *89# . %"+#1%&'<"/#-#
 0C5J#4 "\$%?8&\$- *' #/\$&\$2/#32" # \$*#3+0\$#&+"#/2) \$8" &'3#3-00-128\$# \$*#3 "\$1\$#&<#- /\$#%#"#) &1(<+*2'3#
 *0#/" 12" '1- '<# ' *-' /#&'3#-' \$'+-3-, -32&8#%" \$"+* <" " "-?#=#/" "#E-<6#e&9) #-' #+"068>6# _ *+" * , "+9#2/-'<#
 &<" \$W) &/'3#/-428&\$- *' /#*0#0C5J/#4 "\$%?8&\$- *' #/\$&\$2/#- '#&8&+<"#-//2"#=PM9MMM#/\$" 4#1"88/#^8>9#
 . "#0*2'3# \$%&\$#) *\$%#&' #&11"8"+&\$"3#/\$" 4#1"88#3-, -/-' '#+&\$" #&'3#/2)18* ' &8#/"8"1\$- *' #. -88#-'1+"&/'#
 \$%"#, &+&'1" #*0#5%"#4 "\$%?8&\$- *' #/\$&\$2/#�C5J/6#c " '1"9#4 "&/2+ '<#4 "\$%?8&\$- *' #8", "8/#&\$#0C5J/#
 -' #)8* *3#1&' '*\$#2' &4) -<2*2/8?#3-/\$- '<2-/#%#) "\$. " " #&11"8"+&\$"3#3+0\$#*+#/2)18* ' &8#/"8"1\$- *'6#
 K%"+"0*+"9# . "#. *283#/2<<"/\$#%&\$ 5+* , -3- '<#*+\$%* <' * &8#";5"+4 " '\$&8#", -3"1"1#1* '/-/'\$# . -\$%#
 &' #&11"8"+&\$"3#3-, -/-' '#+&\$" #-' #/ &458" /# . -\$%#&#/"8"1\$"3#18* " #-' /#) "?* '3# \$%" /1*5" #*0# \$%#-/#
 4&'2/1+5\$6##

J-, "#. "#3*#* *\$%&, "#*+\$%<* '&#", -3" '1"#*\$/255*+\$#-'1+&/"3#3-,-/*'+&\$"9#. "%&, "#' * . #
1&+"0288?#3-12// "#3*2+0-'3- </#* ' # cBC#3-,-/*'+&\$"##* '#p11, lines 373-3816#_ *+* , "+9#. "#
& , "#12&8-0"3#*2+*) / + , &\$- * #%&\$#cBC#3-,-/*'+&\$"9# 45%&/-Y- <#%&\$# ' 3"5" '3" '\$, &8-3&\$- * '#
-/3"/+&) 8 "# = ` -/12//-* 'H p14, line 481-492>6##

! "#&8/*#/\$+//#%&\$#&8-4-\$&\$-* '#*0#*2+# . *+(-/#%&\$#*2+#-'0"+"3#1"88#3-,-/*'+&\$"/#&+"#-\$-4"W
& , "+&<"36#d+*+#&' &8?/-/*'0#S: E#-'#/" +&8#/&458"/#2<< /\$%&\$#3-00+" '\$#<" '\$-1#3+-, "+/#4&?#
-45&+\$#3-/\$-1\$#(-' "\$-1/#*0#1"88#3-,-/*'6# [&4 "8?9#%#"DNMT3A#18*'" /#4&?#/#* . #3* . '#. -\$%#-\$-4"9#
TET2#18*'" /#"; 5&'3#&\$#*'/&\$'\$#&\$"##* , "+#-\$-4"9#&'3#0-'&88?9#SRSF2#&'3#JAK2#18*'" /#&+/"#
&8\$"--'#8-0"#)2\$#"; 5&'3#0&/"\$+6#D\$# . *283#)"# , &82&)"8"#*\$#&558?#BCDEG#\$*#/" +&8#8* '<-\$23-'&8#
&458"/#*\$#\$"/\$%* . #/\$"4#1"88#3?'&4-1/#&'3#5&+&4"\$"/#1%&'<"# . -\$%#-\$-4"## . -\$%#3-00+" '\$#
3+-, "+/9#-'1823- '<#12++" '\$8?#2' ('* . '#3+-, "+/#0*+# . %-1%#%#" +/#-'#*/21%#-'0*+4&\$-*'6#K%/-/#/' '\$#
2\$#'#-'#%#"` -/12//-* '=p14, lines 473-480)6

_*+"#<"'"+&88?9#the key point the Reviewer is raising is what novel observations does
our manuscript make# ! "#%&' (#\$%"#G" , - . "+#0*+#"45%&/-Y- <#%&/# ("?#5*- '#&'3#2<< /\$#
%#"+" , -/"3#4&'2/1+5\$#1* '\$&' /#) *\$%#* , "#84"\$%*3*8* <-1&8&'3#) -*8* <-1&8&3 , &'1"/# , &82&)"8"#
\$*#* '\$*#* '8?#%#"Ccb%"4&\$*5* . -/ /#0-"83#)2\$#1+* / /#*\$%"+#0-"83/#/23?-'<#/'4&\$-1#4* /&-1/4#-'#
\$%"+#\$/2"/6#7"8 . 9# . #/'"\$#*2\$#%#"#&+<24" '\$/#&'3#'" . #3&\$#%&\$#/255*+\$#%#/'1* '\$' '\$-*'6#

1. SCIFER provides a substantial methodological advance.

BCDEFG#&88* . /H#

=>#1* 45&+&\$, "8?#1* /\$W"00"1\$- , "#-'0+" '1"#*0#18* '&8#3?'&4-1/#/'/*4&\$-1#\$/2"/#&'3#%" '1"#
1&'#)"#-458"4" '\$"3&\$#8&+<"+#/1&8"6#
#/\$23?#*0#\$/2"/# . %"+#1*8* '?#/"12" '1- <#-/#'*\$#0"&/-)8"6#

A.# To further understand the methodological utility of SCIFER we show how it performs at
different WG sequencing depths6##

! "#+ "W/"12" '1"3#%#"#` [: *#0#%"4&\$*5*-"\$-1#/\$"4#&'3#5+* <" '\$*+#1"88/#=cBdC/>#0+*4#PL#*2\$#
0#NN#-'3- , -32&8/#=&88#/&458"/# . %"+#/#/200-1" '\$#cBdC/# . "+#/#88#& , &-8&)"8">#\$#* , "+&<"#*0#
NOM;6#

D'#\$%#"+" , -/"3#4&'2/1+5\$9# . "#' * . #0+/\$#&'&8?Y"#18* '&8#/"8"1\$-* '#-'#%#"#PN#-'3- , -32&8/#=V#4&8"/9#
L#0"4&8"/9#&'<- '<#0+*4#eN#\$*#ZL#?'&+/#*0#&<"># . -\$%# ('* . '#Cc#3+-, "+#42&\$&\$-*/#&#LM;#&'3#
NOM;#=#new Fig 4a-e and new extended Data Fig.7a-c>#BCDEFG#1* '0+4"3#%&\$#&88#*0#%#" /#"
1&/"/# . "+#"/2)8"1\$#*#/"8"1\$-* '#=#new Fig 4c,d>6#_ *+* , "+#8&8#)2\$#*'"#*0#%#"# ('* . '#Cc#3+-, "+/#
&8#3#S: E/#'&#<+"4" '\$# . -\$%#&#/"8"1\$"3#18*'"#3" '\$-0"3#)?#BCDEFG=#new Fig. 4f>6#d3" '\$-0-1&\$-* '#
0#18 '&8#/"8"1\$-* '#&'3#5&+&4"\$"/#-'0+" '1"#&#NOM;#/"12" '1- <#3"5\$%# . "+#"1* ' /-/' '\$# . -\$%#
%#"#LM;#"/28#/#=#new Fig. 4c,d,g,h>6#c* . , "+#9#&\$#NOM;# . "#-3" '\$-0"3#/"1* '39# /4&88"/#/"8"1\$"3#
18*'" /#/'#/" , "#1&/"9#/-;#*0# . %-1%# . "+#"3+-, "#')?#&/#*0#?"\$2' ('* . '#3+-, "+/#=#new Fig. 4b,e,#
new Extended Data Fig.7b>6#c" '1"9#NOM;# ! JB#2'1* , "+/#5" , &/- , "#18* '&8#/"8"1\$-* '#0*+
('* . '#&'3#2' ('* . '#3+-, "+/#&\$#%#<"+#/"*82\$-*'6#The results are set on p8-10, lines 275-
3476##

! "#' ; \$/23-"3#%#"#"'#1&/"/# . -\$%*2\$# ('* . '#Cc#3+-, "+#42&\$&\$-*/#&#) *\$%#LM;#&'3#NOM;#=#new#
Fig. 5a^#) #4&8"/#&'3#e#0"4&8"/9#&'<- '<#0+*4#VM#*\$#0e#?'&+/#*0#&<">6#_*/\$#/&458"/#%&3#&+*2'3#
TMWPMM#BB [S/# . -\$%#S: E#hNU# . %-8/\$#%#+"#"/&458"/#/\$**3#*2\$# . -\$%#1]MM#BB [S/#=&#NOM;>6#
:558?-'<#BCDEFG#\$*#%#" /#1&/"/9# . "#0*2'39#&\$#LM;# ! JB#=#new Fig. 5b>9#18"&+" , -3" '1"#0*+
18* '&8#/"8"1\$-* '#-'#%#+"#1&/"/#=#NWXX#ZWXX9#PeWXX>9#&'3# . "&(# , -3" '1"#0*+18* '&8#/"8"1\$-* '#-'#
*'"#1&/#=#VW [>6#K%#"+"4&- ' -'<#/-;#1&/"/# . "+#"18&/-/"3#&/# "2\$&88?#" , *8 , -'<#=#new Fig. 5b,
new Extended Data Fig. 8a>6#!%"#5+*)-'<#&\$#NOM;9# . "#+28"3#*2\$#/"8"1\$-* '#-'#VW [9#)2\$#
1* '0+4"3#/"8"1\$-* '#-'#ZWXX#&'3#PeWXX#0*+ . %-1%#NOM;#3&\$#& . &/#& , &-8&)"8"#=#new Fig. 5c>6#
_ *+* , "+9# . "#-3" '\$-0"3#%#+"#&33-\$-* '&8#1&/"/# . -\$%#/"8"1\$-* '#=#PPWXX9#PVWXX9#]WX>9#&/# . "88#&/#
/"1* '3&+?#2)18*'" /#-'#ZWXX9#PPWXX#&'3#PeWXX#=#new Fig. 5c,d, new Extended Data Fig.
8b>6#c" '1"9#* '8?#0*2+*2\$#*0#%#"#"\$'1&/"/# . -\$%*2\$# ('* . '#Cc#3+-, "+# , *8 , "3" '2\$+&88?#=#25\$*#&#
3"\$1\$-* '#%#+" /%*83#*0#NU#S: E>9#2<< /\$- <#5" , &/- , "#/"8"1\$-* '#0*+9#&/#*0#?"\$2' ('* . '#9#3+-, "+/6#

These results are set out on **p10-11, lines 349-366**. This new data highlights the importance of stating the VAF resolution used to distinguish selection from drift.

Finally, and interestingly, the ranges of inferred HSC parameters (ratio of HSC number to HSC division rate, N/λ ; HSC numbers, the number of new SSNVs acquired per HSC division, HSC division rate in both neutrally evolving samples (**new Fig 5e,f**; the results are described on **p11, lines 367-373**) and in samples where clones had been selected (**new Fig. 4g,h and new Fig. 5g,h**; the results are described in **p10, lines 330-341** and **p11, lines 372-373**) were similar when data from 90x and 270x sequencing depth was used.

B. Extending SCIFER to a two-clone model.

In the original version, SCIFER accounted for a single leading selected clone, regardless of whether, or not, there were smaller selected clones present also. Using data from 270x, we noticed that the model fits for a one-clone model were not optimal for **3 out of 18** samples (samples 6-AU, 12-AT, 22-TU). As assumed by the reviewer, the measured data indeed showed two shoulders, raising the question whether these deviations could be caused by a second selected clone.

This observation prompted us to develop an extension of SCIFER to quantitate multiple selected clones. We began by considering selection of two competing clones, born at times t_{s1} and t_{s2} , and conferring a selective advantage of, respectively, s_1 and s_2 (**new Fig. 2d,e**) developing either via branched (**Extended Data Fig. 2a**) or linear evolution (**Extended Data Fig. 2b**) as orthogonal topologies.

To this end, we generalized the model to one normal and $C - 1$ selected clones with generally distinct birth dates. As before, we assume that all clones compete for a limited “space” with carrying capacity N_{ss} . Denoting the competition coefficient between clone c and other clones in the tissue with ρ_c , we model the expected number of cells in clone c , n_c , with a system of ordinary differential equations,

$$\frac{dn_c}{dt} = \lambda_{ss} n_c \left(1 - \sum_{1 \leq j \leq C} \rho_{jc} \frac{n_j}{N_{ss}} \right),$$

with boundary conditions

$$n_c(\tau_c) = 1,$$

$$n_c(t < \tau_c) = 0,$$

$$n_{v_c}(\tau_c) = n_{v_c}(\tau_c - dt) - 1,$$

where the variable v_c reports the identity of the mother clone of clone c and τ_c is the clone’s birth date. The competition coefficients are parameterized such that selected clones experience a reduction of cellular loss by a factor r_c , $0 < r_c < 1$ while maintaining $\sum_j n_j = N_{ss}$ at all times. Together, this yields the competition matrix

$$\rho = \begin{pmatrix} 1 & 2 - r_2 & 2 - r_3 & \dots \\ r_2 & 1 & 2 - r_3/r_2 & \dots \\ r_3 & r_3/r_2 & 1 & \dots \\ \dots & \dots & \dots & \dots \end{pmatrix}.$$

To model the site frequency spectrum (to be compared with the VAF histogram), $S_i(t)$, the expected number of variants that are found in a clone of size i at the time point of measurement, t , we consider the contributions from variants acquired in normal cells and those acquired in the selected clones, as explained in the new Methods section, *Modeling multiple selected clones in a homeostatic tissue* (**p53-56, lines 1306-1392**).

Analyzing this model with multiple selected clones, we found that selection of a second clone indeed manifests itself as an additional shoulder in the cumulative VAF distribution (**new Extended Data Fig. 2c-d**). Practical identifiability from our WGS data was indeed limited to a second clone, and hence we set $C = 3$.

When applying this two-clone version of SCIFER to all 16 270x WGS cases with clonal selection (5-DU, 6-AU, 7-T, 8-UU, 9-DU, 10-D, 11-UU, 12-AT, 13-U, 14-U, 15-N, 16-UU, 17-TT, 18-DU, 20-UT, 22-TU), we identified ten cases with evidence for a second selected clone (**new Fig. 4e and new Fig. 5c**). In two cases, the second clone was associated with a mutation in *TET2*, whilst we found no known driver mutation in the remaining cases. These data further indicate that selection for unknown drivers, occurs frequently in human hematopoiesis, especially at low VAF. Note that apart from a higher resolution of clonal selection, the inference results obtained with 90x and 270x WGS data were consistent, suggesting that stem cell parameters are accurately inferred from both 90x and 270x WGS data (**new Fig. 4g,h, new Fig. 5e-h**).

We also applied this model to the benchmarking case KX004 and found a second selected clone in agreement with the phylogenetic tree (**Fig. 3g, new Extended Data Fig. 3j**). Importantly, the inferred parameters of the stem cell dynamics agreed well between the original leading-clone version and the two-clone version of SCIFER (**new Extended Data Fig. 3g, new Extended Data Fig. 7g**), suggesting that stem cell dynamics can be reliably inferred with the leading-clone model as well.

We describe our results in the Results Section “**SCIFER identifies sequential selected clones**” (p6 lines 187-200) and in Sections “**Benchmarking SCIFER with phylogenetic trees of human hematopoiesis**” (p8, lines 255-256), “**SCIFER detects selection and uncovers clonal complexity in human bone marrow**” (p9, lines 311-317 and p10, 324-329) and “**Clonal selection without known CH drivers**” (p10-11, lines 371-384). We discuss the new data in the Discussion (p14-15, lines 493-502). This important extension to the SCIFER model now provides a foundation to quantitatively study clonal competition, which, as clones evolve and progressively disturb tissue function, may ultimately result in cancer in some cases.

2. SCIFER detects pervasive selection in human brain – exemplar of analysis of a solid tissue without single-cell derived colony WGS.

In our initial submission we stated that a potential use for SCIFER would be to distinguish selection from drift in solid tissues where single cell derived colonies may be hard to grow. To add biological novelty to the submission we now provide evidence that this is indeed the case by using published deep WGS data from Bae et al., 2022¹⁰ on human brain tissue.

We used deep WGS data from 185 human brain samples from 131 individuals aged 4-95 years¹⁰. This cohort comprised samples from 44 normal (neuro-typical) individuals and 87 individuals with neurological disorders (autism spectrum disorder, schizophrenia, Tourette syndrome), taken from different brain regions (cortex, striatum, hippocampus) (**new Extended data Fig. 10a**). Two individuals harbored somatic mutations known as early mutations of human gliomas. In both cases, SCIFER identified selected clones without knowledge of the drivers. Clone sizes agreed with the measured VAFs of the driver mutations (Trisomy 7/Monosomy 10 in **new Fig. 7a**, characteristic of IDH-wildtype glioblastoma, and *IDH2* p.140R/Q in **new Fig. 7b**, an *IDH2* hotspot mutation, which has been described at low incidence in IDH-mutant astrocytoma^{11,12}). Hence, as with the CH cases with known drivers, SCIFER detected putative pre-malignant clones in the human brain.

Of note, in one of the two individuals, an expanded clone was detected in neuronal fractions, while full-blown tumors typically have a glial phenotype; however, both neurons and glia arise from common neural stem cells. Remarkably, in this individual, we found both an ancestral premalignant clone in cortical oligodendrocytes (with *IDH2* and *BCORL1* mutations, **new Fig. 7b**, left panel) and a subclone with further driver mutations (*NRAS G12D* recurrent mutation and *DNMT3A* splice-donor variant) in striatal interneurons (**new Fig. 7b**, right panel). Consistent with the driver status, SCIFER inferred a younger age and a higher growth rate of the striatal subclone. In summary, SCIFER successfully detected and quantified a putative pre-malignant selection in the human brain. Taken together, the additional drivers, the younger

age and the faster growth rate suggest that the striatal clone had evolved from the original IDH2 mutant clone.

Applying SCIFER to the remaining 129 samples, none of which harbored known driver mutations, SCIFER identified clonal selection in 24% of the samples (**new Fig. 7c,d**). The incidence of selected clones increased with age (**new Fig. 7c-e**). In most cases, clonal selection dated back to childhood or early adolescence (**new Fig. 7h**), suggesting a critical phase for clonal selection in early life, where the brain is still under development. Interestingly, clonal selection was not associated with neurological phenotype or brain region (**new Extended Data Fig. 10**). Moreover, we inferred similar stem cell dynamics across different brain regions (cortex, hippocampus and striatum; **new Fig. 7f-g**), suggesting uniform dynamics of neuro- and gliogenesis across the human brain. SCIFER inferred $\sim 10^5$ stem cells, both in the subventricular zone (feeding cortex and striatum) and in the hippocampus, that divide on average 2-3 times per year (**new Fig. 7f, new Extended Data Fig. 10c-d**). Interestingly, the division of neutral stem cells appears to decrease in the first 25 years of life (**new Fig. 7g**). Hence, the focused age incidence of clonal selection in the brain appears to be fundamentally different from the constant rate in hematopoiesis

We now describe this analysis in the Results Section **p12-13, lines 396-446**. We have amended the Abstract **p1 lines 23-28** and in the Discussion **p15, lines 506-523**.

3. Quantitative insight into acquisition of selected clones

Finally, we would like to emphasize that the comprehensive inference of HSC parameters by SCIFER, on 22 individuals, encompassing neutrally evolving and selected clones with both known and unknown drivers, provides new insight into the age-incidence of selection events in hematopoiesis. We show that the acquisition of selected clones in blood, whether they arise by known or unknown drivers, occurs at an approximately constant rate. The near doubling of the number of selected clones we now identified (28 compared to 15 in the first version of the manuscript) substantiated our finding that selected clones begin to originate already in infancy and continue to do so with approximately constant rate throughout human life (**new Fig. 6d**).

2) The mathematical formulations of the model $S_i(t; \mu; \delta, N, \lambda\mu N, r)$ and (8) and (14)-(18), are quite complex and difficult to comprehend. Thus, for better understanding of the model, several instances of the model need to be presented in an Extended Figure, by showing $S_i(t)$ for a wider range of parameters (N, μ, λ) than presented in Fig. 1e-l, in particular for a wider variety of stem cell numbers (N_{ss}) to clarify a big discontinuity between $N=50,000$ and $N=500,000$. For clonal selection, the instance should be presented for the terminal clone size of 5, 10, 15, 20, 25 and 30, and for a variety of N_{ss} , assuming the SSNVs are measured 10, 20, 40, and 50 years after the acquisition of a driver mutation.

We thank the reviewer for their constructive feedback. We have followed their suggestion by adding the requested simulations to the **new Extended Data Fig. 1 (a-c)**. These simulations now give a more detailed account of how the VAF distributions depend on the stem cell parameters. We refer to these changes on **p4, lines 123-124**.

3) As shown previously (for example, Moran-Cusio et al., Cancer Cell, 2011), TET2 deficiency resulted in an expanded stem cell pool. Thus, N_{ss} in CH(+) cases might not be constant but gradually increase over years. Does this model can lead to a different model and estimation of critical parameters?

The reviewer raises the interesting point that the total stem cell number may increase as a result of acquisition of the *TET2* mutation and asks how this might affect the inference of HSC dynamics with SCIFER. To address this question, we added a model variant to SCIFER that allows it to infer parameters in a situation where the selected clone expands without displacing normal stem cells, leading to an overall expansion of the stem cell pool.

Specifically, we assume that the number of normal stem cells, n_1 , remains constant at $n_1(t) = N_{ss}$ for all times, whereas the number of mutant stem cells, n_2 , expands according to $n_2(t) =$

$e^{\lambda_{ss}(1-r)(t-t_s)}$, where t_s is the time point at which the driver mutation was acquired, λ_{ss} is the stem cell division rate and $r, 0 \leq r \leq 1$ is the reduction in cell loss among mutant cells. Hence, the total number of stem cells now amounts to $N_{ss} + e^{\lambda_{ss}(1-r)(t-t_s)}$, if $t > t_s$, and else to N_{ss} .

As with the original version of SCIFER, we consider three contributions to the site frequency spectrum of somatic variants, where some variants have occurred prior to t_s and either do or do not end up in the selected clone, whereas other variants have occurred after t_s , either in the selected clone or in normal cells. As opposed to the original version of SCIFER, where normal stem cells are replaced by mutant stem cells, variants in the normal stem cell population now drift according to a critical, instead of a sub-critical, birth-death process (i.e., they remain unaffected by the mutant clone).

We fit this modified version of SCIFER to the measured variant allele frequency spectrum in 7-T, 17-TT and 22-TU, the three cases where clonal selection was driven by a mutation in *TET2* and found that the inferred model parameters change only marginally. Intuitively, this is understandable, as for most of the time from the origin of the selected clone to detection, the selected clone is of negligible size for competing with normal HSCs. Hence, whether clonal competition is present, or absent, in the HSC population (which, to our knowledge, is not known) does not exert a strong effect on the quantitation of clonal dynamics by SCIFER. We show these results in the **new Extended Data Fig. 7d-f** and report them on **p10, lines 333-336**. Moreover, we describe the mathematical details of the model extension in the Methods Section “Modeling a single selected clone without size compensation” **p52-53, lines 1280-1305**.

4) There has been a debate regarding the dynamics of hematopoietic stem cells (HSC) vs. differentiated progenitors based on lineage-tracing study in the mouse model. Some studies suggest a relatively small efflux from HSCs to progenitors where the mature progenitors differentiated from HSC replenish hematopoiesis for long time, which could be replaced in a short period by the progenitors in the next generation (Sun et al., Nature, 2014). For example, the analysis of the life history of myeloproliferative neoplasms demonstrated that a large clade in MPN phylogenies could be easily disappear (Fig. 2 in #33). Alternatively, those progenitors long lived without achieving equilibrium within the mouse lifetime (Busch et al., Nature, 2015; Pei et al., Nature, 2017). In these cases, the measured SSNVs does not correctly reflect those in the stem cell compartment, where N_{ss} will be inflated because of the presence of differentiated progenitors.

- How could these different underlying stem cell dynamics influence the interpretation of the current results? - Could the authors analyze samples from different time points from the same individuals?

The reviewer asks how cellular dynamics during differentiation alters variant allele frequencies, and, hence, to what extent measured variant allele frequencies reflect those in stem cells rather than progenitor cells.

We agree that this is an important question that is crucial to the interpretation of our results. We also fully agree with the reviewer that measuring samples from different time points from the same individuals could yield valuable information address. Unfortunately, we do not have access to repeated bone marrow or blood samples from our cohort. Therefore, we addressed the reviewer’s question by:

- i) performing stochastic simulations of genetic drift in HSC and progenitor cells compartments.
- ii) comparing inference results obtained with HSPCs sorted from the bone marrow (the main source of original hematopoiesis data in the manuscript) with SCIFER analysis of mature blood cells.

We describe the results in turn:

Stochastic simulations of genetic drift in stem and progenitor cells.

The reviewer emphasizes that quantitation of lineage tracing studies of murine HSCs yielded insight into the underlying differentiation and self-renewal dynamics^{13,14}. The key insight is that HSCs differentiate rarely. In contrast, the progenitors immediately downstream of HSCs, **multipotent progenitors** (MPPs, termed ST-HSCs in¹³) differentiate about 5 times more rapidly than HSCs whilst also undergoing self-renewing divisions. We used these parameter inferences from our murine studies as a guide for simulating HSC and MPP dynamics across human life.

Specifically, we used Gillespie's algorithm to simulate 1,000 HSCs that differentiate into MPPs once a year, while MPPs had a ~5-fold increased differentiation rate and 2.5-fold larger number than HSCs. Drift effects will be stronger with these smaller cell numbers, compared to the 10⁵ or so HSCs in humans, and hence provide an upper bound on expected drift effects in humans. We let progenitor cells divide with rate 4.6/year and differentiate with rate 5/year (**new Supplementary Fig. 1a**). For simplicity, we assumed that on average one neutral SNV was acquired per cell between divisions. In each simulated division, we stored the number of SNVs specific to the daughter cells and stored the mother-daughter relationships, thus encoding the phylogenetic tree of the simulated population (**new Supplementary Fig. 1b**). In total, we simulated ten independent replicates, reporting the results after developmental expansion (0 years) and after 25, 50 and 75 years. From the simulated phylogenies, we computed variant allele frequencies (VAFs) across all cells, HSCs only or MPPs only (**new Supplementary Fig. 1c**). We then assessed the difference between the cumulative SSNV count when comparing MPPs to HSCs, or all stem and progenitor cells to HSCs. At a VAF resolution of 1%, corresponding to the practical resolution limit of 270x WGS data, the average difference between the VAF distributions was marginal, reaching between 5 and 10% at 75 years (**new Supplementary Fig. 1d**). Hence, our simulations suggest that the measured VAF distribution in MPPs is predominantly shaped by the HSC dynamics, whereas genetic drift in long-lived MPP clones has little impact on these data. We describe these results in the **new Supplementary Note 1**.

Moreover, we describe the stochastic simulations for both stem and progenitor cells in Methods (p61) by amending the Section in Methods "**Simulation of phylogenetic trees and in-silico evaluation of model performance**".

Comparing inference results obtained with HSPCs sorted from the bone marrow (the main source of original hematopoiesis data in the manuscript) with SCIFER analysis of mature blood cells.

To assess the impact of cell differentiation dynamics of mature lineages on the inference results with SCIFER, we compared inferences from HSPCs with bone marrow mononuclear cells (MNCs), T cell-depleted mononuclear cells (MNC(-)T)s and peripheral blood (PB) granulocytes, all profiled with 90x WGS.

In the original version of our manuscript, we showed SCIFER results obtained with these cell populations in 5 individuals (former Supplementary Fig. 2). To compare these populations across a larger cohort, we now performed the same analysis for 9 additional individuals with sufficient input material (shown in **new Supplementary Fig. 3**). We found that clonal selection was erroneously identified in unsorted BM MNCs in two individuals (**new Supplementary Fig. 3a**; false positives for 1-N and 4-N). Using T-cell depleted MNCs rectified this; hence clonal selection in T cells (and putatively also in B cells), potentially driven by antigen recognition through TCRs (and possibly BCRs), might perturb the VAF distribution inherited from HSCs. By contrast, we did not erroneously infer clonal selection in PB granulocytes (no false positives), suggesting that clonal selection is detected with a higher specificity in PB granulocytes. Conversely, however, when running SCIFER on PB granulocytes, we missed clonal selection in three cases for which it was detected in HSPCs (**new Supplementary Fig. 3b**; false negatives for 5-DU and 19-D; **new Supplementary Fig. 3c**; false negative for 2-U). In summary, we recommend the analysis of HSPCs to obtain confident results on selection in

stem cells. By contrast, we caution against the use of BM MNCs to probe selection in stem cells, as they may lead to false positive results. Finally, we did not observe false positive results of clonal selection in PB granulocytes, although their analysis may lead to a false negative result (lower sensitivity). We describe these results in the **new Supplementary Note 3**.

5) The authors claim increased time-averaged HSC division rate for all stem cells in CH-positive patients. How the authors confirmed that the division rate of both mutated and unmutated stem cells increased compared with those cells from age-matched individuals?

If we understand the reviewer correctly, they ask how have we confirm that the division rate of both mutant and unmutated HSCs is increased in individuals where selection is operative as compared to age-matched individuals where HSCs drift neutrally. In the population-genetics model underlying SCIFER, the division rate of unmutated HSCs and the selective advantage associated with a CH clone are separately inferred from the cumulative VAF distribution of all somatic variants. This is possible because the VAF histogram contains both information on the drift of variants acquired in normal HSCs and on the selection of variants present in the selected clone. In particular, the selective advantage of the CH clone defines the position of the subclonal shoulder in the cumulative VAF distribution, while the stem cell division rate (together with the size of the stem cell pool) determines the shape of the distribution of somatic variants overall. We set out how SCIFER infers these parameters in detail in Supplementary Note 2 “Parameter dependence of the site frequency spectrum”. To aid intuition, we now show fine-grained simulations that show how the cumulative 1/VAF distribution changes with time in dependence of the number of stem cells and the selective advantage associated with the selected clone (see **new Extended Data Fig. 1a-c**; note that, mathematically, varying the age while leaving the division rate constant yields the same results as varying the division rate while leaving the age constant).

Nonetheless, the population-genetics model underlying SCIFER implements the selective advantage of CH clones as a reduction in cell differentiation rather than an increase in cell division (Model A). This is by choice – an alternative model would implement selection as an increase in the division rate of mutant cells, rather than a decrease in the loss rate (Model B). Hence, a key aspect of the reviewer’s question is how confident we can be that the higher cell division rate, λ_{ss} , in many individuals with CH compared to neutrally evolving cases concerns all stem cells and not only the mutant stem cells. To address this question, we compared the parameter inferences with Model A, where cells in the selected clone divide with rate $\lambda^A = \lambda_{ss}$ and are lost with rate $\delta^A = r\lambda_{ss}$, $r, 0 \leq r \leq 1$, to the inferences with the alternative Model (B), where cells in the selected clone divide with rate $\lambda^B = \lambda_{ss}(2 - r)$ and are lost with rate $\delta^B = \lambda_{ss}$ (note that both scenarios yield a net expansion rate of $\lambda_{ss}(1 - r)$ for the selected clone and hence equally good fits to the measured VAF histogram). We find that the inferred division rate of normal stem cells is very similar when using Model B instead of Model A (**Figure R2**). Hence, even when we link the selective advantage of a CH clone with an increased division rate of mutated stem cells (instead of a smaller loss rate), also the unmutated HSCs in the CH individual are inferred to have a higher division rate than in the majority of the neutral cases. Importantly, this result shows that our inference of HSC division rate is not biased by how we specifically implement the selective advantage of the CH mutation (which is not known experimentally *in vivo*).

Reviewer #1:

Remarks to the Author:

In their study, Körber et al., propose a unique method, SCIFER, to infer the population dynamics of tissue stem cells, discriminating between neutral evolution vs. positive selection, on the basis of the distribution of the cumulative number of SNVs according to VAF value in a given tissue sample using whole genome sequencing. The authors first modeled the clonal dynamics under neutrality and positive selection, to which the observed data in bone marrow CD34+ cells were fitted to investigate the dynamics of hematopoiesis with and without clonal hematopoiesis. After benchmarking SCIFER using published single colony sequencing data, the authors applied the method to newly obtained whole genome sequencing data of bulk bone marrow samples from 22 cases, including seven cases without evidence of clonal selection, 9 known CH mutations, 10 with ≥ 1 CH mutations, and 5 showing clonal selection with no known CH-mutations. The authors found accelerated stem cell division of all stem cells in CH, compared to age-matched non-CH cases, suggesting altered bone marrow environment in CH positive cases. Finally, they concluded that SCIFER can be applied to other renewing somatic tissues to detect and quantify clonal selection. The results are interesting and potentially merit publication. Meanwhile, this reviewer raises several issues to be addressed before the decision is made.

We thank the reviewer for the thoughtful and constructive evaluation review. The manuscript has been improved and expanded by addressing their comments.

6) Major issues: The model of clonal selection seems to assume one selected clone, but some cases used for benchmarking had ≥ 2 major clones. How were these cases analyzed or detected by SCIFER? How do we infer the hematopoietic stem cell (HSC) dynamics? SCIFER is a unique and interesting method. Meanwhile, most of the results obtained using SCIFER have been previously described using colony sequencing combined with bulk sequencing (refs #2, #22, #26, #33 add refs properly). The increased cell division rate of all stem cells could be novel but needs further validation.

We thank the reviewer for raising this important point. It supported us in developing the two-clone version of SCIFER, as explained in response to Point 1 on p6-7 of this Response to the Reviewer's Comments. Specifically, for the benchmarking case with selection KX004¹ we detected a second selected clone and of interest. The key inference from SCIFER on hematopoietic stem cell (HSC) number was extended Data Fig 3j) agree with previous results, using colony sequencing, from one individual³ and the timing of birth of selected clones from three individuals^{1,2}.

Minor issues:

The Reviewer asks if we could validate the increased division rate seen in several CH cases.

7) How were the parameters inferred by parameter fitting for those cases having multiple selected clones. The iterations with parameter estimation using ABC in page 45 are not detailed, where a minimum description of the ABC framework is needed for most readers to understand how iteration works. Moreover, the expression of $M(t)$ in Eq 10B might not be correct, where σ explicitly appears in Sig (t).

The reviewer asks how SCIFER inferred the parameters for cases with multiple selected clones. We employed different experimental and computational approaches to validate the HSC division rate, but found that none of them could address the question. We tried:

The original version of SCIFER accounts for a single selected clone only, and all model fits presented in the original version of the manuscript were obtained with this one-clone model.

Using a telomere-based SCIFER, we compared the inferred parameters of HSCs, including the expression parallel, and applied this extended model to newly generated WGS data and telomere length data. It is important to note that the HSC parameters and selection parameters of the largest selected clone is practically applied (Figure 8). However, we note that a technically similar analysis of telomere lengths in large cohorts from the UK Biobank and the TOPMed consortium revealed, on average, accelerated telomere shortening in leukocytes from individuals with CH, which would be consistent with an increased division rate in all stem cells. This raises the question whether adequate telomere length is a requirement for CH to be detected. A recent large population study showed that longer telomeres are a causative risk factor for CH⁶. Moreover, germline carriers of the POT1

Regarding the second point raised by the reviewer, we apologize that our description of the ABC framework was unclear. We have now added details to the description of parameter estimation in the Parameter estimation section within Methods section on **p56-58, lines 1393-1461** with the additional text in red font.

We thank the reviewer for pointing out a typo in L1080, which we have corrected in the revised manuscript.

8) Derivation of several formula in the method section need more explanation to easy follow-up. For example, it might not be easy to follow the transformation to (18) (+1061). Every expression should be rigorously defined without any ambiguity. L1085-1088, f is not well-defined. Does it express the position of bin? Pois(c)(L1109).

We apologize for a lack of clarity in the Methods section, which now extensively revised with changes in red font. We outline changes to the passages specifically mentioned by the reviewer below.

We now explain the deduction of Equation 18 in more detail (**p51, lines 1255-1270**). This equation models the site frequency spectrum of variants acquired prior to the generation of the selected clone, $S_{i,1}$. These variants may be inherited to the selected clone, in which case they will be present in all selected cells and, additionally, in some of the normal cells. Alternatively, these variants may be present in normal cells only. To distinguish the two cases, we consider a variant that is present in k cells when the driver mutation occurs at t_s . Assuming that the driver mutation is acquired in a random cell, the probability of this variant to be inherited to the selected clone is k/N_{ss} , where N_{ss} is the size of the stem cell pool. The alternative case, where the variant is exclusively present in normal cells, has probability $1 - k/N_{ss}$. In the former case, all $n_2(t)$ selected cells will harbor the variant at the time of measurement, t ; in addition, the $k - 1$ normal stem cells harboring the variant at time t_s may reach a clone size between 0 and $N_{ss} - n_2(t)$, according to the drift dynamics of normal stem cells. In the alternative case, the variant does not end up in the selected clone and hence solely drifts in the normal cells. Taken together, this yields Eq. 18:

$$S_{i,1}(t) = \begin{cases} \sum_{k=1}^{N_{ss}} S_k(t_s) \left[\begin{array}{l} \text{variants drift in normal cells \& present in all selected cells} \\ \frac{k}{N_{ss}} P_{\text{exp},k-1,i-n_2(t)}(t - t_s | \lambda_{ss}, \delta_{\text{eff}}) \\ \text{variants not present in selected cells} \\ + \left(1 - \frac{k}{N_{ss}}\right) P_{\text{exp},k,i}(t - t_s | \lambda_{ss}, \delta_{\text{eff}}) \end{array} \right], i \geq n_2(t), \\ \sum_{k=1}^{N_{ss}} \left(1 - \frac{k}{N_{ss}}\right) S_k(t_s) P_{\text{exp},k,i}(t - t_s | \lambda_{ss}, \delta_{\text{eff}}), i < n_2(t), \end{cases}$$

where $P_{\text{exp},a,b}(t)$ is the probability to drift from a clone size a to a clone size b within time t , λ_{ss} and δ_{eff} are the division and loss rates of normal stem cells, respectively, and $S_k(t_s)$ is the site frequency spectrum of all somatic variants at t_s .

The reviewer also asks about a more rigorous definition of f in former L1085-1088. Indeed, we here refer to the position of a bin in the experimentally determined variant allele frequency spectrum. We now explain this in more detail on **p57, lines 1422-1428**.

Finally, the reviewer asks about Pois(c) in former L 1109. Here, we simulate a sequencing coverage at each locus by drawing random coverages from a Poisson distribution. We now explain this better on **p58, lines 1452-1453**.

We thank the reviewer for pointing us to these passages and hope that the Methods section has now become clearer.

9) Legends in Fig. 3 are incorrect. For example, the legend for f should be that for h?

We thank the reviewer for pointing this out. We apologize for any confusion caused by this error and have now corrected the legend.

Reviewer #2:

Remarks to the Author:

In their manuscript Korber et al. present a statistical method to study mutation dynamics using bulk whole genome sequencing data. Their method is similar to other previous work (e.g PACER). However in addition to leveraging somatic mutation burden to infer age at driver acquisition, they also use VAF distribution taken from WGS to infer mutation selection and division rate. They use simulations and publicly available data to show that application of SCIFER to bulk WGS is able to generate similar results to single cell sequencing data. While some of the math is beyond my comfort level, the overall approach seems reasonable with a few points that I think could be clarified that I detail below. The application of their method to existing data does not reveal new insights into clonal hematopoiesis. The impact of the method could come from application of the method to higher depth WGS datasets where their simulations suggest SCIFER could be used to infer clonality with a similar resolution to single cell sequencing.

We thank the reviewer for the careful reading of our manuscript and the constructive evaluation.

We understand that the reviewer makes two points:

1. The Reviewer states that the impact of SCIFER could come from application to higher depth WGS datasets. We fully agree. In the revised manuscript we provide new data that confirms the Reviewer's astute observation:

a. Re-sequencing our bone marrow samples to a total coverage of 270x. Please refer to our response to Reviewer 1 above, where we set out the data in more detail (see **p5-6 of the Response to Reviewers**).

b. To best exploit these high-resolution bulk data, we have developed a new version of SCIFER that allows the inference of two selected clones. We would like to kindly direct the reviewer to our response to Reviewer 1 above, where we set out the data in more detail (see **p6-7 of the Response to Reviewers**).

c. In addition, we have applied SCIFER to published deep WGS data from 185 human brain samples from 131 individuals aged 4-95 years¹⁰. This cohort comprised samples from 44 normal (neuro-typical) individuals and 87 individuals with neurological disorders (autism spectrum disorder, schizophrenia, Tourette syndrome), taken from different brain regions (cortex, striatum, hippocampus). In our response to Reviewer 1 above, we set out the data in more detail (see **p7-8 of the Response to Reviewers**)

2. The Reviewer states that our study does not reveal new insights into clonal hematopoiesis. We would respectfully suggest we do and have further strengthened these points in the revised manuscript. The new information is:

A. The most comprehensive quantitative study to date on human HSC numbers, rate of SNV acquisition with each HSC division and HSC division rate in 22 individuals, encompassing neutrally evolving and selected clones with both known and unknown drivers.

B. Based on this quantification, we provide new insight into the age-incidence of selection events in hematopoiesis. We show that the acquisition of selected clones in blood, whether they arise by known or unknown drivers, occurs at an approximately constant rate. The near doubling of the number of selected clones we now identified (28 compared to 15 in the first version of the manuscript) substantiated our finding that selected clones begin to originate already in infancy and continue to do so with approximately constant rate throughout human life (**new Fig. 6d**).

C. We provide evidence for a time-averaged increase in division rate of all HSCs in samples with selected clones.

Finally, the reviewer also remarks that “Their method is similar to other previous work (e.g PACER)¹⁵.” Here, we respectfully disagree with the reviewer. We believe that SCIFER is an important methodological improvement and provides a considerably more accurate method to distinguish selection from drift. We set out our reasoning below.

- In PACER, selection is quantified by fitting a heuristic linear regression model to the passenger mutation counts of a patient collective. In this regression model, the number of passenger mutations is the dependent variable and age at blood draw, VAF of the CH driver and the selective advantage are patient-specific independent variables. In contrast to this heuristic setup, SCIFER explicitly models the molecular and cellular processes of mutation acquisition, drift and selection. This has two advantages: First, the detection of selection by SCIFER is based on evolutionary mechanism and hence more accurate than achieved by PACER (see next point). Second, SCIFER is able to quantify the underlying HSC parameters: HSC number, division rate and mutation rate.
- SCIFER uses information from the entire VAF distribution of somatic variants, whereas PACER only relies on the total number of subclonal variants to detect clonal selection. However, our analyses show that the number of subclonal variants is not always an indicator of clonal selection; rather, clonal selection manifests itself in a subclonal shoulder in the somatic VAF distribution (in the revised manuscript we now show in detail how inter-individual heterogeneity in the number of stem cells and their division rate may have marked impact on the number of variants detectable in a tissue; c.f. **Fig. 1d-g,j-l** and **new Extended Data Fig. 1a-c**). Indeed, we identified clonal selection in three cases (2-U, 8-UU and 14-U, **new Fig. 5a**) with comparable SNV counts as in neutrally evolving cases (1-N, 3-N, 4-N and 15-N, **new Fig. 5a**). These cases would not have been identified as selected when solely taking the number of variants into account and hence been missed by PACER.
- PACER is set out to quantify the selective advantage of known driver mutations but does not offer means to detect clonal selection in the absence of known drivers. By contrast, SCIFER identifies clonal selection without knowledge of the underlying driver mutation and can hence aid the detection of as yet unknown driver mutations.
- While PACER infers individual selection parameters from a cohort of individuals, SCIFER quantifies drift and selection for each sample separately. To give an example, PACER does not take into account inter-individual heterogeneity in the passenger mutation rate, whereas SCIFER infers this parameter for each sample separately. Thus, SCIFER quantifies drift and selection genuinely at the level of individuals.

1. Their method does not seem to take into account potential differences in selection of CH mutations over the life course of an individual. This has been suggested to occur on a gene specific-basis in CH.

We fully agree. Prior analysis of VAF in serial samples longitudinally suggest that different genetic drivers may impart distinct kinetics of cell division. Namely, the growth rate of *DNMT3A* clones may slow down with time, *TET2* clones expand at a constant rate over time, and finally, *SRSF2* and *JAK2* clones not only arise late in life but expand faster². Therefore, in recognition of this point, we stress that a limitation of our work is that our inferred cell division rates are time-averaged as we have single time point sample analysis. It would be valuable to apply SCIFER to serial longitudinal samples to test how stem cell dynamics and parameters change with time with different drivers, including currently unknown drivers for which there is currently no such information. This is set out on in the Discussion: **p14, lines 473-480**.

2. In Figure 3l and p I am unsure what “stem cell number” represents and how this was calculated.

In Fig. 3l and p we show the number of HSCs inferred by SCIFER. We agree that the term “stem cell number” might be misleading and re-labeled the axes with “Number of HSCs”.

SCIFER infers the number of HSCs from the shape of the cumulative VAF distribution of somatic variants and we provide details on parameter identifiability in Supplementary Note 2. To give intuition on how the number of stem cells, N_{ss} , influence the measured VAF distribution, we now show simulated cumulative VAF distributions for a wide range of stem cell numbers, starting with a relatively small stem cell pool of 5,000 cells up to a large pool containing 1M stem cells in the **new Extended Data Fig. 1a**. These simulations show how drift alters the VAF distribution distinctly depending on the size of the stem cell pool. Of note, drift effects disappear from the measurable range if stem cell numbers become very large. We reference these additional simulations in the revised Results **p4, lines 123-124**.

3. CH mutations including TET2 are known to impact self-renewal. This increased replicative capacity has been cited to explain why individuals with CH have shorter telomeres (yet longer telomeres predispose to CH). Thus, an increased HSC division rate would be expected.

We completely agree. We note that analysis of telomere lengths in large cohorts from the UK Biobank and the TopMed consortium revealed, on average, accelerated telomere shortening in leukocytes from individuals with CH⁵, which would be consistent with an increased division rate in all stem cells. This raises the question whether adequate telomere length is a requirement for CH to be detected. A recent large population study showed that longer telomere length is a causative risk factor for CH⁶. Moreover, germline carriers of the POT1 mutation that results in longer telomeres have a very high rate of CH⁷. Collectively, these observations are consistent with long telomeres being a permissive requirement for CH, arguably by allowing CH HSC to undergo enhanced cell division. We have added these important insights into the Discussion (**p14, lines 481-492**).

References:

- 1 Mitchell, E. *et al.* Clonal dynamics of haematopoiesis across the human lifespan. *Nature* **606**, 343-350 (2022).
- 2 Fabre, M. A. *et al.* The longitudinal dynamics and natural history of clonal haematopoiesis. *Nature* **606**, 335-342 (2022).
- 3 Lee-Six, H. *et al.* Population dynamics of normal human blood inferred from somatic mutations. *Nature* **561**, 473-478 (2018).
- 4 Feuerbach, L. *et al.* TelomereHunter—in silico estimation of telomere content and composition from cancer genomes. *BMC bioinformatics* **20**, 1-11 (2019).
- 5 Nakao, T. *et al.* Mendelian randomization supports bidirectional causality between telomere length and clonal hematopoiesis of indeterminate potential. *Science advances* **8**, eabl6579 (2022).
- 6 Kar, S. P. *et al.* Genome-wide analyses of 200,453 individuals yield new insights into the causes and consequences of clonal hematopoiesis. *Nature Genetics* **54**, 1155-1166 (2022).
- 7 DeBoy, E. A. *et al.* Familial clonal hematopoiesis in a long telomere syndrome. *New England Journal of Medicine* **388**, 2422-2433 (2023).
- 8 Gabbutt, C. *et al.* Fluctuating methylation clocks for cell lineage tracing at high temporal resolution in human tissues. *Nature Biotechnology* **40**, 720-730 (2022).

- 9 Gabbutt, C. *et al.* Evolutionary dynamics of 1,976 lymphoid malignancies predict clinical outcome. *medRxiv*, 2023.2011.2010.23298336 (2023).
<https://doi.org/10.1101/2023.11.10.23298336>
- 10 Bae, T. *et al.* Analysis of somatic mutations in 131 human brains reveals aging-associated hypermutability. *Science* **377**, 511-517 (2022).
- 11 Buccoliero, A. M. *et al.* Pediatric high grade glioma classification criteria and molecular features of a case series. *Genes* **13**, 624 (2022).
- 12 Kurdi, M. *et al.* The cancer driver genes IDH1 and IDH2 and CD204 in WHO-grade 4 astrocytoma: crosstalk between cancer metabolism and tumour associated macrophage recruitment in tumour microenvironment. *Biologics: Targets and Therapy*, 15-22 (2023).
- 13 Busch, K. *et al.* Fundamental properties of unperturbed haematopoiesis from stem cells in vivo. *Nature* **518**, 542-546 (2015).
- 14 Sun, J. *et al.* Clonal dynamics of native haematopoiesis. *Nature* **514**, 322-327 (2014).
- 15 Weinstock, J. S. *et al.* Aberrant activation of TCL1A promotes stem cell expansion in clonal haematopoiesis. *Nature*, 1-11 (2023).

Reviewers' Comments:

Reviewer #1 (Remarks to the Author):

In their revised manuscript, Körber et al. fully addressed the comments and questions this reviewer raised. In addition, the authors newly applied SCIFER to a published dataset of deep WGS for 185 brain samples from 131 donors, including 44 normal donors and evaluated positive selection in the brain, describing the inferred clonal dynamics of IDH2-mutated clones in a 62-year-old female and the early initiation of clonal selection in the brain. This reviewer raises only a few further questions regarding the new results.

We thank the Reviewer for their careful reading of our revised manuscript, their insightful feedback and the highly pertinent additional questions.

1. The authors have newly developed a two-clone model. Does it improve the detection of clonal selection in the presence of multiple clones compared to the original single-clone model? Is the improvement limited to the cases with deep WGS?

The Reviewer raises the interesting question whether the two-clone model improves the detection of clonal selection as compared to the one-clone model in the presence of multiple clones.

What we did in our study was to perform an iterative approach to identify clonal selection, starting with the one-clone model to identify the leading selected clone. Subsequently, we re-analyzed the cases with the two-clone model. When applying the two-clone model, this allowed us to use constrained priors for the size of the leading selected clone that were informed by the result obtained with the one-clone model (Supplementary Table 10). This is an advantage of the iterative approach. Therefore, however, a direct comparison between the detection of clonal selection with the one-clone model and the two-clone model is not possible based on the analysis we present in our manuscript.

To specifically address the Reviewer's question, we re-analyzed the 10 samples (samples are shown Figures 4e and 5c) with multiple clones at both 90X and 270X, using both the one-clone and two-clone models with unconstrained priors (log-uniform between 0.1% and 100% clone size) and, for comparison, also the two-clone model with constrained priors (the iterative approach described above), and asked how often we detect clonal selection.

At 270x, both the one-clone model and the two-clone model detected at least one selected clone in all of the 10 samples, irrespective of whether we used constrained or unconstrained priors (Review Table 1). Thus, at 270x, the detection of the leading selected clone, in the presence of multiple clones, does not improve with the two-clone model as compared to the one-clone model.

A second selected clone was detected in all 10 samples when using the two-clone model and constrained priors from the detection of the first clone. With the two-clone model with unconstrained priors we detected the second clone in only 4 out of the ten samples, showing the greater sensitivity of the iterative approach.

Review Table 1. Evidence for selection in the 10 cases with multiple selected clones of either a leading clone, or at least 2 selected clones, using different implementations of SCIFER at 90x or 270x WGS.

Coverage	Evidence for selection of a leading clone			Evidence for selection of a second clone	
	One-clone model	Two-clone model		Two-clone model	
	Unconstrained prior	Unconstrained prior	Constrained prior	Unconstrained prior	Constrained prior
270x	10/10	10/10	10/10	4/10	10/10
90x	9/10	7/10	8/10	1/10	1/10

At 90x, the one-clone model detected the leading clone in 9 of the 10 samples. The two-clone model detected a leading clone in 7 samples when using unconstrained priors, and 8 samples when using

constrained priors (Review Table 1). At 90X, a second selected clone was only identified in 1 sample, regardless of whether unconstrained or constrained priors were used.

Overall, our results suggest that the detection of the leading selected clone in the presence of multiple clones is very similar with the one-clone model and the two-clone model. As expected, we found that the two-clone implementation of SCIFER is more informative at 270x. We added this observation to the results section (**p.10, lines 326-328**). Finally, our results suggest that detection of a second selected clones is facilitated by an iterative approach, where the leading clones are first detected using the one-clone model and additional selected clones are detected when re-running SCIFER with constrained priors for the size of the leading selected clone. We added this observation to the Methods (**p. 57, lines 1430-1431**).

2. In contrast to hematopoietic samples, brain tissue, and most of other tissues, consist of heterogeneous cell components of different cell of origin, including astrocytes, oligodendrocytes, neurons, microglial cells, etc.. It is therefore informative to discuss how this heterogeneity in cell type affects the detection of clonal selection, the interpretation of cell division/mutation rates, and the number of stem cells. Do the inferred values for these variables make any sense? In other words, what cell types do these values refer to? This is a general problem to be answered when SCIFER is applied to solid tissues, which are typically composed of many cell components, immune cells, endothelial cells, blood components, fibroblasts, and epithelial cells. Does this cause any limitation?

The Reviewer raises the important question of how SCIFER performs on unsorted tissue samples. This is an important question, especially with relation to the brain samples, where neurons, astrocytes and oligodendrocytes ultimately derive from neural stem cells (NSCs, although these lineages might be supplied by independent progenitors downstream of NSCs in adult glio/neurogenesis and are likely to have different turnovers), whereas microglia is primarily of fetal hematopoietic origin. Microglia constitute a minority of glial cells (~7%) across different brain regions¹ and hence the vast majority of microglial somatic variants will be hardly detectable even at 270x and hence not influence our inference. However, astrocytes, forming the majority of glial cells in the cortex in adult humans, will turn over whereas cortical neurons hardly do. To assess how somatic variants accumulate and drift in such a tissue, we have performed additional stochastic simulations with a multipotent stem cell pool that generates two cell types during development. During adulthood, however, only one of the two cell types is renewed, whereas cells of the second lineage remain static. The two lineages could for example correspond to astrocytes and neurons in the human brain, and we assumed that they constitute 60% and 40% of the tissue, respectively.² Our simulations suggest that variant allele frequencies in the renewing cell population mirror the dynamics in the stem cell population, whereas variant allele frequencies in the static population differ from those in stem cells. Variant allele frequencies in the tissue overall are thus shaped by the average of the two populations. As a consequence, SCIFER will estimate the average cell renewal in the tissue, which will overestimate the renewal of the static fraction and underestimate the renewal of the renewing population. Therefore, we think that analyses on unsorted tissue can serve as a starting point to understand the stem cell dynamics in a heterogeneous tissue on average. However, we fully agree with the Reviewer that a more accurate quantitation of the stem cell dynamics in the human brain will require a separate analysis of sorted cell populations.

We have added these additional simulations in the new Supplementary Data 1, together with a new Supplementary Figure 1e-h. We refer to these simulations in the results section (**p. 13, lines 446-450**). Moreover, we now discuss the limitations of our inferences on heterogeneous cell populations in the Discussion (**p. 16, lines 545-553**).

3. In particular, multiple mutations were detected in NC7, including those affecting DNMT3A (1.91%), BCORL1 (2.99%), and NRAS (1.84%), in addition to an IDH2 (3.55%) mutation, but only in a small fraction of the sample with small VAFs (ref#17). Thus, it is impossible to determine if these mutations are in the same clone or not, whereas in Fig.7b, SCIFER analyzed the data as if they were found in the same clone. All of these mutations were common in myeloid neoplasms, while DNMT3A mutations were rarely found in glioblastomas. Thus, these mutations may not reside in the neural tissue but some of them might be present in contaminated blood cells.

The Reviewer asks the excellent question of how confident we can be that the putative driver mutations identified in patient NC7 reside in the same clone, and in the neuro-glial fraction of the brain tissue. In Bae et al., Science, 2022, this patient was studied in detail. Fortunately, multiple cell populations were purified (FAC-sorted) to determine the cellular fraction harboring these mutations. As correctly noted by

the Reviewer, the putative driver mutations were detected at low VAF (1-4%) in bulk tissue. However, when sorting for NeuN⁺/Sox10⁺ cortical oligodendrocytes and NeuN⁺/CITP2⁻ striatal interneurons, the authors measured these mutations at markedly higher VAF [*IDH2* (15%), *BCORL1* (7%) in cortical oligodendrocytes, *IDH2* (22%), *BCORL1* (16%), *NRAS* (10%) and *DNMT3A* (6%) in striatal interneurons]. These data suggest that the putative driver mutations in *IDH2*, *BCORL1*, *NRAS* and *DNMT3A* indeed arose in the neuro-glial fraction rather than contaminating blood cells. Crucially, we show our SCIFER results on these sorted populations in Fig. 7b and we emphasize this point on **p. 12 (lines 417-418)**.

We fully agree with the Reviewer that SCIFER cannot uniquely distinguish whether all driver mutations are in the same clone or whether different clones co-occur at similar sizes. In cortical oligodendrocytes, SCIFER infers a selected clone at 6-9% VAF that agrees with the VAFs of the mutations in *IDH2* and *BCORL1* (within credible intervals). By contrast, in striatal interneurons, SCIFER infers a selected clone at 15% VAF that agrees with the VAFs of all four mutations in *IDH2*, *BCORL1*, *NRAS* and *DNMT3A* (within credible intervals). Together, these data suggest an evolutionary path where an initial clone was driven by mutations in *IDH2* and *BCORL1*, with subsequent clonal evolution with one or two subclone(s) driven by mutant *NRAS* and *DNMT3A*. Nevertheless, we agree that we cannot formally rule out that two or three independent clones co-occur at indistinguishable VAF in striatal interneurons. We have added text to reflect this uncertainty to the results section (**p. 12, lines 425-429**).

Reviewer #3 (Remarks to the Author):

Korber et al. develop a method (“SCIFER”) that uses relatively deep WGS data to detect the presence of a clonal expansion and infer important parameters governing the clonal evolution in blood including (i) the occurrence time and expansion rate of the positively selected clone (if detected) and (ii) parameters governing the neutral dynamics of the stem cells such as population size (N), symmetric division rate (λ), and mutation rate (μ). They then apply this method to various clinical samples to show that it recapitulates results from more comprehensive (and expensive!) phylogenetic approaches and apply it to new samples to distinguish between samples governed largely by neutral dynamics and those showing clear clonal expansions. They also use it to detect the presence of clonal expansions with no known driver (as previously described by Poon et al. 2021 and Mitchell et. al. 2022).

Overall this is beautiful work and outstanding in its quality and presentation. There are a number of original developments. The application of proper population genetic theory to the WGS data offers a rational approach to detecting clones under positive selection in WGS and therefore improves significantly on previous approaches such as “PACER” (or the original Williams et al. Nature Genetics 2018 on which PACER is based). The key advance of this method in my view is the ability to infer parameters governing the clonal evolution of blood on an individual basis rather than using ensembles of individuals (as the data scale is provided by the breadth of WGS and does not require thousands of individuals to develop statistically meaningful distributions). The major biological results largely confirm previous estimates for parameters governing stem cell dynamics in blood (Lee-Six 2018, Watson 2020 and Mitchell 2022) and I see this as a good validation of the method. The method as applied to the blood data suggests a possibly interesting result linking selection to alterations in HSC division rates would be novel and very interesting — though as outlined below in more detail I think it is likely this may be explained by other effects (as should be mentioned in the discussion).

I wish to say at the outset that wholeheartedly recommend this be published. I have two comments on the method and its application to the data that I think would be good for the authors to consider mentioning in the revision. The authors have already attempted to address point 1 below in the revision, but not point 2. I suspect both of my points below could be addressed via the discussion to caution the reader against over-interpreting some of the results. Overall, I commend the authors for their beautiful work.

We thank the Reviewer for his careful evaluation of our manuscript and his positive and supportive feedback. This is very meaningful to us given his expertise in the field.

The issue of oligoclonal expansion. The biggest limitation of the method in my opinion is that it does not appear to handle oligoclonal expansion i.e. when there are multiple independent clones that are expanding in the blood (other reviewers pointed this out). As can be seen from almost all of the people in Mitchell 2022, when expansions occur they are almost always quite oligoclonal. As I understood it the method only fits a single dominant clonal expansion (note the the updated version now handles 2

clones). This is quite a limitation because the model presented here will mistake oligoclonal expansion — which lead to too many SSNVs at moderate VAFs but with no clear “jump” in the cumulative distribution — for stronger genetic drift. In samples where there is oligoclonal positive selection the model effectively fits a larger HSC division rate (or smaller N). I think this effect explains some of the reported association between samples with selection and increased HSC division rates. For example if we take the results in Figure 4c (N4 and N5). The authors interpret this “tick up” of the cumulative at lower VAF as being driven by drift and in order to explain why it occurs at VAFs of ~5% they infer a stem cell number (in units of the symmetric division time N/λ) to be about ~10-fold lower than other samples. An alternative explanation for this effect (and a more likely explanation in my opinion) is that this tick up is driven by positive selection on multiple clones, none of which causes a clean “step” in the cumulative. The reason the model prefers to fit a neutral theory to these data (with slightly wonky parameters) rather than positive selection is because the theory for positive selection does not include oligoclonal effects — hence it prefers to infer neutrality with anomalously small parameters for N/λ . I’m not suggesting the authors redo the theory to include this effect (though they now have tried this in the revision) but they should make it clear that in samples where there is oligoclonal expansions (which will be most samples!) that this effect could cause pretty large skews in their estimates for stem cell parameters and therefore those should be treated with caution.

This effect is also present in the pseudo-bulk analysis of the data from Lee-Six 2018 / Mitchell 2022 data in Figure 3. For example, the tree from Lee-Six et al. has a single 5-mer clone, a single 4-mer clone and 3 independent 3-mer clones that arose in adulthood (after 20-years of age). The original paper essentially only fitted to the 2-mers and therefore incorrectly concluded that the evolution in this person was entirely neutral. In fact it is very unlikely that the 5-mer or the 4-mer or 3x 3-mer clones can arise through drift using their own inferred parameters. In these cases it is more likely to be the beginning of an oligo clonal expansion. SCIFER will be quite sensitive to these effects and this is likely part of the reason why the inferences from SCIFER applied to this example infer N/λ to be about a factor of 4 lower than the inferences from the original paper.

This effect is also likely present in Figure 6. For example in U2 and U3 (samples where there is no clear “step” in the cumulative). In effect the model will have an instability where it goes from inferring a selected clone at VAFs close to the limit of detection (~5%) to a scenario where it says things are neutral, but with wonky parameters. This instability will derive from the fact the the underlying two hypotheses (one major selected clone vs entirely neutral) are not accounting for the actually most likely scenario which is oligoclonal positive selection.

We thank the Reviewer for succinctly pointing out the question of oligoclonal expansions and whether selection of small clones can be erroneously attributed to drift by SCIFER, thus potentially biasing parameter estimates.

Indeed, we found that our deeper sequencing, and developing a multi-clone model, shed light on this question, as explained in detail below. Both cases singled out by the Reviewer, N4 and N5 in the original version of the manuscript, are now found to contain small clones with unknown drivers (and hence are now relabeled as 11-UU and 13-U, respectively). This demonstrates that SCIFER is able to detect such small clones (and, in principle, can also be applied to more than two clones). We agree with the Reviewer that this issue needs proper consideration, especially for parameter estimation, and we have added a discussion of this point on **p.15 (lines 511-517)**. In particular, we have pointed out that clones at the resolution limit of sequencing which are not detected by SCIFER may bias the parameter inference to show, erroneously, a drift effect.

In more detail regarding the two samples, labeled N4 and N5 in our initial submission, the Reviewer suggests that clonal selection may have been mistaken as drift. In the first revision, we had re-sequenced these samples to a higher coverage of in total 270x and **indeed detected small selected clones in both of them**, in line with the Reviewer’s very astute observation. The discovery of additional selected clones at higher sequencing depth led us to re-label our samples during the first revision so that the samples originally labeled as N4 and N5 are now identified as 11-UU and 13-U. We sincerely apologize for any confusion that may have arisen from this. For reference, the posterior probabilities for clonal selection with a one-clone model at 90x and a two-clone model at 270x can be compared in the new Fig. 5b,c; the corresponding model fits are shown in Extended Data Fig. 8a,b. Interestingly, the inferred stem cell parameters did not change markedly in these cases when comparing the model fits obtained with 90x data (where no selected clones were identified) to those obtained with 270x data

(where selected clones were identified). However, the inferred stem cell numbers were indeed higher at 270x, though this difference was not significant (Fig. 5h).

Separation of the population size and division rate. I am sorry to be a stick-in-the-mud but I remain unconvinced that the HSC symmetric division rate and the population size parameters can be robustly separated using the approach described. The arguments given were (i) use of absolute counts of variants and (ii) including the developmental phase in the model. I don't think either of these can give an estimate of the symmetric HSC division rate in the homeostatic phase which is what is needed to separately infer N and λ . The first argument falls down because in adulthood the only robust estimate for mutation rates in HSCs comes from WGS of single cells from someone of a given age. This means we have very robust estimates for the mutation rate *per year* (~20 per cell per year) but no direct way of estimating the mutation rate per symmetric division. The second argument falls down because while there are good estimates of the mutation rate per symmetric division during development (2-3 per HSC per division) this does not tell one anything about the mutation rate per symmetric division in adulthood. I am of course biased because I do not believe separating these parameters robustly is possible, but I am open to being proved wrong on this. However I did not find the arguments or data presented to be very compelling and it was hard to determine exactly how the inferences worked. I think the authors should make it clear in the discussion that the ability to separate N and λ relies on some assumptions which may or may not hold.

We thank the Reviewer for his careful evaluation of the theory underlying our parameter inference. As correctly pointed out by the Reviewer, our inference of the steady-state stem cell number, N_{ss} , and the stem cell division rate, λ_{ss} indeed relies on the developmental phase. Here, we make the assumption that mutation rate remains unchanged over the human lifespan. We fully agree that this assumption should be pointed out more clearly, not only in the Methods and have now stated this explicitly in the main text (p. 7, lines 229-230) and in the Discussion (p. 14, lines 494-499). We are not aware of experimental data that speak to this question in the hematopoietic system. However, we would like to add that our assumption is consistent, at least by order of magnitude, with the mutation load in human hematopoietic stem and progenitor cells in newborns reported by Abascal et al. (2021)³ when factoring in rapid cell division during development.

The Reviewer also points out that mutation rates in adult HSCs have been estimated from single-cell phylogenies, but notes that these estimates measure the number of mutations acquired per year rather than per cell division. We fully agree with this. We would like to clarify that we did not use any published estimate for the mutation rate μ in our inference. The use of the absolute number of mutations yields an estimate for the product between the mutation rate and the total number of stem cells during homeostasis (μN_{ss} ; equation A.4 in Supplementary Note 2). This product is solely inferred from the available mutation data, with no prior estimate for mutation rates in HSCs being used. Separation between μ and N_{ss} in our inference then relies on the developmental phase assuming that mutation rates remain constant over the human lifespan, as discussed above.

Best,

Jamie Blundell

References

- 1 Dos Santos, S. E. *et al.* Similar Microglial Cell Densities across Brain Structures and Mammalian Species: Implications for Brain Tissue Function. *The Journal of Neuroscience* **40**, 4622-4643 (2020). <https://doi.org:10.1523/jneurosci.2339-19.2020>
- 2 Azevedo, F. A. *et al.* Equal numbers of neuronal and nonneuronal cells make the human brain an isometrically scaled-up primate brain. *Journal of Comparative Neurology* **513**, 532-541 (2009).
- 3 Abascal, F. *et al.* Somatic mutation landscapes at single-molecule resolution. *Nature* **593**, 405-410 (2021).